# Climate velocities and species tracking in global mountain regions

Wei-Ping Chan[1,2,3,4], Jonathan Lenoir[5], Guan-Shuo Mai[1], Hung-Chi Kuo[6], I-Ching Chen[7,8 ✉] & Sheng-Feng Shen[1 ✉]

Mountain ranges contain high concentrations of endemic species and are indispensable refugia for lowland species that are facing anthropogenic climate change[1,2]. Forecasting biodiversity redistribution hinges on assessing whether species can track shifting isotherms as the climate warms[3,4]. However, a global analysis of the velocities of isotherm shifts along elevation gradients is hindered by the scarcity of weather stations in mountainous regions[5]. Here we address this issue by mapping the lapse rate of temperature (LRT) across mountain regions globally, both by using satellite data (SLRT) and by using the laws of thermodynamics to account for water vapour[6] (that is, the moist adiabatic lapse rate (MALRT)). By dividing the rate of surface warming from 1971 to 2020 by either the SLRT or the MALRT, we provide maps of vertical isotherm shift velocities. We identify 17 mountain regions with exceptionally high vertical isotherm shift velocities (greater than 11.67 m per year for the SLRT; greater than 8.25 m per year for the MALRT), predominantly in dry areas but also in wet regions with shallow lapse rates; for example, northern Sumatra, the Brazilian highlands and southern Africa. By linking these velocities to the velocities of species range shifts, we report instances of close tracking in mountains with lower climate velocities. However, many species lag behind, suggesting that range shift dynamics would persist even if we managed to curb climate-change trajectories. Our findings are key for devising global conservation strategies, particularly in the 17 high-velocity mountain regions that we have identified.

Mountainous regions represent 25% of Earth's land surface and are rich in biodiversity, owing in part to their steep climatic gradients and complex topography[1,2]. The assumption that mountain species are responding faster to anthropogenic climate change through rapid upward range shifts leading to potential mountaintop extinctions has attracted extensive research[3,4,7-9]. Whether species are closely tracking the rate of climate warming is assessed chiefly by comparing the velocities of species range shifts with the velocities of climate change; that is, the rates at which isotherms move through the geographical space[3,4,10-12]. Past studies that assessed climate velocities have focused mainly on horizontal velocities, in km per year; that is, how fast isotherms are moving along the latitudinal and longitudinal clines of the horizontal plane (see the seminal work from Loarie et al.[12] for terrestrial systems; this was then applied to marine systems by Burrows et al.[13]). Because isotherms are located closer to one another in mountainous regions, horizontal velocities of isotherm shifts are much slower and potentially omnidirectional in mountains, whereas they are much faster and oriented mainly poleward in the lowlands[13]. However, we know that climate warming also causes terrestrial species to shift along mountain slopes and thus not only horizontally but also 'vertically' when projected along elevation gradients—moving at very different speeds (usually expressed in m per year), and mainly upward but sometimes downward[3,14,15]. Despite this knowledge, global maps of the velocities of isotherm shifts projected along the vertical dimension of elevational clines in mountain regions still do not exist. This shortfall stems partly from the complex topography and the scarcity of weather stations in most mountain ranges globally[5,16], which makes it difficult to accurately measure vertical velocities of climate change in mountain regions worldwide. Therefore, it is still an open question whether mountain species better track isotherm shifts vertically in elevation rather than horizontally in latitude.

Because we still lack global maps of the velocities at which isotherms are shifting vertically along elevation gradients as the climate warms, most local studies compute a rough estimate of this vertical projection of climate velocities by relying on a constant lapse rate of temperature (LRT). The LRT is defined here along mountain slopes as the normalized temperature difference at approximately 2 m above ground level between a low-elevation and a high-elevation weather station and thus it differs from a sensu stricto vertical lapse rate measured above a single geographical position. According to the laws of thermodynamics[6], the LRT is 9.8 °C per km in the case of dry air[1,6]. Nonetheless, given that Earth's atmosphere is not entirely dry, the LRT experienced by terrestrial organisms in reality will be less steep than 9.8 °C per km. Because of that, most studies that have compared the observed velocities of

[1]Biodiversity Research Center, Academia Sinica, Taipei, Taiwan. [2]Department of Organismic and Evolutionary Biology, Harvard University, Cambridge, MA, USA. [3]Bachelor Program in Data Science and Management, Taipei Medical University, Taipei, Taiwan. [4]Rowland Institute at Harvard University, Cambridge, MA, USA. [5]UMR CNRS 7058, Ecologie et Dynamique des Systèmes Anthropisés (EDYSAN), Université de Picardie Jules Verne, Amiens, France. [6]Department of Atmospheric Sciences, National Taiwan University, Taipei, Taiwan. [7]Department of Life Sciences, National Cheng Kung University, Tainan, Taiwan. [8]Department of Biology, Stanford University, Stanford, CA, USA. ✉e-mail: chenic@ncku.edu.tw; shensf@sinica.edu.tw

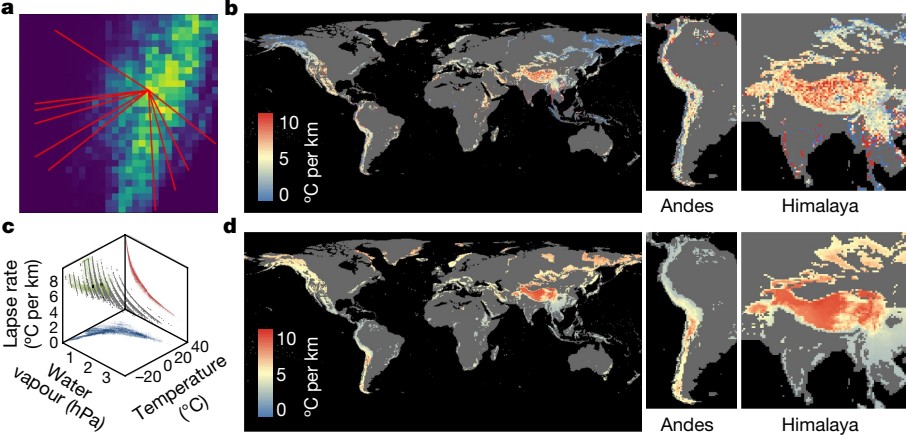

**Fig. 1 | Assessing the adiabatic LRT either through satellite observations (SLRT) or by using a mechanistic approach that accounts for water vapour (MALRT).** **a**, An example mountain range in Taiwan with a series of elevation transects, in red, defined by the highest peak at one end of the gradients and several foothills and valleys at the other end of the gradients. The background raster layer depicts the mean elevation (in m above sea level) for each spatial unit of 0.05° (around 5 km at the equator) resolution. Details can be found in the Methods and in Extended Data Fig. 1. **b**, Global map of the SLRT, generated at 0.5° (around 50 km at the equator) resolution across all mountain regions worldwide (except Antarctica) using satellite observations from 2011–2020. **c**, Three-dimensional plot showing the effect of mean annual temperature and mean annual water vapour pressure on the absolute magnitude of the MALRT (in °C per km). **d**, Global map of MALRT, generated at 50-km resolution across all mountain regions worldwide (except Antarctica) using climatic data from 2011–2020. Note that the colour scheme does not show the full range of data to prevent highly skewed visualization driven by extreme outliers.

species range shifts along elevation gradients with the velocities of climate change inside a given mountain range inferred the vertical shift of isotherms by relying on a constant rate of 5.5 °C per km for the LRT[11]—a constant that is borrowed from limited ground observations concentrated in Europe[7,17]. Using this fixed rate, one can assume that if the temperature increases by 1 °C over a given period of time, then it is expected that isotherms will move upslope by about 181.8 m during that same period, which gives a vertical velocity that varies depending only on the magnitude of temperature change per unit of time. However, the LRT is not constant and varies across elevation gradients among mountain ranges as well as within a single mountain range[18–21]. For instance, by using long-term climatology (30-year means) from 269 weather stations in northern Italy, 205 in the Tyrol area and 166 in the Trentin–upper Adige region, covering a wide range of elevations, one study[21] found that the annual mean of the LRT ranges between 5.4 and 5.8 °C per km in the Alps. In the southeastern Tibetan Plateau, the LRT was estimated[22] to reach 8.5 °C per km. This large variation in the LRT partly stems from water vapour pressure because if the air condenses moisture as it cools—for example, in cloud forests—it gains some heat from condensation, which slows the cooling rate. Thus, moisture and surface temperature generate spatial variability in the LRT and consequently also generate spatial variation in the velocities at which isotherms may shift along mountain slopes as the climate warms by a given amount of temperature increase. Assessing mountain climate velocities by explicitly considering the determinants of the LRT is a crucial step in improving our understanding of species range shifts under anthropogenic climate change. Here, instead of relying on a constant LRT value of 5.5 °C per km in the Alps or of 8.5 °C per km in the Himalayas, we propose two different methods to map the spatial variation in the LRT, and we generate more meaningful estimates of the vertical velocities of isotherm shifts in mountain systems worldwide. First, we use satellite observations of land surface temperatures at fine spatial resolution to compute a satellite-derived version of the LRT (SLRT), based on local slope estimates of the relationship between temperature and elevation (Fig. 1a and Extended Data Fig. 1); and second, we use a more mechanistic approach based on the moist adiabatic LRT (MALRT), building on the laws of thermodynamics[6] (Fig. 1c and Extended Data Fig. 2a,b). By combining information on the spatial variation of the SLRT and the MALRT at relatively fine spatial resolution

worldwide with data on the magnitude of temperature change over time per spatial unit, we then compute maps of the vertical velocities of isotherm shifts in mountain systems: one that is based on satellite observations (SLRT); and one that mechanistically accounts for water vapour pressure conditions (MALRT). These two global maps of the vertical velocities of isotherm shifts in mountain regions are also compared to a third naive map that is based on a constant LRT of 5.5 °C per km. By using these global velocity maps, we subsequently identify the mountain regions with the highest vertical velocities of isotherm shifts in the world, and we quantify the variation in velocity values along several elevation gradients worldwide. Finally, we relate those vertical velocities of isotherm shifts, in m per year, to empirical observations of species range shifts, also in m per year, along several elevation gradients in mountain systems worldwide.

We found that there was very large spatial variation when mapping the lapse rate at a global extent (Fig. 1), either from satellite observations (SLRT; Fig. 1b) or from the laws of thermodynamics (MARLT; Fig. 1d), with values ranging (at the 5th and 95th percentiles) from −5.14 to 8.45 °C per km and from 2.94 to 8.09 °C per km, respectively. Although the two global maps show a certain degree of spatial agreement (Supplementary Results), the SLRT shows much shallower lapse rates than does the MALRT in mountain regions that are located at higher latitudes, such as in northeastern Siberia, Alaska and northwestern Canada (Fig. 1b,d). The mountain regions showing the steepest lapse rates are located in the Himalayas, with values that are very consistent with the values recently reported for the southeastern Tibetan Plateau, which range between the values of free-air dry (10 °C per km) and moist (6.5 °C per km) adiabatic lapse rates[22]. For comparison purposes and external validation, we also extracted data from the Global Historical Climatology Network[23], focusing on empirical field data recorded by weather stations situated in mountain regions worldwide. We manage to obtain temperature lapse rates from 144 weather stations (station-based LRT; see Methods) across a total of 48 mountain sites from 2011 to 2019 (Extended Data Fig. 3a). This validation exercise confirms that there are very few mountain regions worldwide in which the network of weather stations is dense enough along mountain slopes ($n > 2$) to compute the LRT. Nevertheless, we found a positive relationship between the station-based LRT calculated from these very limited networks of weather-station data and our computations

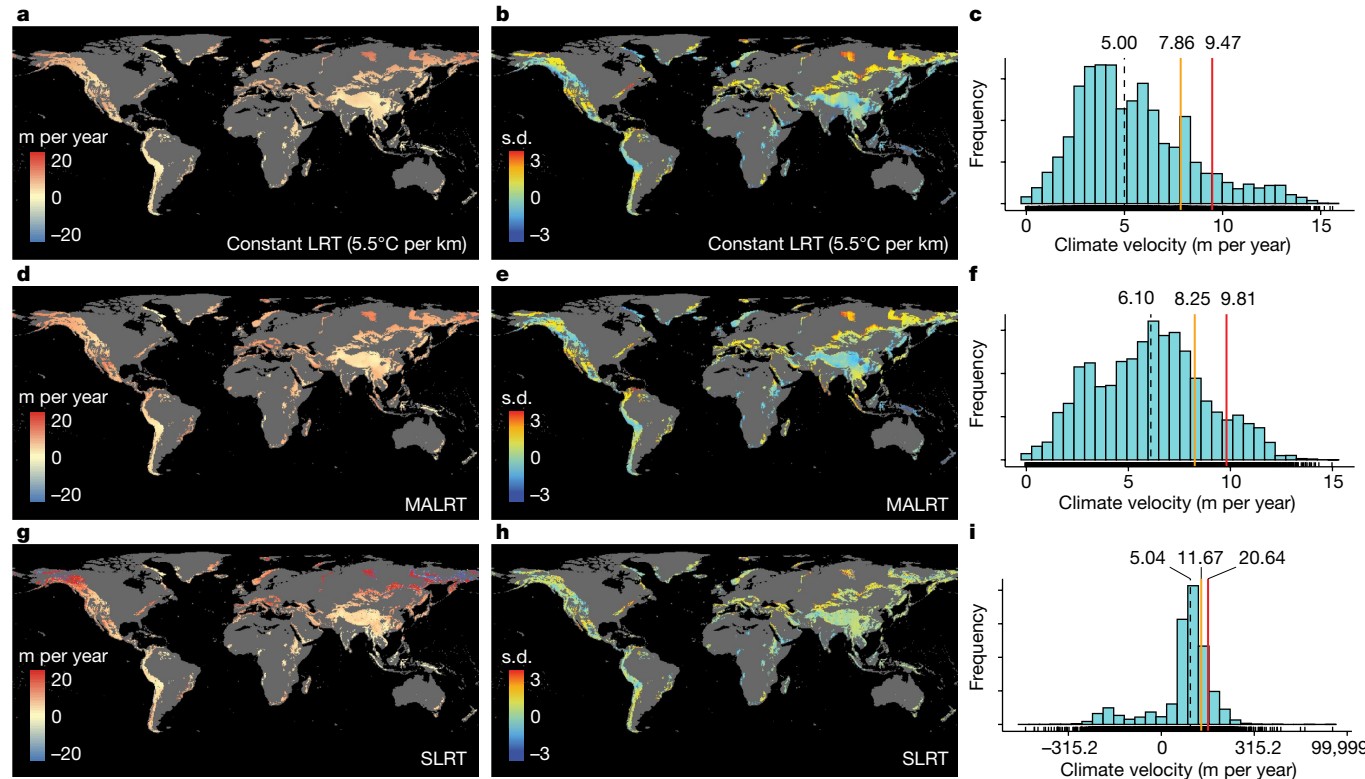

**Fig. 2 | Mapping the vertical velocities of isotherm shifts across mountain regions globally. a**–**i**, Vertical velocities of isotherm shifts (m per year) in mountain regions worldwide using a constant LRT (**a**–**c**), the MALRT (**d**–**f**) or the SLRT (**g**–**i**) (1971–1980 versus 2011–2020). **b,e,h**, Normalized value from the corresponding panel (**a,d,g**) to show clear spatial variation in each panel. **c,f,i**, Histograms of the velocity values across all mountain regions for the constant LRT, the SLRT or the MARLT, respectively. Note the $\log_{10}$ scale for the histogram displaying the range of velocity values for the SLRT. The SLRT values

were rescaled using the function $\mathrm{sign}(x) \times \log_{10}(\mathrm{abs}(x) + 1)$ to ensure that the shifting direction is preserved and to avoid interference from the value range of logarithmic transformation. Black dashed lines indicate the median; yellow solid lines show the 80% quantile; red solid lines show the 90% quantile. The corresponding values are labelled above. Note that the colour scheme does not show the full range of data to prevent highly skewed visualization driven by extreme outliers.

of the MALRT (linear regression, $F_{1,46} = 5.54$, p = 0.02, $R^2 = 0.108$, n = 48, Extended Data Fig. 3a). By contrast, the relationship between the SLRT and the station-based LRT did not reach statistical significance (linear regression, $F_{1,46} = 0.774$, $P = 0.38$, $R^2 = 0.017$, $n = 48$; Extended Data Fig. 3b). Owing to the relative scarcity of weather-station data and the fact that these data are concentrated mainly in North America and Europe, our subsequent analyses will focus solely on our computations of the MALRT and the SLRT.

After combining maps of the spatial variation in the LRT with data on the rate of temporal changes in mean annual temperature (Extended Data Fig. 2c), we found notable differences in the vertical velocities (in m per year) of isotherm shifts depending on the approach we used (Fig. 2), with the constant LRT-based and MALRT-based estimates generally yielding conservative climate velocities and the SLRT-based climate velocities showing the greatest variability. Velocity values for the SLRT-based map ranged from highly negative (−26.01 m per year; at the 5th percentile) to highly positive (34.08 m per year; 95th percentile) (Fig. 2g–i). By contrast, the MALRT-based map shows velocity values ranging (at the 5th and 95th percentile) from 1.81 m per year to 10.83 m per year. When we combined the SLRT-based velocity map with the MALRT-based velocity map to reach a consensus map on the mountain regions most threatened by climate change (Methods and Fig. 3a,b), we found that 32% of the surface area covered by mountains worldwide, Antarctica excluded, is exposed to high vertical velocities of isotherm shifts that exceed the 80th percentile by either the MALRT (80th percentile: 8.25 m per year; Fig. 3) or the SLRT (80th percentile: 11.67 m per year; Fig. 3). We delineated 17 mountain regions that are partly exposed to high vertical velocities, including those in

the Alaska–Yukon region, western America and Mexico, Appalachia, the Brazilian highlands, Greenland, Scandinavia, the Mediterranean basin, southern Africa, the Ural mountains, the Iran–Pakistan region, the Putorana mountains, Mongolia, northern Sumatra, the Kodar mountains, Yakutiya, northeast Asia and Kamchatka (Fig. 3c and Supplementary Data 1). Intuitively, higher rates of warming lead to higher vertical velocities of isotherms shifting faster along elevation gradients. This is the case chiefly in dry regions with a low water vapour pressure, such as Greenland, the Putorana Plateau in northern Siberia, Kamchatka, Mongolia and the Alaska–Yukon region—owing probably to the limited heat capacity of these arid areas[24,25] (Fig. 3d). In addition, by relying on laws of thermodynamics, we can also anticipate that regions with higher surface temperatures and/or higher water vapour pressure might also generate high vertical velocities because of shallower lapse rates: isotherms will shift faster along such elevation gradients for the same amount of temperature change over time. Notably, these regions are not necessarily those showing significant surface warming over time. For instance, northern Sumatra, the Brazilian highlands, southern Africa and Iran–Pakistan are typical representatives of such shallow lapse rates with little surface temperature increase (Fig. 3c,d). These are mountain regions threatened by high vertical velocities of isotherm shifts that have been difficult to detect in the past by surface temperature change alone, and thus are particularly worthy of further investigation.

We further compared the effects of high warming rates and steep temperature lapse rates, which act as compensatory effects on climate velocities, between arid and more humid regions. We found that in arid mountain regions with a low water vapour pressure, the temperature

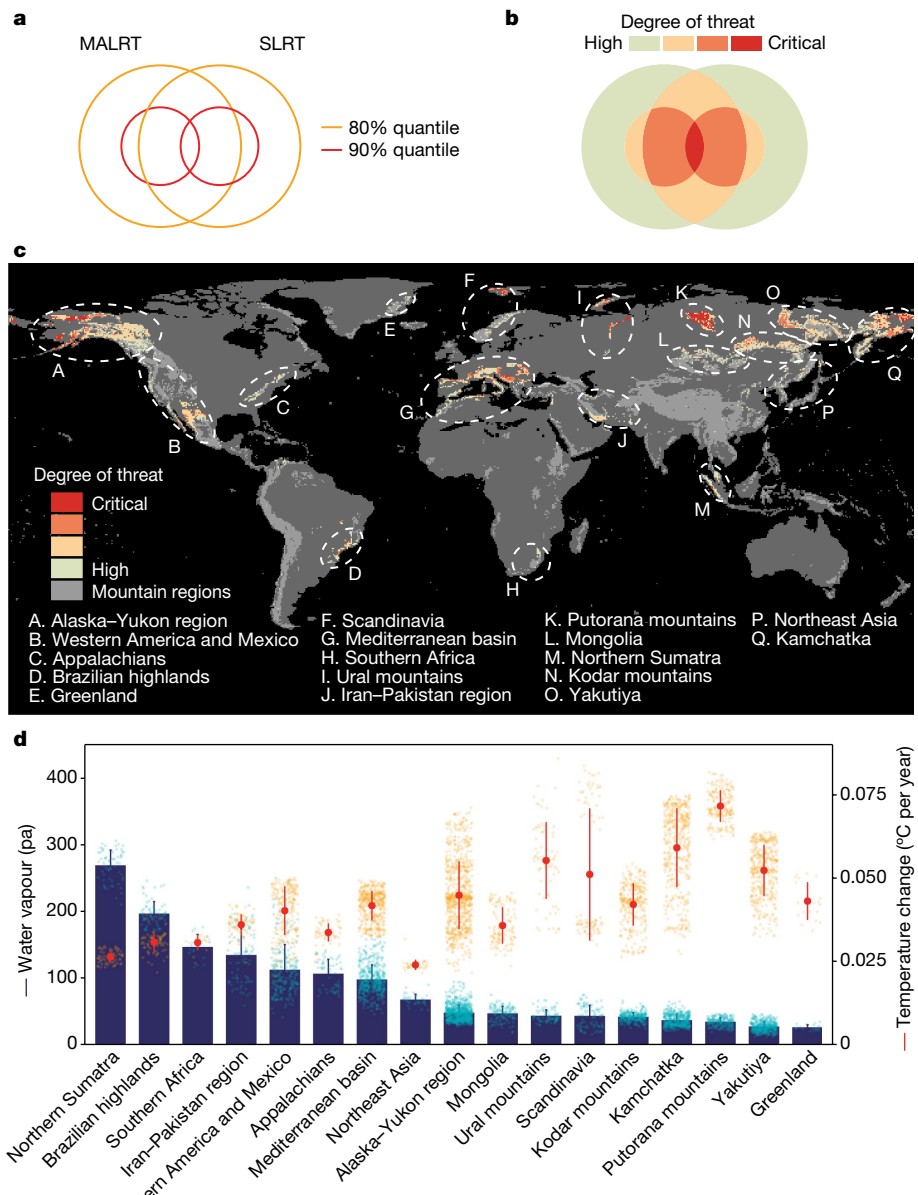

**Fig. 3 | Identifying mountain regions threatened by high vertical velocities of isotherm shifts and underlying mechanisms.** Consensus map of the vertical velocities of isotherm shifts as estimated from the SLRT or from the MALRT (see Fig. 2). **a**–**c**, Mountain regions in which velocities are greater than the 80% quantile (that is, retaining 20%) in the calculation of either the MALRT or the SLRT are labelled as critically threatened (**a**,**b**) and displayed in red (**c**). **d**, Orange points and segments represent the mean annual temperature change between the periods 1971–1980 and 2011–2020; blue bars represent the mean water vapour pressure during 2011–2020 for each of the 17 mountain regions affected by relatively fast vertical velocities of isotherm shifts. Error bars represent s.d. See Supplementary Data 1 and 'Data availability' for a comprehensive breakdown for each region, including sample size information. Considering that near-zero SLRT values result in extremely high climate velocity, we removed 1% outliers that are close to zero in **c**. Data with alternative levels of outlier removal (0.5%, 2% and 5%) are shown in Supplementary Fig. 2. Supplementary Data 3 provides a high-resolution map.

lapse rate accounts for 3.6% of the observed variation in climate velocity, whereas changes in surface temperature account for 96.4% of the observed variation, on the basis of the random forest analysis we performed. A detailed analysis using the Shapley value further revealed that steeper lapse rates have a smaller negative effect on climate velocities compared with higher warming rates, which increase climate velocities (Extended Data Fig. 4a). In humid regions, the temperature lapse rate accounts for 11.32% of the observed variation in climatic velocity, whereas changes in surface temperature explain 88.68% of the observed variation, on the basis of the random forest analysis we performed. The Shapley value analysis showed that steeper lapse rates still have a smaller negative effect on climate velocities than do higher warming rates (Extended Data Fig. 4b). Of note, the explanatory power of the lapse rate in wet mountains is nearly four times higher than it is in arid mountains. This difference is likely to be due to the lower magnitude of the surface temperature increase in wetter mountains (Extended Data Fig. 4c,d). Although the explanatory power of the lapse rate is, in general, relatively much lower than that of the warming rate, the striking differences that we found between arid and humid regions, in terms of the relative importance, affects the spatial variation that we report in the vertical velocity of isotherm shifts.

Focusing on the MALRT-based velocity map, we found a complex pattern of elevation-dependent velocities for isotherm shifts (also known as climate velocities; Fig. 4), with the highest vertical velocities

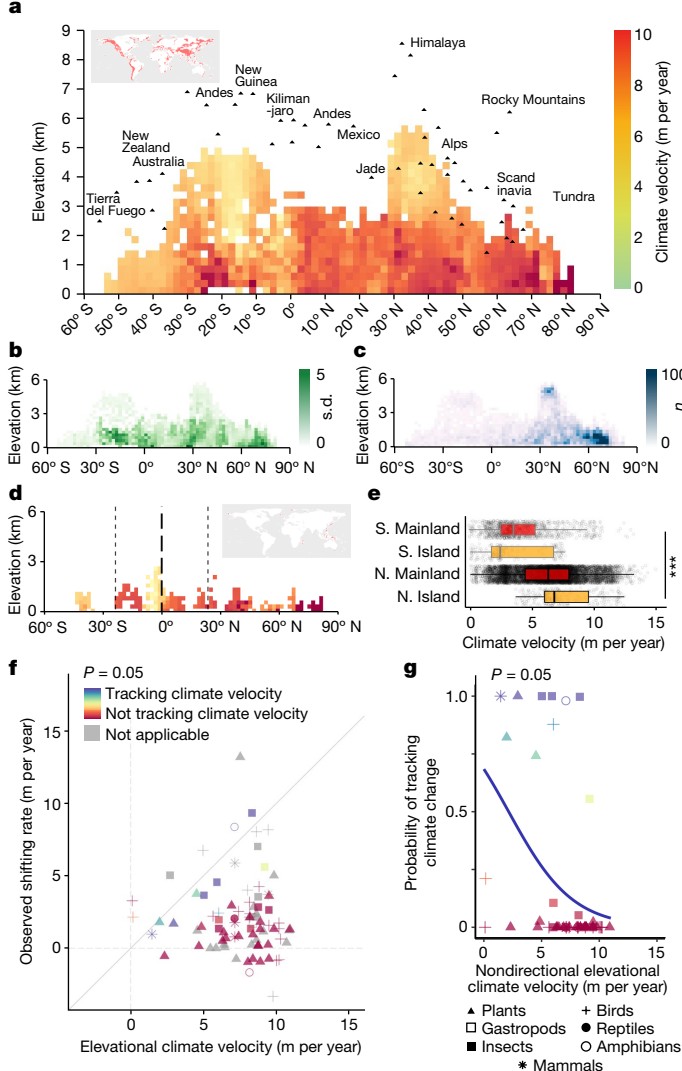

**Fig. 4 | The velocities of climate change (1971–2020) along latitude–elevation gradients and in mountain islands. a**, Mean climate velocity of mountains worldwide. Mountain summits are labelled for reference. **b,c**, The corresponding s.d. (**b**) and sample size (**c**) for **a**. **d**, Mean climate velocity of mountain islands. The s.d. and sample size for **d** can be found in Supplementary Fig. 3. The colour legend in **d** is the same as in **a**. **e**, The comparison between mainland and islands in the Northern and Southern hemispheres relies on ANOVA and post-hoc Tukey HSD tests. Other than the $P = 0.002$ between Southern Hemisphere mainland (S. Mainland) and Southern Hemisphere island (S. Island) (by Tukey HSD test), $P < 10^{-16}$ is shown in all statistics (labelled as ***). The centre line of the box plot represents the median; box limits, upper and lower quartiles; whiskers, 1.5 times the interquartile range. The sample sizes for S. Mainland, S. Island, Northern Hemisphere mainland (N. Mainland) and Northern Hemisphere island (N. Island) are 1,222, 199, 10,331 and 284, respectively. **f**, Observed species range shifts against the vertical velocities of isotherm shifts. Areas labelled as 'not applicable' (in grey) denote instances in which the number of records in a taxonomic group falls below the stipulated minimum (in this case, 30) required to conduct a meaningful statistical comparison to the predicted environmental climate velocities. **g**, The different probabilities of species tracking climate velocities under a $P = 0.05$ threshold. Only mean values are shown. Upward and downward shifts are shown together with their absolute values. For results based on different $P$ value thresholds, see Extended Data Fig. 6d,e. A total of 83 taxon–region pairs are plotted. Each plot represents 1 to more than 400 raw data points. See Extended Data Fig. 6b,c for details and Supplementary Fig. 4 for raw data points. All statistics used a two-tailed approach without adjustment for multiple comparisons.

of isotherm shifts being concentrated at low elevations. This was especially the case in the Northern Hemisphere and at a latitude of 20–30° S in the Southern Hemisphere, whereas the lowest vertical velocities were located at high elevations in the Himalayas and the Andes. Statistical results indicate that isotherm velocities are significantly higher at lower elevations (slope: −0.285 m per year·km, degrees of freedom (df) = 12,028, $t = -4.243$, $P < 0.001$) and higher absolute latitudes (slope: 0.048 m per year·deg, df = 12,028, $t = 24.163$, $P < 0.001$) in the Northern Hemisphere, whereas the magnitude of the effect significantly changed in the Southern Hemisphere ($P < 0.001$ for all interaction terms composed of elevation, latitude and hemisphere; see Methods). In the Southern Hemisphere, the elevational effect is stronger with a more negative slope estimate (slope: −1.178 m per year·km), but the latitudinal effect was completely reversed compared with the Northern Hemisphere (slope: −0.040 m per year·deg). The reversed latitudinal effect we detected here is likely to be due to the reduction of land area towards higher absolute latitudes in the Southern Hemisphere, where oceans predominate over landmasses, leading to a relatively higher water vapour pressure (Extended Data Fig. 2b) and consequently a lower temperature rate (Extended Data Fig. 2c). We further analysed the effects of changes in surface temperature and the MALRT on the rates of isotherm shift with elevation (Supplementary Fig. 1). We found no significant linear correlation between the rate of surface temperature change and elevation when the effect of latitude was statistically controlled. However, the MALRT becomes steeper with increasing elevation, leading to lower vertical velocities of isotherm shifts at higher elevations compared with lower elevations (that is, a steeper MALRT corresponds to lower vertical velocities of isotherm shifts). On islands in the Northern Hemisphere, we found higher vertical velocities of isotherm shifts (7.46 ± 2.33 m per year) exceeding, on average, the mean vertical velocity we found across all main continents in the Northern Hemisphere (6.29 ± 2.61 m per year; Fig. 4d,e; df = 3, $F = 352.9$, $P < 0.001$). These results suggest that mountain islands in the Northern Hemisphere are even more threatened by the effects of climate change than are mountains on the mainland, and this poses a high threat to island biodiversity given that mountain islands have many endemic species[26,27]. However, mountain islands in the Southern Hemisphere do not show vertical velocities of isotherm shifts that are as high as those in the Northern Hemisphere (Fig. 4e).

Next, we used our estimates of the vertical velocities of isotherm shifts in mountains and linked them to empirical data on the velocities of species range shifts along mountain slopes. We used a carefully curated dataset—BioShifts[4]—which provides the vertical velocities of species range shifts (in m per year along elevation gradients) per taxonomic unit after standardizing the raw range shift estimates reported by authors in their original studies. Because our analysis shows that the MALRT has a much greater explanatory power for predicting the velocities of species range shifts than does the SLRT (Supplementary Results and Extended Data Fig. 5), we report only on the relationship between the velocities of species range shifts along elevation gradients and the vertical velocities of isotherm shifts in mountains as calculated by the MALRT. Indeed, the Akaike information criterion (AIC) values from our models are 35,887, 37,016 and 51,398 for the MALRT, constant LRT and SLRT, respectively, ranking from best to worst in terms of model fit. This discrepancy between the MALRT and the SLRT is likely to be due to the fact that the satellite (MODIS) data measure the actual land surface temperature, which is influenced by microscale surface properties such as albedo, emissivity, rock type and vegetation cover. Hence, for the SLRT, the calculated lapse rate is characterized by considerable noise. Moreover, the SLRT data are available mainly in cloud-free conditions, which intensify these spatial variations. As a consequence, satellite data present several limitations, and thus have a limited capacity to explain species range shifts compared with insights obtained from theoretical calculations of the MALRT. Comparing the vertical velocities of isotherm shifts based on the MALRT with

the observed rates of species range shifts, the probability that a given taxonomic unit tracks the vertical velocities of isotherm movements decreases sharply with increasing absolute velocities of isotherm shifts (Fig. 4f,g). Thus, we found that species seem to track climate change only at lower velocities along the elevational gradients, irrespective of the taxonomic group (Fig. 4g, Extended Data Fig. 6d,e and Extended Data Fig. 7). These results reveal the potentially catastrophic effects of rapid climate change on mountain biodiversity. Although the MALRT will probably undergo changes over time owing to temporal variations in the spatial distribution of temperature and water vapour along elevation gradients, it is important to note that the effects resulting from a shallow MALRT are expected to be worrisome.

Our assessment of mountain climate velocity yields a mechanistic understanding of the variability in mountain climate change globally. The thermodynamic theories of the MALRT, which consider water vapour and latent heat release, suggest that threats to mountain biodiversity can occur in the absence of rapid surface warming. As our range shift analysis shows, species are unlikely to track isotherms quickly enough to match the high velocities at which isotherms are moving along some elevation gradients. Our results suggest that the vertical distance between isotherms in mountains is a crucial factor driving species migration. Likewise, on the basis of thermodynamic theory, colder and drier conditions at higher elevations make temperature lapse rates steeper, which, in turn, leads to a contraction of the vertical distance separating isotherms (that is, isotherm spacing contracts when projected on the vertical axis), generating lower vertical velocities of isotherm shifts. This suggests that in many mountain regions, the vertical shift of isotherms decreases with increasing elevation. From the perspective of isotherms shifting upslope owing to warming, higher elevations will experience a slower rate of isotherm shift, meaning that organisms can reach habitats with suitable temperatures by moving shorter vertical distances. However, a steeper temperature lapse rate also means that the environment changes more rapidly with elevation. Therefore, in the case of mountains with a broader base and narrower peaks[28], warming might result in a reduction of habitat area for organisms. Because the shape of a mountain affects the amount of habitat available to organisms[28], understanding the velocity of climate change, as well as quantifying the suitable habitat area under warming conditions, will be essential for understanding the effects of climate change on mountain biodiversity.

Moreover, our findings suggest that all taxonomic groups will be similarly affected in their abilities to track isotherms along mountain slopes. Considering that the distance of climate tracking is several orders of magnitude shorter in elevation compared with latitudinal gradients, the moving capability of organisms is less likely to be the key constraint in mountain systems. Mountainous regions, with their complex topography, occupy a relatively smaller proportion of landmasses compared with other terrains in the lowlands[28]. As described above, the available habitat area for organisms in mountain regions is influenced by the shape of the mountain, and many mountains exhibit a reduction in area with increasing elevation. This, combined with biotic interactions such as interspecific competition[29,30], might collectively limit the ability of mountain species to track isotherm shifts in the future. Mountains that we identified as facing high risks under climate change are particularly threatened by biotic attrition[17], biotic homogenization[31], population extirpation[32–34] and changing ecosystem properties[35]. Many of these mountains are located in biodiversity hotspots (for example, Sundaland, Irano-Anatolia, southern Africa, the Mediterranean basin, the Atlantic forest, Mesoamerica, the California Floristic Province and Japan)[36,37], reinforcing the need to develop climate-change adaptation strategies for the conservation of mountain biota. Other climatic drivers and mechanisms such as precipitation, snow albedo, radiation flux variability, aerosols and land-use changes can also influence energy balance regimes and further mediate mountain climates[5,38,39]. Despite many efforts to collect data on species range shifts in mountainous regions, the vast majority of data on species range shifts are still concentrated in Europe and North America[4]. This also creates uncertainty in assessing the biological effects of climate change at a global extent.

We emphasize that our results are crucial for assessing the vulnerability of mountain regions to climate change globally. By integrating surface temperature and water vapour pressure data with a thermodynamic model, we are able to make effective qualitative comparisons of global lapse rates and identify regions with comparatively higher or lower climate velocities. In particular, this approach enhances the explanatory power of our methodology over other existing methods (such as satellite data analysis) for assessing global species range shifts. However, it is important to recognize that our thermodynamic model still suffers from a low predictive accuracy when compared with field measurements of temperature lapse rates, and we cannot accurately quantify local-scale lapse rates solely on the basis of thermodynamic models. This highlights the need for refined mountain meteorological networks along elevational gradients to improve our holistic understanding of the processes that underlie local temperature lapse rates along mountain slopes. Furthermore, some studies have shown that changes in precipitation patterns can affect the range shifts of mountain species[15,40], but historical data on precipitation patterns along mountain slopes are extremely scarce compared with data on temperature lapse rates. For that reason, establishing weather stations that also monitor precipitation patterns along mountain slopes remains key for assessing the large-scale effects of precipitation changes on mountainous organisms. We call for the establishment of networks to monitor climate change and its effects in mountain biodiversity hotspots, especially in mountains that are threatened by high velocities of isotherm shifts, such as those we have identified in our study.

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

## Methods

### Approaches for mapping the LRT

Before producing global maps of the vertical velocities of isotherm shifts across mountain regions worldwide, we first had to compute global maps of the LRT. To do this, and as well as using a constant LRT for comparison purposes, we used two different approaches for mapping the LRT. On the one hand, we used a statistical or correlative approach relying on satellite observations (SLRT). On the other hand, we used a more mechanistic approach that relies on the laws of thermodynamics to account for the effect of air moisture (MARLT). Please note that all statistical tests were performed using a two-tailed approach.

### Assessing the LRT through satellite observations

In assessing the SLRT, we focused on daily land surface temperature data from the MODIS Land Surface Temperature and Emissivity (MOD11C3) product[41]. These data, encompassing the period 2011–2020 and featuring a native spatial resolution of 1 km at the equator, were averaged from both daytime and night-time observations. Monthly mean values from this product were aggregated at an annual resolution to derive the mean annual temperature, which was subsequently averaged over the 2011–2020 decade. To harmonize the spatial resolution for subsequent computations with other gridded products relying on the Climate Research Unit (CRU) Time-Series (TS) 4.05 data, the MODIS data were aggregated, using the mean value, from their native spatial resolution to a 0.05° resolution (Extended Data Table 1), which is approximately 5 km at the equator, ensuring that there were ample grid cells for subsequent analyses. Using a moving window centred on a grid cell of 0.5° resolution, which is about 50 km at the equator, elevational transects were derived to empirically compute the LRT from satellite observations. This involved pinpointing regional peaks and foothills in a 1.5° by 1.5° window centred on the target grid cell of 0.5° resolution, with elevation data sourced from a digital elevation model (DEM) that was aggregated to match the 0.05° resolution of the aggregated MODIS grid (Extended Data Fig. 1a). From these peaks and foothills, elevational transects connecting the nearest topographical features were established (Extended Data Fig. 1b,c). Linear regressions between mean annual temperature and elevation, both at the 0.05° resolution, were subsequently fitted for each transect intersecting the target 0.5° grid cell (Extended Data Fig. 1d–f). All pixel units intersected by a focal transect were considered, even if only marginally. Transects yielding significant lapse rates ($R^2 \geq 0.5$ and $P \leq 0.05$) were retained, with the slope coefficient ($\beta$) representing the SLRT value in °C per m (later converted to °C per km). If more than ten transects intersected a target 0.5° grid cell, the median SLRT value was calculated to mitigate biases from transect count extremities. Within the framework of our SLRT computations, the median transect count per grid cell was 8, with an interquartile range of 12 (Extended Data Fig. 8a,b). We noticed that a higher transect availability in a grid cell was correlated with increased average $R^2$ values between temperature and elevation ($R^2 = 0.16$, $P < 0.001$; Extended Data Fig. 8c), underscoring the dependency of the reliability of the SLRT on the number of accessible transects.

### Assessing the LRT from first principles

To compute the MALRT, we extracted monthly mean temperature and monthly mean water vapour pressure data from the gridded CRU TS4.05 database (at 0.5° spatial resolution), covering the decade 2011–2020 to match the time period covered by satellite observations (see 'Assessing the LRT through satellite observations'). In the CRU TS4.05 dataset, both monthly mean temperature and monthly mean water vapour pressure were derived from local weather stations and processed to obtain the final values[42,43]. The MALRT of each grid cell was computed using the following formula:

$$\Gamma_w = g \frac{1 + \frac{H_v \gamma}{R_{sd} T}}{C_{pd} + \frac{H_v^2 \gamma \epsilon}{R_{sd} T^2}}$$

where $\Gamma_w$ is the moist adiabatic lapse rate in Kelvin per metre, $g$ denotes Earth's gravitational acceleration (9.8076 m per s[2]), $H_v$ denotes the heat of vaporization of water (2,501,000 J kg[-1]), $R_{sd}$ denotes the specific gas constant of dry air (287 J kg[-1] K[-1]), $\epsilon$ denotes the dimensionless ratio of the specific gas constant of dry air to the specific gas constant for water vapour (0.622), $C_{pd}$ denotes the specific heat of dry air at constant pressure (1,005 J kg[-1] K[-1]) and $T$ denotes the air temperature (K). The parameter $\gamma$ is the mixing ratio of the mass of water vapour to the mass of dry air:

$$\gamma = \epsilon e/(p - e)$$

where $e$ represents the water vapour pressure of the air and $p$ represents the pressure of the air. Here, $p$ was derived from the barometric formula (see Supplementary Methods).

The processing of climatic variables (from monthly data to annual data) was done using Python v.3.7.9. Note that the original MARLT values, expressed in Kelvin per metre, were subsequently transformed into °C per kilometre for comparative purposes with the SLRT and the constant LRT. The increase in mean annual surface temperature and mean annual water vapour pressure both cause a decrease in the MALRT (see Fig. 1c).

For comparison purposes, the same approach was also applied to the datasets available from the 'Climatologies at high resolution for Earth's land surface areas' data (CHELSA v2.1)[44] after the datasets were aggregated from the native spatial resolution at 1 km to 0.5° spatial resolution, using the mean value. Note, however, that the data are available only for the period 2011–2019 and do not entirely cover the 2011–2020 decade. Information on water vapour is not available in CHELSA, so water vapour was derived by multiplying relative humidity and the saturated water vapour obtained by applying the Clausius–Clapeyron equation[45]. This derived MALRT using CHELSA data shows high consistency with that derived from the CRU dataset, with the strength of the correlation varying slightly depending on the elevation band considered (ranging from 0.79 to 0.96, $P < 0.001$; Extended Data Fig. 9).

### Computing the vertical velocities of isotherm shifts

To assess the vertical projection of the velocities at which isotherms are moving along elevation gradients in mountain regions as the climate is warming globally, we combined information on the spatial variation, at 0.5° spatial resolution, of the LRT, assessed through either the SLRT or the MALRT method, with data on the rate of temperature change over time per spatial unit. For computing the temporal rate of temperature change per spatial unit of 0.5° resolution, we used mean annual temperature time series from the gridded CRU TS4.05 dataset covering the period 1971–2020. More specifically, for each spatial unit of 0.5°, we first averaged the mean annual temperature for the periods 1971–1980 versus 2011–2020 before computing the difference between the two and dividing this difference by the time duration in years (40 years), so that the magnitude of temperature change was expressed in °C per year. The gridded layer of temporal changes in mean annual temperature between 1971–1980 and 2011–2020 was subsequently divided by the gridded layer of either the SLRT or the MARLT, expressed in °C per km, such that the vertical projection of velocity values on a map is expressed in km per year. For further comparison with the velocities of species range shifts, usually reported in m per year, we multiplied the vertical velocity map by 1,000 so that the unit is in m per year. Finally, we also generated a map of the vertical velocities of isotherm shifts in mountain systems using a constant LRT of 5.5 °C per km to be used as a control for what is usually done in the scientific literature to compute the vertical velocities of isotherm shifts in mountains[11,46,47].

## Comparing and validating LRTs against station-based measures

To validate our maps of the SLRT and MALRT, we used an external dataset of the LRT along elevation gradients by relying on field observations from local weather stations. We extracted time series of monthly temperature data from several weather stations belonging to the Global Historical Climatology Network that extend to 2019 (ref. 23). First, we selected weather stations covering the period 2011–2019: (1) when more than eight years of data were available; and (2) only if more than 10 months were recorded per year. Then, to match our gridded LRT values with station-based LRT values, we selected only the weather stations that are located within or in the vicinity of each grid cell belonging to a given mountain region. In particular, we collected data from the weather stations located within the central grid itself along with weather stations located within the eight adjacent grid cells, forming a nine-cell cluster, which we term a 'mountain site', within a mountain region, for ease of reference. Mountain sites that included at least three weather stations at different elevations were used for the computation of the station-based LRT. After excluding two extreme outliers from the set of station-based LRT values we computed, we ran two separate linear models (with two-tailed statistical tests) to assess the relationship between station-based LRT values (the response variables) and either MALRT or SLRT values as separate explanatory variables.

## Identifying the mountain regions that are most threatened by climate change

We can use the climate velocities calculated above, which carefully consider the spatial heterogeneity that affects the LRT, to determine which mountains around the globe are threatened by the highest velocities of isotherm shifts as a surrogate of the vulnerability risk for mountain biota as climate warms. We simultaneously considered both the MALRT- and the SLRT-based approaches (Fig. 2d–i) to accommodate the heterogeneity of climatic conditions that is inherent to the complex topography and sparse instrumental data available in mountain regions. We defined high-risk mountain areas as those with velocity values of isotherm shifts exceeding the 80th percentile calculated by either method (Fig. 2f,i). The threat level was then defined by the intersection or union of the highest 20% or 10% velocities of isotherm shifts of either method (Fig. 3a,b). Given that SLRT values close to 0 will provoke extremely high climate velocities, we removed 1% of outliers that were close to zero when we plotted Fig. 3c. Other levels of outlier removal (0.5%, 2% and 5%) can be found in Supplementary Fig. 2.

## Analysing the distribution of vertical velocities in the elevation–latitude plane

In addition to mapping the spatial distribution of the vertical projection of the velocities at which isotherms are shifting along mountain slopes worldwide and to better understand how velocity values distribute along elevation gradients at a global extent, we investigated the distribution of vertical velocity values across the bidimensional space of the elevation–latitude plane. Because the exposure to climate warming is greater at higher elevations[5,16] and towards higher latitudes in the Northern Hemisphere[48], we expect a non-random distribution of vertical velocity values in the elevation–latitude plane. Because the MALRT mechanistically incorporates the effects of surface temperature and water vapour pressure on the vertical velocities of isotherm shifts, and the biological analyses also suggest the importance of the MALRT over the SLRT in explaining the observed variation in the velocities of species range shifts (see Supplementary Information), we decided to focus solely on the MALRT-based velocity map to analyse the distribution of velocity values in the elevation–latitude plane. To do that, we reorganized all 12,036 spatial units from the MARLT-based velocity map at 0.5° resolution into a raster image with pixel units of 250-m resolution along the elevation axis and 2° resolution along the latitude axis. For each cell of the elevation–latitude plane, we computed and

plotted the mean vertical velocity as well as the standard deviation and the sample size.

In the case of mountain islands, we repeated the above analysis for the elevation–latitude plane representation but relied on spatial data at finer resolution. Islands are defined as landmasses smaller than Australia and surrounded by water[49]. In this study, the DEM that we used is derived from the Shuttle Radar Topography Mission (SRTM)[50] rather than from the CRU's DEM. The SRTM[50], boasting a finer spatial resolution of 30 m, offers superior suitability for island detection, particularly for insular landforms proximate to the coast that remain unconnected to the mainland. Greenland is not included because it is not surrounded by the ocean in the dataset. These analyses were run in Wolfram Mathematica v.12 (ref. 51). The comparison between mainland and island velocities of isotherm shifts was done separately for the Northern and Southern hemisphere by the mean of a one-way ANOVA with post-hoc Tukey HSD test[52].

To test whether the vertical velocities of isotherm shifts are greater at higher elevations in general and greater towards higher latitudes in the Northern Hemisphere, we ran a multivariate least square regression with elevation, absolute latitude, hemisphere (a factor variable with two levels: Northern versus Southern), the two-way interaction terms between all possible combinations of two of the three independent variables as explanatory variables explaining the mean vertical velocity of isotherm shifts, and also the three-way interaction terms (elevation, absolute latitude and hemisphere). This analysis was done on the basis of the original raster map (longitude–latitude) before summarizing into latitude–elevation dimensions.

## Probability of species tracking isotherms: comparing biological and climate velocities

We used the BioShifts database[4] which provides quantitative data on the velocities of species range shifts (in m per year along the elevation gradient). To assess how the vertical velocities of isotherm shifts, after incorporating the spatial variation in the MALRT, relate to the observed velocities of species range shifts along elevation gradients, we first extracted empirical observations of species range shifts along the elevation gradients of mountain regions as delineated by original studies, thus excluding latitudinal range shifts. Then, we extracted the vertical velocity values for isotherms at the centroid of a given mountain region for which we could retrieve elevational range shift data from BioShifts (https://doi.org/10.6084/m9.figshare.7413365.v1). To avoid substantial spatial variation from studies conducted on a larger spatial extent, such as those spanning national or continental areas, we specifically chose datasets covering a spatial extent that approximates the resolution of our environmental dataset (0.5°). Hence, we focused on spatial features or polygons (that is, the spatial delineation of the study areas) smaller than approximately 100 km × 100 km (1° × 1°) to ensure that the environmental variables at the centroids of these polygons were less susceptible to spatial variation. A total of 5,452 datasets were retained for our subsequent analyses. To achieve this, we superimposed the centroid of the spatial polygons or shapefiles, as provided in the BioShifts database, of each of the selected study areas associated with elevational range shift data onto the MALRT-based velocity map. Here, we decided to focus solely on the MARLT-based map of the vertical velocities of isotherm shifts, because the MALRT is better correlated to the velocities of species range shifts than the SLRT is (see Supplementary Information).

Then, we computed the likelihood that a specific species from a designated taxonomic group (plants, birds, mammals, gastropods, insects, amphibians or reptiles; details provided in the Supplementary Information) tracks the vertical velocities of isotherm shifts within a particular mountainous area. To achieve this, we randomly resampled a fixed number of elevational range shift observations for each taxonomic group in each mountain region. This ensured relatively consistent and balanced sample sizes across all of the examined mountain regions

and taxonomic groups. More specifically, for each taxonomic group in each mountain region (that is, the source region provided in the original dataset[4] and available as shapefiles (.shp files) in the BioShifts database), we set the maximum sample size to $n$ (see below for a sensitivity analysis on the effect of $n$) and resampled $n$ records if the number of records was greater than $n$ (see Extended Data Fig. 6a). If the total number of records for a given taxonomic group in a given mountain region was less than $n$, all records were used. The randomly sampled data on the observed velocities of range shifts were then compared to the corresponding set of vertical velocity values as obtained from the MALRT-based velocity map for that focal mountain region. To test for statistical differences between the two, we used a nonparametric method−the bilateral Wilcoxon signed rank test. This procedure (plotting and statistical comparison using a Wilcoxon signed rank test) was then iterated 1,000 times (see Extended Data Fig. 6a) and we calculated the number of iterations in which the empirical velocities of species range shifts did not differ significantly from the corresponding vertical velocities of isotherm shifts (that is, did not reach the significance level of $P < 0.05$; see Extended Data Fig. 6) and divided it by the total number of iterations (1,000). The obtained proportion value, ranging between 0 and 1, gives the probability that a given focal taxonomic group has more or less tracked the vertical velocity of isotherm shifts in the focal mountain region as the climate warms globally. A logistic-type (probit) function was then applied to estimate the probability curve. We also performed a sensitivity analysis by setting different maximum sample sizes for $n$ (10, 20, 30, 40, 50, 60, 70, 80, 90 and 100), and the results became stable when $n$ was larger than 30 (Supplementary Data 2), so we decided to set $n = 30$ to address the problem of studies with a small sample size. The data processing and statistical analysis in this section were done in R v.4.04 (ref. 53).

## Reporting summary

Further information on research design is available in the Nature Portfolio Reporting Summary linked to this article.

## Data availability

The data supporting the findings of this study are available in the paper and at https://doi.org/10.5061/dryad.1rn8pk0wm. CRU TS4.05 is available at https://crudata.uea.ac.uk/cru/data/hrg/; MOD11C1 at https://lpdaac.usgs.gov/#nav-heading; MOD11C2 at https://lpdaac.usgs.gov/#nav-heading; MOD11C3 at https://lpdaac.usgs.gov/#nav-heading; EarthEnv at https://www.earthenv.org/; ETOPO1 at https://www.ncei.noaa.gov/products/etopo-global-relief-model; SRTM at https://www.earthdata.nasa.gov/sensors/srtm; GMBA at https://www.gmba.unibe.ch/services/tools/mountain_inventory_v1; CHELSA at https://chelsa-climate.org/; GHCN at https://www.drought.gov/data-maps-tools/global-historical-climatology-network-ghcn; and BioShifts at https://doi.org/10.6084/m9.figshare.7413365.v1. Source data are provided with this paper.

## Code availability

Code is available at https://doi.org/10.5061/dryad.1rn8pk0wm.

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

**Acknowledgements** We thank Y.-S. Jang, T.-C. Hsieh, S.-H. Wang and C.-Y. Lin for their help in the early development of this study; J.-H. Chen for providing statistical suggestions at the early stages of this study; and C. D. Thomas, R. K. Colwell, R. R. Childers, S. Ashe and C.-N. Chou for their comments on the early version of the manuscript. We acknowledge grants 108-2314-B-001-009-MY3 (S.-F.S.) and 104-2311-B-006-006-MY3 (I.-C.C.) from the Ministry of Science and Technology, Taiwan, and grants AS-SS-106-05 (S.-F.S.) and AS-SS-110-05 (S.-F.S.) from the Academia Sinica.

**Author contributions** Conceptualization: S.-F.S. Methodology: S.-F.S., I.-C.C. and W.-P.C. Formal analysis: W.-P.C. Random forest analysis: G.-S.M. Visualization: W.-P.C. Writing (original draft): S.-F.S., I.-C.C. and W.-P.C. Writing (review and editing): S.-F.S., I.-C.C., J.L., W.-P.C., G.-S.M. and H.-C.K.

**Competing interests** The authors declare no competing interests.

**Additional information**
**Correspondence and requests for materials** should be addressed to I-Ching Chen or Sheng-Feng Shen.

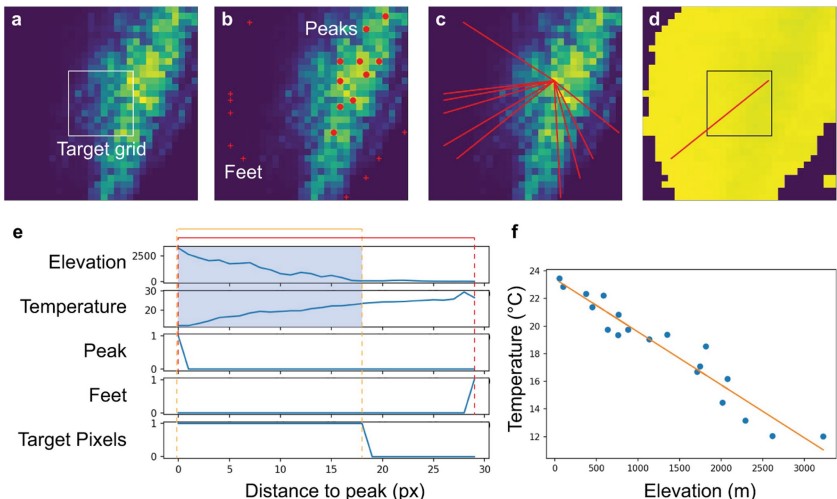

**Extended Data Fig. 1 | Diagram illustrating the calculation of SLRT. a**, 10 × 10 grids (all at 0.05-degree spatial resolution) of the DEM were included in a target grid (0.5-degree spatial resolution). **b**, Mountain peaks and feet/valleys are automatically searched and identified. **c**, A transect can be defined by a peak-foot/valley pair. **d**, An exemplar transect on the land surface temperature map (at 0.05-degree spatial resolution). **e**, The elevation and the mean annual temperature data across 2011–2020 along the exemplar transect are specified. **f**, The relationship between mean annual temperature and elevation. The data points were extracted according to **e**. The regression line is provided as the orange solid line, where the slope (beta in the regression model) is considered as the lapse rate of a transect.

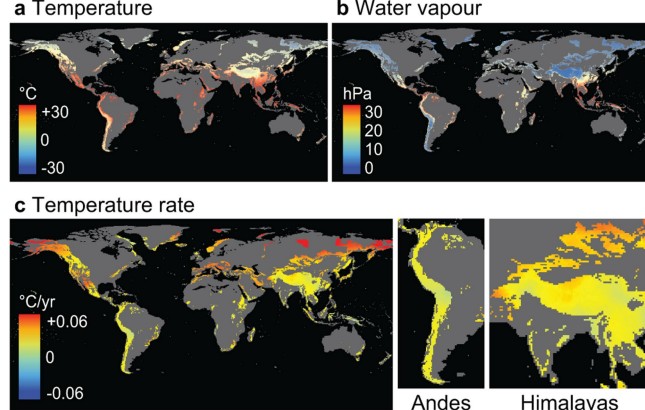

**Extended Data Fig. 2 | Map of temperature (2011–2020), water vapour (2011–2020) and rate of temperature change over time (1971–1980 versus 2011–2020) in global mountains. a**, Averaged mean annual temperature (2011–2020). **b**, Averaged mean annual water vapour (2011–2020). **c**, Temperature differences between the two periods divided by the temporal period (40 years). Note that the colour scheme does not show the full range of data to prevent highly skewed visualization driven by extreme outliers.

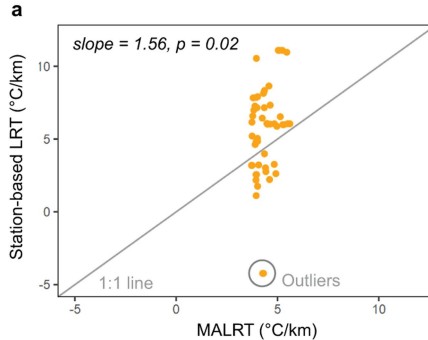 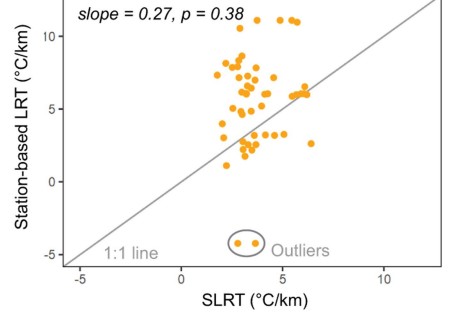

**Extended Data Fig. 3 | Comparison between MALRT, SLRT and weather-station-based LRT. a,b**, Scatterplots show MALRT values against station-based LRT values (**a**) and SLRT values against station-based LRT values (**b**) for available mountain sites (*n* = 48). Two conspicuous outliers (circled in grey) derived from the weather-station data were excluded. The regressive slopes of the plots are labelled. The statistics were done using a two-tailed approach without adjustment for multiple comparisons.

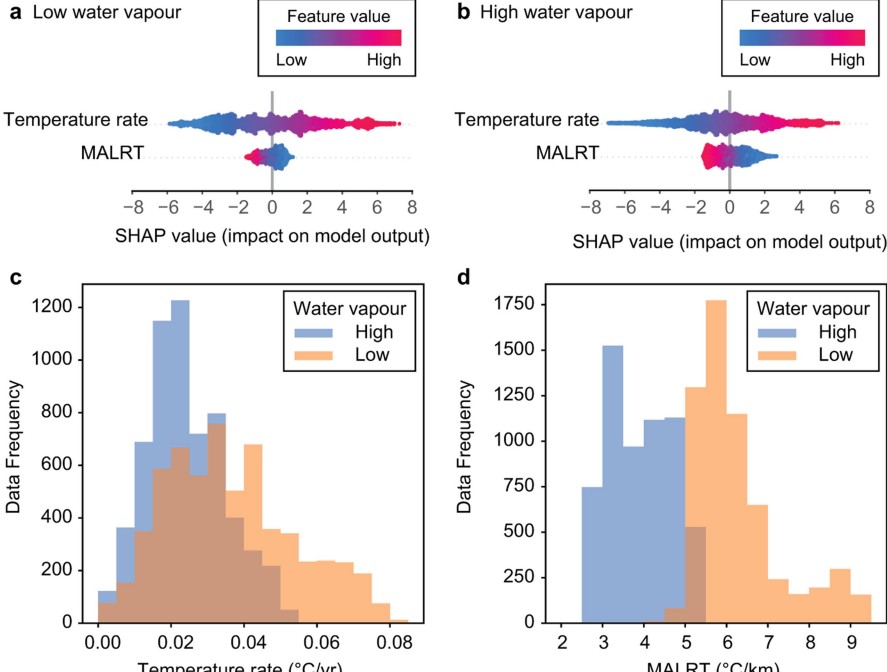

**Extended Data Fig. 4 | Comparative analysis of temperature rate and MALRT in relation to water vapour. a,b**, SHAP (SHapley Additive exPlanations) value distributions that elucidate the model's decision-making process under low (**a**) and high (**b**) water vapour conditions. In these sub-figures, each point signifies a prediction made by the model. They are coloured according to the feature's (temperature rate and MALRT) value, creating a spectrum that indicates the feature's effect; warmer colours symbolize higher values and cooler colours represent lower values. The *x* axis demonstrates the SHAP values, portraying the magnitude and direction of a feature's effect on the model's output, with negative values suggesting a decrease and positive values indicating an increase in the prediction. **c,d**, Histograms of temperature rate (**c**) and MALRT (**d**) under conditions of high (blue) and low (orange) water vapour. The *x* axis corresponds to temperature rate and MALRT, measured in °C per year; the *y* axis represents the frequency of occurrence.

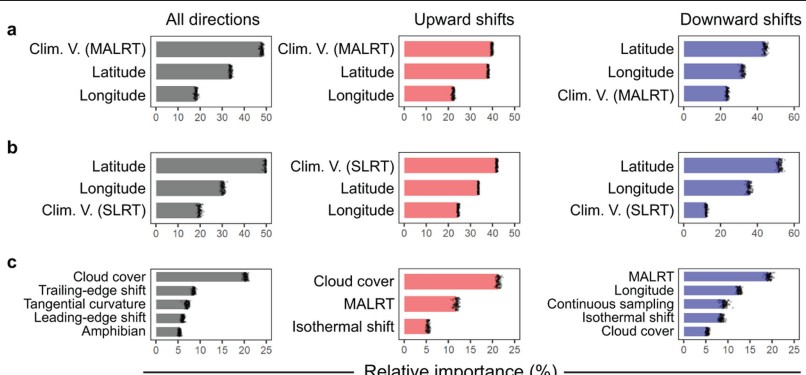

**Extended Data Fig. 5 | Relative importance of variables in explaining the velocities of species range shifts on the basis of a random forest model.** Different columns indicate different datasets. **a**,**b**, Models include the basic geographical factors (latitude and longitude) as well as the vertical velocities of isotherm shifts (Clim. V.) derived from either the MALRT (**a**) or the SLRT (**b**). **c**, Models consider all possible factors influencing the velocities of species range shifts (Supplementary Methods). The centre and the error bars indicate mean and s.d., respectively. Sample sizes for datasets filtered for upward shifts, downward shifts and all directions are 3,635, 1,401 and 5,452, respectively.

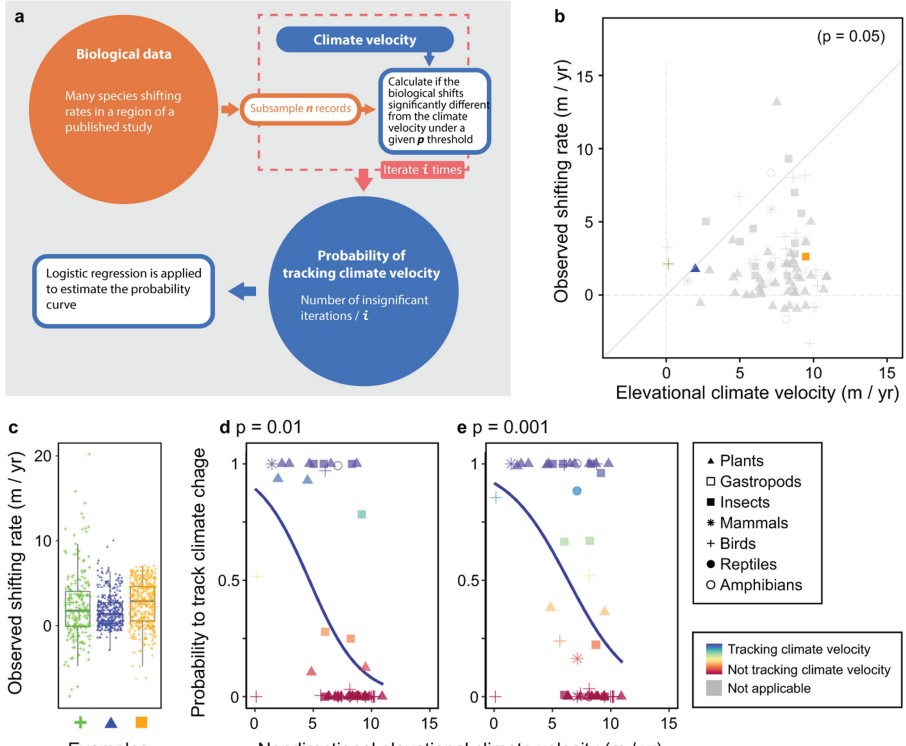

**Extended Data Fig. 6 | Probability of tracking the vertical velocities of isotherm shifts (climate velocities) for mountain species. a**, Diagram summarizing how the probability of tracking climate velocities was calculated ($i = 1,000$). **b**, Replicate of Fig. 4f with colour-labelled exemplar taxonomic groups. The raw values are shown in **c**. The centre line of the box plot represents median; box limits, upper and lower quartiles; whiskers, 1.5 times the interquartile range. The sample size for three examples are 219 (green), 372 (blue) and 433 (orange). **d,e**, The different probabilities of species tracking climate velocities under different $P$ thresholds ($P = 0.01$ (**d**) and $P = 0.001$ (**e**)). A total of 83 taxon–region pairs are plotted. Each plot represents 1 to more than 400 raw data points. Only mean values are shown. Upward and downward shifts are shown together with their absolute values. For raw data points, see Supplementary Fig. 4. The statistics were done using a two-tailed approach without adjustment for multiple comparisons.

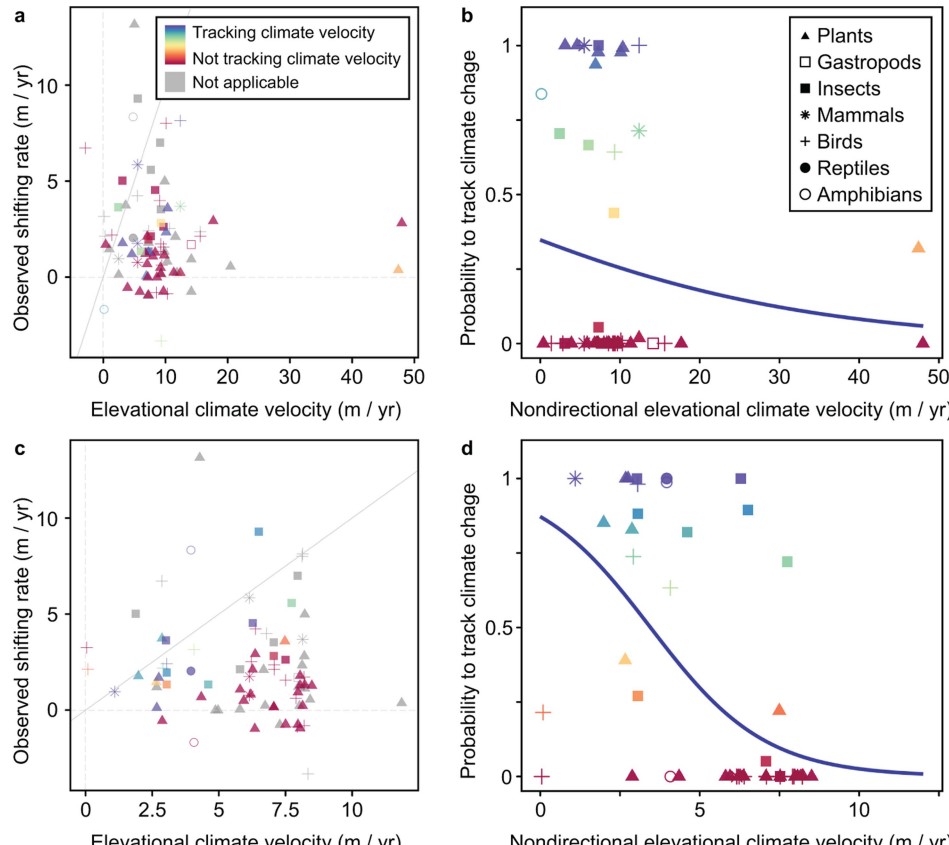

**Extended Data Fig. 7 | Maximum trackable climate velocities based on SLRT and constant lapse rate. a,b**, Velocities based on SLRT. **c,d**, Velocities based on constant lapse rate (5.5 °C per km). The relationships between observed shifting rate and elevational climate velocities are shown in **a,c**. Only mean values are shown. The probabilities that species may track climate velocity are shown in **b,d**.

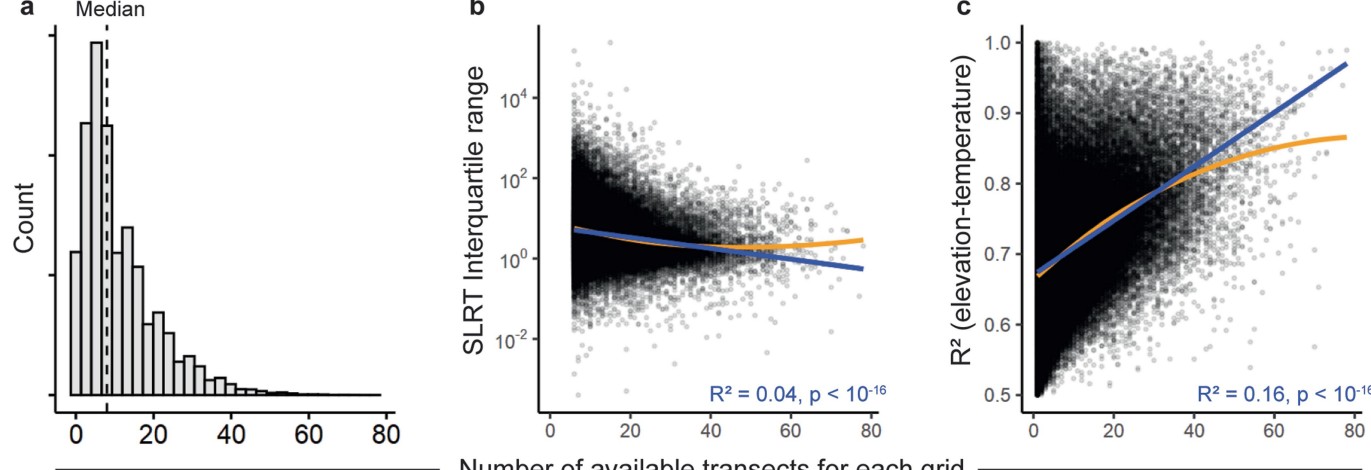

**Extended Data Fig. 8 | Influence of the number of available transects on SLRT results. a**, Distribution of the number of available transects. **b**, Correlation between the number of available transects and the SLRT interquartile range. **c**, Correlation between the number of available transects and the averaged $R^2$ between elevation and temperature. Blue lines indicate simple regression between the two variables, with statistics labelled at the bottom right of each panel. Orange lines represent LOESS (locally estimated scatter plot smoothing) lines. The statistics were done using a two-tailed approach without adjustment for multiple comparisons.

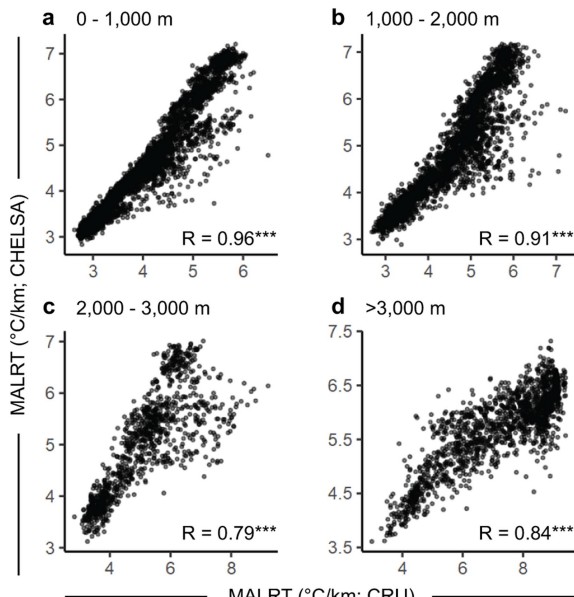

**Extended Data Fig. 9 | Comparison between CHELSA and CRU data in global mountain regions (2011–2020). a–d**, Comparisons of MALRT at different elevational ranges (0–1,000 m (**a**); 1,000–2,000 m (**b**); 2,000–3,000 m (**c**); and more than 3,000 m (**d**)). Statistics are labelled at the bottom right of each panel. Significance levels are indicated: ***$P < 10^{-16}$. The statistics were done using a two-tailed approach without adjustment for multiple comparisons.

**Extended Data Table 1 | Climatic and environmental variables used in this study**

| Variables | Original resolution[a] | Operational resolution used in our analysis[b] | Temporal span | Data source |
|---|---|---|---|---|
| Temperature (ambient) | 0.5° (~50km) | 0.5° (~50km) | 1971-1980 | CRU TS4.05[43] |
| | | | 2011-2020 | |
| Water vapour | 0.5° (~50km) | 0.5° (~50km) | 2011-2020 | CRU TS4.05[43] |
| Precipitation | 0.5° (~50km) | 0.5° (~50km) | 2011-2020 | CRU TS4.05[43] |
| Cloud cover | 0.5° (~50km) | 0.5° (~50km) | 2011-2020 | CRU TS4.05[43] |
| Temperature (land surface for SLRT) | 1 km | 0.05° (~5km) | 2011-2020 | MOD11C3[41] |
| Aspect (sine) | 1 km | 0.5° (~50km) | NA | EarthEnv[54] |
| Aspect (cosine) | 1 km | 0.5° (~50km) | NA | EarthEnv[54] |
| Slope | 1 km | 0.5° (~50km) | NA | EarthEnv[54] |
| Profile curvature | 1 km | 0.5° (~50km) | NA | EarthEnv[54] |
| Tangential curvature | 1 km | 0.5° (~50km) | NA | EarthEnv[54] |
| Terrain evenness | 1 km | 0.5° (~50km) | NA | EarthEnv[54] |
| Terrain homogeneity | 1 km | 0.5° (~50km) | NA | EarthEnv[54] |
| Terrain roughness | 1 km | 0.5° (~50km) | NA | EarthEnv[54] |
| Vector Ruggedness Measure | 1 km | 0.5° (~50km) | NA | EarthEnv[54] |
| Elevation (for MALRT) | 0.5° (~50km) | 0.5° (~50km) | NA | CRU[c, 43] |
| Elevation (for SLRT) | 1 min (~1.8km) | 0.05° (~5km) | NA | ETOPO1[55] |
| Elevation (for island identification) | 1 sec (30m) | 0.5° (~50km) | NA | SRTM[50] |
| Enhanced vegetation index | 1 km | 0.5° (~50km) | 2011-2020 | MOD13C2[56] |
| Land use/land type | 500 m | 0.05° (~5km) | 2011-2019 | MCD12C1[57] |
| Ecofacet | 250 m | 0.5° (~50km) | 1950-2000 | [58] |
| Mountain regions | Polygon | 0.5° (~50km) | NA | GMBA V1.2[59] |
| Biogeographical realms | Polygon | 0.5° (~50km) | NA | [60] |
| Global mountain cloud forests | 1 km | 0.5° (~50km) | 1971-2009 | EarthEnv[54] |
| Temperature (CHELSA) | 30 sec (~1km) | 0.5° (~50km) | 2011-2019 | CHELSA[44] |
| Relative humidity (CHELSA) | 30 sec (~1km) | 0.5° (~50km) | 2011-2019 | CHELSA[44] |
| Temperature (weather station) | NA | NA | 2011-2019 | GHCN[23] |

The spatial resolution and temporal span are also shown. [a]A data provider may provide a dataset with multiple resolutions, which were calculated from the original one with the finest resolution. [b]The resolution of each data layer may change in different steps of our calculation, so the resolution used in the main calculation is provided. [c]No specific version is provided. See refs. 54–60.

# Reporting Summary

## Statistics

For all statistical analyses, confirm that the following items are present in the figure legend, table legend, main text, or Methods section.

| n/a | Confirmed | |
|---|---|---|
| ☒ | ☐ | The exact sample size ($n$) for each experimental group/condition, given as a discrete number and unit of measurement |
| ☒ | ☐ | A statement on whether measurements were taken from distinct samples or whether the same sample was measured repeatedly |
| ☐ | ☒ | The statistical test(s) used AND whether they are one- or two-sided<br>*Only common tests should be described solely by name; describe more complex techniques in the Methods section.* |
| ☒ | ☐ | A description of all covariates tested |
| ☒ | ☐ | A description of any assumptions or corrections, such as tests of normality and adjustment for multiple comparisons |
| ☐ | ☒ | A full description of the statistical parameters including central tendency (e.g. means) or other basic estimates (e.g. regression coefficient) AND variation (e.g. standard deviation) or associated estimates of uncertainty (e.g. confidence intervals) |
| ☐ | ☒ | For null hypothesis testing, the test statistic (e.g. $F$, $t$, $r$) with confidence intervals, effect sizes, degrees of freedom and $P$ value noted<br>*Give P values as exact values whenever suitable.* |
| ☒ | ☐ | For Bayesian analysis, information on the choice of priors and Markov chain Monte Carlo settings |
| ☐ | ☒ | For hierarchical and complex designs, identification of the appropriate level for tests and full reporting of outcomes |
| ☒ | ☐ | Estimates of effect sizes (e.g. Cohen's $d$, Pearson's $r$), indicating how they were calculated |

*Our web collection on statistics for biologists contains articles on many of the points above.*

## Software and code

Policy information about availability of computer code

| Data collection | R4.0.4; Python 3.7.9 |
|---|---|
| Data analysis | R4.0.4; Python 3.7.9; Wolfram Mathematica 12; sklearn v022.2.post1; shap v0.41.0 |

For manuscripts utilizing custom algorithms or software that are central to the research but not yet described in published literature, software must be made available to editors and reviewers. We strongly encourage code deposition in a community repository (e.g. GitHub). See the Nature Portfolio guidelines for submitting code & software for further information.

## Data

Policy information about availability of data

All manuscripts must include a data availability statement. This statement should provide the following information, where applicable:
- Accession codes, unique identifiers, or web links for publicly available datasets
- A description of any restrictions on data availability
- For clinical datasets or third party data, please ensure that the statement adheres to our policy

The authors declare that the data supporting the findings of this study are available within the paper, its Extended Data Table and at https://doi.org/10.5061/dryad.1rn8pk0wm.
CRU TS4.05 at https://crudata.uea.ac.uk/cru/data/hrg/
MOD11C1 at https://lpdaac.usgs.gov/#nav-heading

MOD11C2 at https://lpdaac.usgs.gov/#nav-heading
MOD11C3 at https://lpdaac.usgs.gov/#nav-heading
EarthEnv at https://www.earthenv.org/
ETOPO1 at https://www.ncei.noaa.gov/products/etopo-global-relief-model
SRTM at https://www2.jpl.nasa.gov/srtm/
GMBA at https://www.gmba.unibe.ch/services/tools/mountain_inventory_v1
CHELSA at https://chelsa-climate.org/
GHCN at https://www.drought.gov/data-maps-tools/global-historical-climatology-network-ghcn
BioShifts at https://doi.org/10.6084/m9.figshare.7413365.v1

# Research involving human participants, their data, or biological material

Policy information about studies with human participants or human data. See also policy information about sex, gender (identity/presentation), and sexual orientation and race, ethnicity and racism.

| Reporting on sex and gender | N/A |
|---|---|
| Reporting on race, ethnicity, or other socially relevant groupings | N/A |
| Population characteristics | N/A |
| Recruitment | N/A |
| Ethics oversight | N/A |

Note that full information on the approval of the study protocol must also be provided in the manuscript.

# Field-specific reporting

Please select the one below that is the best fit for your research. If you are not sure, read the appropriate sections before making your selection.

☐ Life sciences　　☐ Behavioural & social sciences　　☒ Ecological, evolutionary & environmental sciences

For a reference copy of the document with all sections, see nature.com/documents/nr-reporting-summary-flat.pdf

# Ecological, evolutionary & environmental sciences study design

All studies must disclose on these points even when the disclosure is negative.

| Study description | The result data and figures presented in this paper are derived from published maps and datasets. |
|---|---|
| Research sample | A comprehensive list of data sources can be found in Extended Data Table 1. The mountain regions have been defined by experts, and the specific definition of islands can be found in the Methods section. |
| Sampling strategy | In general, all available data pixels were included in our analyses. Specific details for certain statistics, such as bootstraps, can be found in the Methods section, where they are further elaborated upon. |
| Data collection | Please find Extended Data Table 1 for details. |
| Timing and spatial scale | Our focus was on global mountain data spanning the periods of 1971-1980 and 2011-2020. |
| Data exclusions | Given that SLRT values approaching 0 can induce extremely high climate velocities, we excluded the 1% of outliers proximate to zero when generating Fig. 3c. In determining the relationship between MALRT and SLRT, we omitted 2% of the extreme values from the satellite data, encompassing both the upper and lower extremities. Furthermore, two conspicuous outliers derived from the weather station data were excluded before conducting the linear regression. Although incorporating these outliers might have enhanced the explanatory power of the weather station data, we surmise that such a result could be an artifact. As such, we opted to exclude these two outliers. |
| Reproducibility | The procedure has been repeated three times, and the results have consistently yielded similar outcomes, except for the steps involving randomization, which exhibit consistent but slightly different results. |
| Randomization | Randomization was incorporated into the procedure for calculating SLRT and in comparing biological range shift with MALRT. Further details regarding this randomization process can be found in the Methods section and Extended Data Figure 6. |
| Blinding | The concept of blinding treatment is not applicable in this context, as the analytical pipeline used for environmental datasets is unlikely to be influenced. |

Did the study involve field work? ☐ Yes ☒ No

# Reporting for specific materials, systems and methods

We require information from authors about some types of materials, experimental systems and methods used in many studies. Here, indicate whether each material, system or method listed is relevant to your study. If you are not sure if a list item applies to your research, read the appropriate section before selecting a response.

## Materials & experimental systems

| n/a | Involved in the study |
|-----|-----------------------|
| ☒ ☐ | Antibodies |
| ☒ ☐ | Eukaryotic cell lines |
| ☒ ☐ | Palaeontology and archaeology |
| ☒ ☐ | Animals and other organisms |
| ☒ ☐ | Clinical data |
| ☒ ☐ | Dual use research of concern |
| ☒ ☐ | Plants |

## Methods

| n/a | Involved in the study |
|-----|-----------------------|
| ☒ ☐ | ChIP-seq |
| ☒ ☐ | Flow cytometry |
| ☒ ☐ | MRI-based neuroimaging |

