## [Peer Review File · Nature]

Manuscript Title: Climate Velocities and Species Tracking in Global Mountain Regions

Reviewer Comments & Author Rebuttals

Reviewer Reports on the Initial Version:

Referees' comments:

Referee #1 (Remarks to the Author):

As the global climate warms, mean temperature isotherms will move uphill in mountain regions, and it is important to attempt to quantify this process – particularly the rate at which this is occurring. The rate of movement (in a particular mountain range) will depend on the amount of regional warming, coupled with the lapse rate (the rate of decrease of mean temperature with elevation). This paper attempts to calculate shifting elevation limits of isotherms for mountain regions worldwide, which is a worthwhile endeavour, but in my view is not 100% successful in doing this.

The main assumption, critical to the whole paper, is that the mean lapse rate in a mountain range can simply be determined using mean temperature and atmospheric moisture content alone – i.e. by deriving the moist adiabatic lapse rate or MALR. The MALR is a theoretical construct which applies to the free atmosphere and tells us how fast the temperature should decrease with elevation within a cloud when latent heat of condensation is released. At higher temperatures more moisture can be released which results in a shallower lapse rate. Lapse rates of mean temperatures on the ground surface however (relevant for migration of species) are not the same as the theoretical free-air MALR for many reasons. The first is that the Earth's surface does not behave the same as the free atmosphere, the mountain slopes being subject to local heating (in the sun for example) and cooling (e.g. ponding of cold air near the ground at night). The second is that the MALR only applies when there is condensation or cloud formation (i.e. not all the time). In dry conditions or when clouds form well above the mountain summits, by day the DALR should apply (which is much steeper), and at night the lapse rate often reverses due to cold air drainage and other local scale processes. The end product is that the surface-based lapse rate of mean temperature is nowhere near the MALR in many cases, and varies substantially in mountains worldwide. Because the lapse rate used in this paper is unrepresentative, so will be the calculated climate velocity and therefore the comparison with species movements may also be flawed.

As well as the theoretical problems, the way the MALR is calculated in this study is also arguable. The dataset CRU TS 3.24 is not well explained. I think it comes from the reference below (I searched online) because it is not referenced in the paper (a major omission). It is critical to cite the original reference and acknowledge the datasets fully.

Harris, I., Jones, P.D., Osborn, T.J. and Lister, D.H. (2014), Updated high-resolution grids of monthly climatic observations – the CRU TS3.10 Dataset. *Int. J. Climatol.*, 34: 623–642. doi: 10.1002/joc.3711 https://crudata.uea.ac.uk/cru/data/hrg/cru_ts_3.24.01/Release_Notes_CRU_TS3.24.01.txt

It is stated that mean temperature and vapor pressure from 2011-2015 (a short five year period) are used to calculate lapse rates, but surely as temperatures change over time, so could the lapse rate. The 5 year moisture (and temperature?) values are combined with a 35 year “rate of warming” to calculate climate velocity. Would not 35 year mean lapse rates be more accurate? 5 years is not really long enough to eradicate short term variability. It is also the case that since the MALR depends on temperature, it will increase (become steeper) with elevation (i.e. the actual MALR profile at anyone time is not even linear) but in this paper only one lapse rate appears to be calculated for each pixel based on 5 year means (at least I think so, but it is not clearly explained).

There is next to no information about the data (i.e. stations) that go into the CRU dataset – so there is

no guarantee that interpolated temperature and moisture values are realistic, particularly in mountain regions which are the focus of this study – some discussion of this issue at the very least (and quantification of possible error or uncertainty in lapse rate estimation) would be expected in an article in this journal.

Finally, this whole method assumes that the lapse rate does not change in a warmer climate, and that the climate velocity can be obtained for a given location if the lapse rate is known (i.e. all isotherms are moving uphill at the same rate). The situation whereby lapse rates could steepen (isotherms converging closer together on a slope) or weaken (isotherms spreading further apart) is not acknowledged. Yet we know that elevation dependent warming is common in the literature (reference 10). This is another way of saying lapse rates may change in a warmer world and the spacing between isotherms may change. The 0degC isotherm may shift by a different amount in comparison with the 10degC isotherm, and therefore climate velocity in a particular mountain range at a particular location cannot be summarised with one rate. It may depend on the critical temperature being discussed. I am wondering whether the derived MALR could at least be compared with the observed lapse rate in the vicinity of the pixel in the CRU dataset (for the same 5 year period) – this would tell us whether using the theoretical MALR is anywhere near the actual mean lapse rate recorded on the mountain surface – at least in the historical data.

So overall, I find that there are quite a lot of conceptual issues raised by this paper, and I do not think the authors have considered these limitations or addressed them. I am not an ecologist and therefore I confine myself to the climate aspects and the derivation of climate velocity estimates, but these are central to the paper's findings since it is against this that the species shifts are compared. I cannot therefore recommend publication in its current form, especially in a high-profile publication like Nature.

My more specific comments below refer to line numbers

Lines 61-63. This surely incorrect and you divide the surface temperature change by the lapse rate (not what you state) – see line 90 where you state the opposite.

Line 74: the MALR is always lower (not often) – and it is not -6.5degC/km. This figure is a global mean lapse rate with no physical basis.

Line 82: This statement is vague.... Does this mean 2.5degC/km in late summer but 7.5degC/km in spring – or both values in both seasons etc...

Line 97: What is a "mountain grid"? – just one grid point.... You normally calculate a lapse rate between locations (so you would need more than one grid point). This needs clarification.

Line 102: The rate of surface temperature change and the mean temperatures/vapour pressures are from different periods... this does not seem appropriate

Line 116: I would refer to lapse rates as shallow or steep (not high and low which is ambiguous).

Line 144: Yes, but your method assumes that the lapse rate does not vary and so specifically outlaws elevation-dependent warming (see my earlier comment)

Line 148: Since temperature decreases with elevation, your MALR will increase with elevation, so climate velocity will decrease with elevation (assuming the same amount of warming). Surely this is in part intrinsically built into your assumptions, so it is not surprising you get this result. What is more surprising is the accelerated climate velocity with elevation in the tropics.

Line 158 ff: It is unclear why you combine NW/SE and SW/NE aspects for example. Some more explanation is needed. Opposing aspects will be the most different in terms of climate regimes and microclimates so why combine them into one category? This will have the effect of damping the differences between categories because you are missing main windward/lee contrasts which occur within the categories not between them.

Line 164 ff: The main result of the topographic analysis (Figure 4) is that water vapor appears to be higher on E-Equator mountains. Many of the other differences are insignificant. This result I think is because upslope winds come from the SW (in the northern hemisphere) and NW (in the southern hemisphere) and these directions align with the prevailing moisture tracks in mid-latitudes, so the orographic enhancement effect is maximised where the ranges directly face the moisture source.

Line 207: Many ecological studies rely on lapse rates from sounding balloons? Do they? Which? References need to be given here. An observed free air lapse rate is probably better than the theoretical MALR, although both are problematic.

Line 221. Yes, this is a catch all sentence which sums up why the MALR is of limited use, and why we need more monitoring on the ground (i.e. the last sentence at line 227-229 which I agree with).

Line 234-236 – Both temperature and vapor pressure were “averaged across coarse spatial extent”? What does this mean? This is vague but also concerning.

Line 238: What are T and T2 at a grid point – surely a grid point is just a point and only has one value. This relates to my confusion about how a temperature lapse rate can be derived from a grid point.

This whole explanation is unclear.

Line 254: This implies averaging from monthly to annual values – which again hides seasonal variability. The assumptions behind this are never discussed.

Line 270: Diagrams or maps would make the explanation of elongation and orientation clearer, and how they relate to aspect. You never define how aspect is calculated. It sounds as though it is applied to a subset of elongated (Sierra-like) mountain ranges.

Line 303: At this point you say a grid is 1 x 1 degree resolution... which implies it is multiple cells (the actual data is 0.5 degree resolution – line 234?). So for the first time it appears a grid is more than one cell or pixel (but this is unclear).

Line 312: So the studies may be from different time periods, but the lapse rates and climate velocity are always from 2011-2015. This seems like a limitation.

Line 352 ff: This last section is incomprehensible to me, but I admit I am not an ecologist – I did try to read it several times, but I think English hampers the explanation.

General point about figures: it is confusing that there are three levels of figures: main figures, extended data figures, and supplementary figures. This may be the style of the journal but it is so confusing for the reader. I am not convinced that all 17 figures are needed.

Extended data Figure 2: Why do some climate velocities have error bars but not others?

Extended data Figure 3: Not needed

Extended data Figure 6: I get the feeling that this could be important as some sort of validation attempt, but given the current labelling I find it difficult to interpret. One interpretation appears to be that using MALR is not that different from using a standard rate of 5.5degC/km (yellow symbols) apart from some outliers which does not make sense to me.

Referee #2 (Remarks to the Author):

NOTE: the review documents for download did not include the mandatory “Reporting Summary” document, so my review does not include that important document.

The manuscript by Chan et al. is, in some ways, a re-analysis and update of the meta-analysis published by Chen et al. (Science 2011), focusing explicitly on elevational shifts in organisms in response to temperature, rather than both elevational and latitudinal shifts. The current manuscript starts with a quandary: Chen et al. 2011 reported surprisingly little concordance between predicted elevational shifts in regions (based off of warming temperatures) and observed elevational shifts. Chan et al., here, go on to try to fix this issue by re-calculating expectations (based off the moist adiabatic lapse rate) of shifts, and then testing these new expectations against the previous meta-analytic data of empirical range shifts. The vast majority of this present manuscript is not about ecology, per se, but about variation in moist adiabatic lapse rates (and, by proxy, elevational climate velocities) around the world and in differing conditions. Nevertheless, the manuscript concludes that Chen et al. (2011) were overly-pessimistic about elevational tracking, and that, with updated

expectations, more species appear to be tracking than previously thought. The most interesting finding is that elevational tracking appears to occur more in areas with lower thermal velocities, implying that high climatic velocities inhibits the ability of species to track climate.

Greater tracking at lower climate velocity is very intriguing, for the implication stated above, but unfortunately cause is very poorly connected to effect in this case. In particular, low climate velocity is coincident with a wide variety of other things, each of which could also impact the degree of (real or perceived) climatic tracking, therefore leaving the interpretation ambiguous. For example, as the manuscript describes, higher velocity occurs in more arid regions, where precipitation is likely to play a relatively greater role as a mechanism. This study only looks at average thermal niche tracking, and ignores all other aspects of the climate (e.g., seasonality, extremes, and non-thermal variables), many of which now have well demonstrated mechanistic links impacting range dynamics over time. One variable of importance for species is precipitation, which has been implicated as being critical for structuring range shifts for many species, from trees to mammals. Using an example dataset from this very meta-analysis, Tingley et al (2012) demonstrated that elevational range shifts were heterogeneous, and that the direction of range shifts (which could be calculated as the degree of climatic tracking) was equally influenced by precipitation as temperature. Yet in the present manuscript and meta-analysis, only thermal tracking is evaluated. This is a problematic limitation because, as stated above, the manuscript finds that tracking fails in areas of high climatic velocity, which are also described as being arid areas. Well, arid areas are exactly the areas where we would most expect species to track precipitation – perhaps even more so than temperature. So is the failure of species to track temperature in areas of high thermal velocity due to the fact that the thermal velocity is high, or is it because species aren't tracking temperature at all in those areas? The inability of this study to differentiate between these two well-established ecological hypotheses is a major limitation, and ultimately means that I don't feel the present manuscript adds much new to the already expansive literature on elevational range shifts and their trends.

Line Edits:

53. Is it unclear? I don't know any study that suggests conclusively that species are keeping pace – whereas many studies suggest that species are lagging.

60. "No opportunity" seems to be an incorrect interpretation of the data and their findings.

67. Many studies correct for variation in actual lapse rates; for example, Elsen et al. 2017. Saying that actual lapse rate variation hasn't been considered in ecological research is an overstatement.

142. I found the statement about the importance of climate adaptation strategies to be a throw-away line that lacked the specificity necessary to be convincing. Climate adaptation is a huge and growing area, and some strategies are more effective than others. Just saying developing strategies is "important" is overly vague.

145. This study here suggests that it has been hard to study warming in mountainous regions due to poor climatic sampling of mountains regions. This is true, but I don't understand how this study escapes from that same limitation. The estimates of both lapse rates and climatic velocities assumes the validity and accuracy of the changing temperature over time in mountain regions. It seems like the current trends in climatic velocities could easily just be artifacts of poor modeling of mountain temperatures in the CRU data. How do we know that the failure to track high climatic velocity results isn't an artefact of poor CRU data in those particular regions?

154. The relationship described in this section (on slope and aspect) is interesting and the results are largely expected, but I also imagine that these results are entirely scale-dependent (i.e., dependent on the spatial scale of the unit of analysis). The manuscript only analyzes these trends at a single scale, which may not be a scale that is representative of biodiversity mechanisms.

References:

Elsen, P.R., Tingley, M.W., Kalyanaraman, R., Ramesh, K. and Wilcove, D.S. (2017), The role of competition, ecotones, and temperature in the elevational distribution of Himalayan birds. *Ecology*, 98: 337-348. <https://doi.org/10.1002/ecy.1669>

Referee #3 (Remarks to the Author):

A. Summary of the key results

The authors show larger variations in climate change velocity across the world's mountains than previously acknowledged, which they attribute to over-simplifications in the way previous models were built. They also cross-analyse the literature on species' ability to track climate change along mountain slopes, both in continents and on islands, concluding that many species not moving fast enough – which implies a major and underestimated threat to mountain diversity worldwide.

B. Originality and significance: if not novel, please include reference

The models are new as well as the region-specific estimates for climate change velocity. These estimates are likely to play an important role in climate change research, in particular for mountainous regions since they have been challenging to model properly due to a variety of reasons (natural and artefactual). Although the authors promote the main significance of the work for biodiversity science, their models and data are likely to have important impacts on other areas of knowledge and practice, such as sustainable agriculture, climate change mitigation to human settlements and livelihoods, the spread of diseases, water security, etc.

To me the most interesting result was the one presented in Fig. 2D, which I think will have a particular impact in the way that those regions may be targeted for further fine-scale climate change research and monitoring.

C. Data & methodology: validity of approach, quality of data, quality of presentation

The text is easy to follow and compelling, explaining complex terms and formulae in accessible ways even for a non-climatologist. The figures are quite good, although some labels and information could be made clearer to read (eg the use of a black background, and colours against a grey map, are somewhat challenging although I am not sure how much better they could be).

I am not able to assess the validity of the approach used to derive the climate models.

The choice of biological datasets is well explained, but I was somewhat disappointed that not more studies were available for the comparative (meta) analysis. As a result, the results were heavily biased towards certain parts of the world (e.g., 17 out of 47 studies were from the UK, and very few from eg Latin America or Africa). I would have imagined (but do not know by fact) more suitable

datasets to be available, considering vast international collaborations such as GLORIA (Global Observation Research Initiative in Alpine Environments), ITEX (International Tundra Experiment), and large data providers such as sPlot.

The definition of 'mountains' (l. 87) does not differentiate flat areas above 1000m (eg, high plateaus) from those that are indeed areas of varying relief, which are the ones most relevant for biodiversity shifts along slopes. See e.g. the proposal in <https://link.springer.com/article/10.1007/s00035-016-0182-6> . It may be adequate to evaluate whether the findings of this paper are robust to different mountain classifications.

l. 108: it would be good to clearly define 'mountain areas' – are individual cells, or mountain ranges / individual mountains what the authors refer to? This has biological implications: if a whole range is under high climate velocity, species may struggle to keep track of their climatic niche, but if there are individual cells of low velocity within a mountain range that on average has high velocity, this could mean that those particular cells are especially important for providing rescue to species distributed across the range (a point the authors may want to make).

l. 109: the use of 'biodiversity hotspots' should be defined, as it is used in different ways in the literature. Do the authors refer to Myers' original definition and subsequent updates?

l. 199-200: it would be good to expand here, in quantitative and/or qualitative terms, on the result that "mountain species are generally lagging behind the climate velocities" – the reader is now referred to two figures but this is such an important result that a brief summary seems well placed here.

l. 227: many papers end with similar statements of what should be 'priority areas for conservation', but in truth we know this has little value unless a much more solid recommendation can be provided. As mentioned above, perhaps the single low-velocity cells in the 24 mountain regions with overall high climate velocity might be a more tangible focus for conservationists, provided that they still contain natural habitats and biodiversity enough to allow such rescue. High-resolution maps and data from this study should be provided in formats amenable for use by those truly taking these findings into account in spatial conservation prioritisation.

D. Appropriate use of statistics and treatment of uncertainties

I do not feel qualified to properly assess the choice of statistical methods, but my impression is that the authors have considered and incorporated several sources of uncertainties in their analyses and when reporting specific results. There should be further assessment of uncertainties related to the biodiversity datasets, or at least a discussion on the potential impact of biases (see also below).

E. Conclusions: robustness, validity, reliability

The authors mention in some places the lack of data availability for weather stations, but there is little discussion on the potential impact of the relatively scarce biodiversity data on the precision and accuracy of the analyses, and the conclusions derived from them. This is by no means the authors' fault – as this is a well-known and general problem – but it would be good to discuss further the potential effect of data biases (eg taxonomic and spatial) in biodiversity data. For instance, are the results for Italy and France in extended figure 6 true outliers, or related to denser data for those countries? It may also be helpful to plot the locations from Sup Table 2 onto a global map.

The biodiversity conclusions in the abstract (l. 38-42) are very general and unquantified – 'more

cases'... 'generally lagging'... I think they should be more precise and concrete.

The last lines in the abstract (42-45) make some strong claims in conclusion, but I could not find where in the data or analyses those claims were directly derived.

F. Suggested improvements: experiments, data for possible revision

The biological implications are rather thin – which seems a bit odd, since the main interest and implications of these findings will most probably be on the effects of climate change precisely on biodiversity. However, I think several interesting aspects could be further explored with the data at hand. For instance, what are the taxonomic (eg orders, families) and functional (eg trees/herbs) groups / mostly affected (i.e., not tracking climate change)? Are those groups/species phylogenetically constrained or randomly spread across the tree of life? Are there any particular biomes or biogeographical regions that are mostly affected? And so on – so that the reader really can *learn* more about potentially losers (and perhaps winners) of climate change in mountains. Fig. 4e covers only very large groups, and the figure is a bit too crowded to really extract useful information. Right now, there are not as many general learnings derived from the study as there could certainly be.

G. References: appropriate credit to previous work?

There is a vast field of research in this and related fields and the study does a good job in referencing what I perceive as some of the most important studies. I understand there are word limitations, but some potentially additional research of relevance would be:

- Work by Kenneth Feeley, Miami (there one publication cited)
- Work by Christian Körner, Basel
- <https://science.sciencemag.org/content/334/6056/660>

H. Clarity and context: lucidity of abstract/summary, appropriateness of abstract, introduction and conclusions

Although the text is well written and flows well, there are a few minor language improvements/typos that could be improved (e.g. line 55 'if they were TO track climatic changes...'; l. 222 delete 'in'; l. 327 'taken cared here'). In particular the Methods session could profit from careful external editing for style.

Similarly, there are some instances of unnecessary repetition (e.g. line 67 repeating line 62).

Please ensure that all tools etc are properly described and referenced upon first use (eg pySpark, l. 255, R in l. 350), as well as abbreviations (eg CRU).

Signed by
Alexandre Antonelli, 17 Feb 2021

Referee #4 (Remarks to the Author):

Mountain ranges provide important refugia for species in times of rapid global change. In order to find a favourable climate, on mountains, organisms just have to travel few kilometres while along

latitudinal gradients a comparable change in climate would demand travelling several thousands of kilometres. This makes mountain ranges particularly relevant for the conservation of biodiversity over the next decades of climate change. But how fast is the climate changing with elevation in global mountain areas and are mountain species able to track it? These are the important questions the authors deal with in their paper "Climate velocities and lagged species elevational sifts in mountain ranges". Chan et al., first, quantify the MALR globally based on thermal dynamic theory and data on temperature and water vapor. Second, the authors quantify, again globally, the velocity of climatic change (using data on ground temperature over the last decades and estimated for the MALR) and identify regions showing high speed of change. Third, they use data on temporal changes in biodiversity (several data sets sampled around the world from the literature) to test if species can track their climatic niches and if this differs between mountains showing high or low speed of climate velocity. They find strong global variation in the MALR and in the velocity of climate change on mountains which can better explain a lag of climate tracking by species than previous data sets.

I think that the topic of the paper is of interest to readers of several disciplines as it addresses the consequences of climate change and produces a new global data set on the rate of climate change in mountain areas. To my best knowledge, this is something really missing in the literature.

However, as I point out below I am currently (based on the data presented in the paper) not convinced about the analysis conducted by the authors. My main point is that the authors, as far as I can see, do not present any validation of their model predictions for the MALR and for the predictions on climatic changes along elevation gradients over time. I think this is important as the variation in MALR and the variation in climate velocity is presented as a key finding and touches all subsequent analyses.

Please find below some detailed critic:

- The authors use a very "rough" definition for mountains. They simply define mountains as all areas >1000 m asl. Classical definitions of mountains additionally or exclusively consider the steepness of the area. This has considerable consequences: For example, in Chan et al. the steep Andean eastern slopes below 1000 m are not considered as mountains while large parts of southern Africa are considered as mountains even though the area is quite flat (it is simply just above 1000 m). I do not know if this has any influence on the results. Might be that this is just a minor critical point but I am quite sure that people working mountain ecology, geography or climatology will see this critical. I would here suggest to recalculate the key results additionally for other mountain definitions and put this into the supplements, e.g. for the standard definitions by Körner et al. 2016, Kapos et al. 2000

- The MALR and climate velocity data is calculated from a model based on terrain surface temperature and vapor pressure of the CRU TS 3.24 data set. However, the authors do not present any validation of their predictions using true climate station data from mountains. What is the error in the prediction? Even though the authors write that climate station data is rare for mountain areas there should be some data sets (e.g. Appelhans et al. 2016 report a lapse rate of 5.5°C per km elevation for the southern Kilimanjaro and not a rate of about 3.5 as suggested by the model). I think that a check of the accuracy of the predictions is extremely important as the variation in MALR and consequent changes in climate velocity is presented as one of the major findings of the paper. All the results are depending on the accuracy of this prediction. It is therefore at this point hard to judge the value of paper if this information is missing.

- The CRU data sets offer a resolution of 0.5° so ca. 50 x 50 km. If I think of isolated mountains this may cover both the rainy and the dry slopes of mountains. So, if moisture plays a role, the model will predict for both sides of the mountain the same MALR but I guess that the species data sets used for

testing range shifts in the paper were often sampled on a specific side of the mountain which could lead to potential errors in the predicted change. Could that be a problem?

- CRU TS 3.24 was used for deriving temperature and vapor pressure data: I checked the CRU download area and saw that version 3.24 was withdrawn due to errors. Did you really use this version and not the corrected version 3.24.01? I see that there are always some delays from analysis to the final paper but I still wondered why not the more recent versions were here used (CRU TS 4.x). Additionally, I guess that there is more than one climate model than CRU TS which could be used here. It should be justified why the authors used this climate data set and not others. Generally, as the CRU TS data set is the base of all analyses I would introduce it with 2-3 sentences in the main text.

- 16 of the 41 data sets (39%) which were used for testing the relationship between elevational shifting rates of species and elevational climate velocity are from the UK, which are not identified as a mountain region in the paper (as far as I can see there is not a single pixel of mountain area indicated here...using the definition of areas > 1000 m). This should be aligned (either use another definitions of mountains, or exclude it from the analysis, or justify the use of the data)

- It does not become clear how much better the new predictions of climate velocity are in comparison to the older estimates for predicting the observed shifting rate. The authors write that a general lag of upslope migrations observed in the past literature can be explained by wrong estimations of climate change on mountains but they now also found a quite general lag of upslope migrations.... I would here recommend to add some analysis how much better the new climate velocity data fits to the range shifts compared to the old data. Additionally, I find the observation of a general lag in species range shifts on mountains (which is not so much observed along latitudinal gradients) still really interesting – particularly given that the dispersal distances for tracking climate are some orders of magnitude smaller than along latitudinal gradients. I saw that in an earlier Science paper by some members of the author team this finding is discussed but I would advice to discuss this also briefly in this paper.

Minor comments:

234-237: It did not fully become clear to me if the authors just used the TS3.24 data set or if they did some additional analyses using one weather station records here. If the former is the case, I would simply add: "In the TS3.24 data set, both mean annual temperature....." to make this clear.

It is really difficult to understand how the authors calculated the probability of species tracking the climate velocity. In particular from the main text but also from the methods: Here it is stated that "First of all, we used the bootstrap technique to subsample the dataset to control the inconsistencies induced by having different sample sizes across studies. For each taxon in each region, we set the sample size to n and drew n records (n in Fig. 4a)." But what is a record? I recognized that two different kinds of data sources were used here but even for detailed species data it is unclear to me what a record is: a species? Or the single observation of a species on a study site/elevation? I think that this could be described more precisely so that it is easier for the readers to follow what the authors tested here.

I would also advice to briefly describe the results based on the older approach of directly predicting mean change in species elevations to the mean change in the shift of temperature along the elevation gradient as this approach is more direct and easier to understand.

To sum up, I think that the paper has potential but it remains unclear how much the climate predictions (MALR and climate velocity) reflect reality.

Referee #5 (Remarks to the Author):

General comments

A. Summary of the key results

Using a formula to compute moist adiabatic lapse rate (MALR), instead of using a constant adiabatic lapse rate (ALR) of ca. 5.5°C to 6°C per km of elevation, the authors aim at: (1) refining the velocity at which isotherms are shifting vertically along the elevational gradient in mountain regions worldwide and (2) assess whether this new estimate of climate velocity better explains the velocity at which species are shifting upslope as climate warms. Deriving MARL, at 0.5° resolution (ca. 55 km by 55 km at the equator) across all mountain regions on Earth, the authors found that MARL ranges between 3 to 9°C/km (vertically). Then, the authors computed, for each spatial grid cell of 3025 km², the temporal change in temperature conditions between 1971-1975 and 2011-2015 and divided this value by the amount of years over this time period to generate a temporal gradient in °C/yr. Finally, the authors divided the "local" MARL value (°C/km) of the focal grid cell by the "local" temporal gradient (°C/yr) of the same grid cell, which gives the vertical velocity of isotherms along elevational gradients in km/yr or in m/yr (when multiplying by 1000). Using MARL, instead of a constant ALR value, the authors found that the vertical velocity of isotherms along elevational gradients ranges between -16.67 m/yr to 16.8 m/yr, averaging 5.42 m/yr, which is slightly more than the average vertical velocity found with a constant ALR value (ca. 4.56 to 4.98 m/yr) (Extended Data Fig. 1). When relating the velocity of isotherm shifts with biodiversity data (velocity of range shifts), the authors found that the probability for species to lag behind climate warming is higher when the vertical velocity of isotherms is higher.

B. Originality and significance

This is a very interesting study and the novelty of the authors' study clearly lies on the use of the MARL to compute the vertical velocity of isotherm shifts under climate change. Indeed, by doing so, the authors genuinely account for the spatial variation in water vapour pressure on the value of the adiabatic lapse rate (ALR). Yet, I do have four major and important concerns (listed below throughout the different sections of my general comments) that I think the authors should consider carefully to improve the quality of their work.

First of all, considering novelty statements, the authors are wrong when claiming that the use of "local" or spatially variable and spatially-explicit ALR has not been explicitly considered in ecological research (see the authors' statement in lines 65-67). In fact, a very recent study (Lenoir et al. 2020: reference #9 in the authors' reference list) did also derive local and spatially-explicit ALRs before relating the local vertical velocity of isotherm shifts along a given elevational gradient to the observed velocity of species range shifts along the focal elevational gradient. However, Lenoir et al. (2020) did not account for the effect of varying water vapour pressure on local ALRs, which is what I would consider novel in the authors' study. Hence, I suggest the authors to better emphasize what is the main novelty in their study (cf. accounting for variation in water vapour pressure) and to make a direct comparison with the approach used in Lenoir et al. (2020) to compute local ALRs. By doing so, it will be clearer for the reader what is the main novelty in this study. Besides this issue of better justifying novelty in light of the most recent scientific literature (cf. reference #9), I would also recommend the authors to carefully check the most recent meta-analyses on elevational range shifts: Guo et al. 2018; Rumpf et al. 2019; and Mamantov et al. 2021.

C. Data & methodology: validity of approach, quality of data, quality of presentation

About the validity of the overall approach, it seems that the authors used temperature data at a very coarse spatial resolution (CRU data at 0.5° which is about 55 km by 55 km at the equator) to compute the vertical velocity of isotherm shifts along elevational gradients. This is my second major concern given how fast temperature are changing across 1 km distances in mountain regions (cf. topoclimatic variation) (Scherrer & Körner 2010). Because of that, the authors may underestimate the availability of local escapes for species redistribution in mountain regions (Scherrer & Körner 2010) and thus overestimate the velocity at which isotherms are actually shifting upslope as climate warms. Given the authors' main focus on the most recent period of climate warming (1971-2015) (see extended Data Figure 8), I would recommend the authors to use finer global climatic grids such as TerraClimate (<http://www.climatologylab.org/terraclimate.html>), WorldClim (<https://www.worldclim.org/>) or CHELSA (<https://chelsa-climate.org/>) to compute local ALRs using the same approach as in Lenoir et al. (2020) (see the subsection entitled "Climate velocity" in the Methods section) for comparison purposes with the "local" MALR the authors derived.

My third major concern is about data quality and the representativeness of the biodiversity data that the authors used for assessing the magnitude of observed species range shifts (cf. Supplementary Table 2). Indeed, looking more closely at the content of Supplementary Table 2, I am afraid the authors are missing quite a lot of important data on species elevational range shifts, including data from within their literature search period (up to 2017) (e.g. Angela & Daehler 2013; Bodin et al. 2013; Brusca et al. 2013; Dainese et al. 2017; Frei et al. 2010; Koide et al. 2017; Kuhn et al. 2016; Ploquin et al. 2014; Rowe et al. 2015) as well as some of the most recent studies (e.g. Geppert et al. 2020). Most of these data are now freely available throughout the BioShifts database (Comte et al. 2020). Hence, I suggest the authors to at least download the BioShifts database to get a more comprehensive set of raw data on species range shifts along the elevational gradient. It is rather important that the authors make sure to use the most updated dataset on species elevational range shifts. As it is now, the picture depicted by the authors in their main findings might be biased and far from representative of the current knowledge on species elevational range shifts (see also the most recent meta-analyses on species elevational range shifts: Guo et al. 2018; Rumpf et al. 2019; Mamantov et al. 2021).

D. Appropriate use of statistics and treatment of uncertainties

Finally, my fourth and last major concern relates to the statistical analyses the authors used to link the magnitude of observed shifting rate for plants and animals distributed along elevational gradients against the velocity at which isotherms are shifting vertically as climate changes (cf. Fig. 4). Indeed, except for balancing sample size among studies, the authors did not really correct for methodological differences among the studies from which they extracted data on species range shifts. Yet, it has been clearly demonstrated in the scientific literature that methodological biases can account for a very substantial variation in the data (Brown et al. 2016 and reference #9). Such methodological differences among studies must be accounted for either by running mixed-effect models on the raw range shift data or by running a dedicated meta-analysis to compute the pooled effect size. None of these methods or approaches have been used by the authors here, which is rather problematic. For instance, using the "metagen()" function from the "meta" package in R and specifying the method argument to the Sidik-Jonkman method, the contribution of a given study to pooled effect size is weighted by sample size and the degree of variation in the study's data, such that a given study with many species range shift values and little variation in range shifts across species has a stronger influence on the value of the statistic than a study with few species and a high level of range shift variation. Alternatively, the authors could use mixed-effect models and add study ID as a random intercept (at the very least) in their models when relating the vertical velocity of isotherm shifts to

observed shifting rate across several taxonomic groups. Also, the authors should consider to add taxonomic information as a random term in their models (see reference #9 for a similar approach) to account for potential phylogenetic signal in the residuals of their models. Hence, there is no need to subsample the data to balance sample size, as the authors did when computing the probability of species tracking climate velocity, one just needs to use an appropriate statistical tool: either a true meta-analysis approach (e.g. Mamantov et al. 2021) or a mixed-effect modelling approach on the raw data (e.g. reference #9).

E. Conclusions: robustness, validity, reliability

Given the four major concerns I have listed above and highlighted in parts B, C and D, I am afraid that the robustness, validity and reliability of the authors' main findings (and thus conclusions) are questionable in their current state, thus requiring more work to achieve a greater level of robustness, validity and reliability (see suggested improvements below and in my specific comments).

F. Suggested improvements: experiments, data for possible revision

As already mentioned in each of my four major concerns, suggested improvements include: (1) to better emphasize the novelty of the authors' work that lies on the use of an ALR metric that account for spatial variation in water vapour pressure whereas former work deriving local ALR did not account for the effect of water vapour pressure; (2) to derive local ALRs, based on spatially fine resolution climatic grids from WorldClim or CHELSA, following the method already used in reference #9 and for comparison purposes with the approach proposed by the authors (MARL); (3) to improve the representativeness of the data on species elevational range shifts by downloading the BioShifts database and querying raw data on species elevational range shifts; and (4) use more appropriate statistical tools, such as mixed-effect models or a true meta-analytical framework, to better account for methodological differences among studies.

G. References: appropriate credit to previous work?

Overall, the authors are providing appropriate credit to previous work, but see my suggestions in part A as well as the list of references I am providing at the end of my review and after my specific comments to the authors.

H. Clarity and context

Overall, the manuscript is well written but sometimes the methods are not clearly described and quite difficult to follow. Some information are not clearly provided in the main text or the Methods section but are hidden in the captions of the figure, such as the way the authors computed the temporal gradient in temperature changes. I have provided several suggestions in my specific comments to the authors to improve clarity in the main text and the Methods section.

Specific comments

Line 31: What do you mean by "local surface temperature" here? What is the spatial resolution you considered here to refer to local? According to the coarse spatial resolutions you used in your analyses (0.5°, which is about 55 km by 55 km at the equator), this does not sound very local to me but rather regional, no?

Line 33: Do you mean "Considering" instead of "Consider"?

Lines 52-53: Here, stating that “whether species are keeping pace with shifting climates remains unclear” is a bit of an overstatement (see reference #9 for an attempt to study this important question). We know have a better knowledge on whether species are keeping pace with shifting climates. More specifically, we know that marine species are more likely to track the velocity of climate change whereas terrestrial species are more likely to lag behind climate change and that it depends a lot on baseline climatic conditions and human pressures on the environment (see reference #9).

Line 54: “makeS it” rather than “make it”?

Line 55: “if they were TO track” instead of “if they were track”?

Lines 59-60: See also findings from reference #9 and especially Fig. 5a for the most likely drivers behind the lag.

Lines 65-67: This is not true, see the Methods section in reference #9 for an alternative method on how to derive local and spatially-explicit adiabatic lapse rates that are region specific. There is also a very rich scientific literature that could and should be acknowledged here regarding the spatial variability of the adiabatic lapse rate (see Rolland 2003; Kirchner et al. 2013; Cordova et al. 2016).

Line 72: Are you sure about that value of 9.8°C/km? Do you have a reference for justifying this value? In the scientific literature I know on adiabatic lapse rate (see Rolland 2003; Kirchner et al. 2013; Cordova et al. 2016), this is not the value that I found.

Lines 79-82: See also Rolland (2003), Kirchner et al. (2013) and Cordova et al. (2016) for values of adiabatic lapse rates from other mountain regions (Andes and the European Alps).

Lines 87-88: This is way too coarse (3025 km²) for mountainous regions. Why not using finer spatial resolutions (cf. 1 km²) that are available from WorldClim and CHELSA? Regarding the definition of mountain regions worldwide, I would strongly recommend to use Körner et al. (2017) rather than such an arbitrary threshold of 1000 m because 1000 m can be considered lowlands close to the equator but definitely not towards the poles. Indeed, the Scandinavian mountains are barely visible on your maps (cf. Fig. 1) based on this very crude definition of mountain regions. Please have a look at Körner et al. (2017) and especially Fig. 4 for a more comprehensive coverage of mountain regions worldwide. Besides, shapefiles of mountain regions are available for free (see Körner et al. 2017).

Line 96: At which spatial resolution? Why only considering the 2011-2015 period to compute MARL? Why not computing the mean MARL over the entire study period 1971-2015? Have you considered that MARL may also change over time? Maybe worth discussing, no?

Line 97: How exactly did you assess the temporal rate of temperature change? Sorry, but this is not clear, neither here nor in the Methods section. I think I found the information (you compared the period 1071-1975 with the period 2011-2015) while scrolling the Figures (e.g., see caption in Fig. 2) and the Extended Data Figures (e.g., see caption of Extended Data Fig. 2). Is that how you did: for each grid cell, the mean during 2011-2015 minus the mean during 1971-1975 and then dividing the difference by 45 years between 1971 and 2015. Is that right? Sorry, but I clearly need more information on how the rate of temporal temperature change has been assessed. This needs to be clearly explained at least in the methods section. Besides, you should probably consider a more comprehensive assessment of the temporal change in temperature conditions, which would be to use all data from all years over the study period (1971-2015) and run a linear regression of mean annual temperature against time (year) to extract the specific slope coefficient estimate in °C/yr (see

Methods section in reference #9). Note also that a more relevant approach to compute the vertical velocity of isotherm shifts as climate warms for subsequent comparison with the velocity of species elevational range shifts (cf. biodiversity data), which is what you are aiming at, is to use the same time period as the one covered in the different studies that assessed species elevational range shifts (see the Climate velocity subsection in the Methods section in reference #9 for a more detailed description of the approach). Indeed, comparing the rate of temperature change between 1971-1975 and 2011-2015 against the velocity of species elevational range shifts that was estimated between, let say, 1995 and 2005 is rather problematic given the temporal mismatch between periods covered by the focal study and the study period you used to assess the vertical velocity of isotherm shifts.

Lines 100-101: Maybe it is worth discussing here this finding in light of the existing scientific literature on the adiabatic lapse rate in different regions of the world (see Rolland 2003; Kirchner et al. 2013; Cordova et al. 2016). Are the findings reported here matching with the existing scientific literature focusing on a restricted set of mountain regions? Maybe worth considering an independent validation based on data from the scientific literature?

Lines 103-106: Is this distribution of climate velocities based on MARL statistically different from the distribution of climate velocities based on static ALR values (cf. Extended Data Fig. 1)? It looks rather similar although slightly shifted to the right for values based on MARL. Maybe worth assessing how much they differ from methods based on constant ALR values. What about comparing the distribution of climate velocities based on MARL with the distribution of climate velocities that you would obtain based on local ALRs following the approach described in reference #9? This is worth testing given the focus of your study on MARL. What do you think?

Lines 119-123: What about local ALRs using the methods described in reference #9? Would this lead to similar findings or is the MARL leading to different patterns: does it matter to account for water vapour pressure when assessing local ALRs?

Lines 128-130: Good that you could do this despite the rather coarse spatial resolution (55 km by 55 km) you used in your analyses. Maybe consider using a finer spatial resolution of 1 km².

Line 132: Please consider using polygons provided in Körner et al. (2017) instead of using an arbitrary threshold of 1000 m for considering a region as mountainous. Körner et al. (2017) also covered mountains on islands.

Line 152: "undeRstanding"

Line 156: You mean the general aspect and orientation of the mountain, right? This may differ quite strongly from the local aspect values at 1-km resolution or even finer spatial resolutions. Indeed, a mountain that is globally oriented in north-south direction (cf. the Andes) can still harbour locally a lot of north-facing and south-facing slopes that will dictate local climate velocities. Same for mountain globally oriented in an east-west direction (cf. the Pyrenees). There, it is possible to also find east-facing and west-facing slopes affecting local climate velocities. I don't see how this analysis on the general orientation of a given mountain range can inform on climate velocities given that local slopes may face the sun from so many different directions within the focal mountain range and thus affect the climate velocity.

Line 157: At which spatial resolution did you consider aspect here? Sorry, this is not clear how you investigated different aspects here. By the way, note that elongation (the term you used in the Methods section) and aspect are two different things, right? To me aspect refers to the direction towards which a given slope is facing. That is different from the way you defined elongation based on

ellipses in the Methods, right?

Lines 177-179: See my general comment on the potential lack of data on species elevational range shifts (cf. Supplementary Table 2) in comparison with what exists (also prior to 2017) in the scientific literature (see the BioShifts database that is available for free and which can be downloaded from figshare: Comte et al. 2020). By missing important data on species elevational range shifts from the scientific literature, your findings are likely not representative of the reality. So please, consider extracting species elevational range shifts from the BioShifts database to run your analyses relating the vertical velocity of isotherm shifts against observations of species elevational range shifts.

Line 180: "species uplifts" sounds a bit strange, no? I see the parallel you want to do with mountain uplift but I am not sure it applies to species...

Lines 179-183: Well, excepted for three outliers (2 in France and one in Italy), the overall observed relationship in Extended Data Fig. 6 is pretty close to the 1:1 relationship, no? Did you run a linear model to assess the coefficient estimate of the slope parameter? If yes, does it significantly deviate from 1 (not from 0)?

Lines 184-185: See also findings from reference #9 regarding elevational range shifts. Conclusions are very consistent.

Line 186: What do you mean by "a taxon"? Do you mean at the species level? Or do you refer to a given taxonomic group? Sorry, but this is not clear what is meant here by "taxon" in terms of the taxonomic resolution you used in your analyses.

Lines 186-188: What about phylogenetic signal in the residuals of your models? Did you assess the phylogenetic signal in your model residuals? If there is any phylogenetic signal left, you may have to consider using PGLS approaches to account for that. Not also that sample size effect could be accounted for directly by adding sample size a covariate in your models. In fact, given the methodological differences among the different studies from which you extracted data on elevational range shifts, you should consider using a mixed-effects modelling approach to such potential biases (see reference #9).

Line 193: What do you mean by "percentage of numbers"?

Line 197: Why using the mean value of shifts instead of the raw values (species by species) as depicted in Extended Data Fig. 9? I don't understand how you computed the mean value of shifts here. Is it the mean across different taxa belonging to the same taxonomic group or across different regions and periods from the same taxon? Could you please clarify this point in the Methods section?

Lines 210-212: Well, you did not really test that, right? To do so, you need to compare how much variation you can explain in the observed velocity of species elevational range shifts when using (1) your estimate of the vertical velocity of isotherm shifts based on MARL versus when using (2) the vertical velocity of isotherm shifts based on a constant ARL or based on local ALRs following the approach used in reference #9. If you do not test that, you cannot really conclude on your estimate of the velocity of isotherm shifts being more accurate, right?

Lines 221-224: Indeed, and this is especially more pronounced if you use a very coarse spatial resolution of 0.5° (or worst 1°) which is about 55 km by 55 km at the equator (or worst about 111 km by 111 km at the equator).

Lines 234 vs. 303: First you mention a spatial resolution of 0.5° (cf. CRU data) which is about 55 km by 55 km at the equator and then you mention a spatial resolution of 1°, which is around 111 km by 111 km resolution at the equator (very coarse). So, in the end, what is the spatial resolution you used to run your analyses? Sorry to insist on that bit it is not clear from the main text and the Methods section. It is actually very confusing given the different information provided in the different sections of the manuscript.

Line 234: Here you mentioned that temperature data were averaged over every 5 years, but throughout which time period? Did you make several five-year intervals starting from 1971 and up to 2015? Sorry, this is unclear.

Line 235: I am a bit lost here. Did you use gridded data (0.5° resolution) or the raw point data from the network of weather stations as you mentioned here? Please clarify.

Line 237: What is the grid cell size here? Is it 0.5° (55 km by 55 km at the equator) resolution?

Line 247: So did you get the values of the parameter "e" (water vapour pressure) in a gridded format (0.5° resolution) from the CRU dataset? Are these values of water vapour pressure provided every month or every year (monthly mean or annual mean values)? Sorry, but I miss important information to understand what you did exactly.

Line 252: What is the h parameter? Is it altitude or elevation? These are not the same thing.

Line 255: "climate velocity" is repeated two times.

Line 255: What does "bySpark" mean? Is it a model, a method? Do you have a reference to cite for that? Sorry, this is unclear.

Line 258: What is the spatial resolution that you used here? Did you use the native SRTM spatial resolution of 90 m by 90 m?

Line 261: What is "Wolfram Mathematica 9"? Is it a software? Do you have a reference to cite for that?

Line 269: Again, at which spatial resolution?

Line 277: Again, see Körner et al. (2017) for a better source of "expert-identified" mountains.

Line 286: Sorry, but looking at the content of Supplementary Table 2, it does not really look like exhaustive. Please, consider using data from the BioShifts database (Comte et al. 2020) which provides raw range shift estimates at the species level. See also the most recent meta-analyses on elevational range shifts (Rumpf et al. 2019; Mamantov et al. 2021) to improve the level of exhaustiveness in light of the most recent scientific literature on the topic.

Lines 290-292: Which one did you keep in such situations? Why not considering all the data that is available? This is part of the variation observed in the raw data. Why removing this variation?

Lines 294-295: See the BioShifts database (Comte et al. 2020) to retrieve polygons of the study regions for which raw range shift data have been extracted. Maybe this could be a useful source of information for your work.

Line 319: Why running such a bootstrap approach to balance sample size? Why not using all data available and then use the information on sample size per study as a covariate in the models, following the approach used in reference #9?

Lines 330-331: Again, such methodological bias can be directly accounted for in a mixed-effects modelling approach (see reference #9). No need to subsample the original raw data. Besides, sample size is not the only methodological difference among the different studies used here to extract data on species elevational range shifts (cf. Supplementary Table 2). Such methodological differences among studies must be accounted for explicitly either through mixed-effect models using the raw range shift data (see reference #9) or using a dedicated meta-analysis approach to compute the pooled effect size (see the `metagen()` function from the `meta` package in R) (see Mamantov et al. 2021).

Line 332: How many is n here? How did you set the value of the sample size n ? Sorry, it is not clear here how n is chosen.

Line 334: Please consider using raw elevational range shift data from the BioShifts database (Comte et al. 2020) as some of these studies for which you have only summary statistics might be included in the BioShifts database with their raw data.

Lines 340-341: What about iterations exceeding or lagging behind the mean expected velocity under the assumption of full synchrony in species responses? Did you run unilateral or bilateral Wilcoxon signed-rank tests for the comparison against the expected theoretical mean?

Lines 345-349: Here, you provide more information on the sample size issue and you seem to run a sensitivity analysis. Yet, the outputs in Supplementary Table 3 do not really allow to conclude on the influence of sample size on the main results given that the sensitivity analysis could only be run for a small set of studies (about 10 out of 47). Sorry, but Supplementary Table 3 does not really convince me on the absence of effect regarding sample size. Besides, and like I said earlier, there is no need to subsample the original data. It is better to keep all the raw data available and use an appropriate statistical tool like mixed-effect models or a true meta-analytical framework.

Line 360: What do you mean by "which is going to be described later"??? Do you mean in another paper?

Lines 364-366: I don't see these results in the Figures, the Extended Data Figures or in the Supplementary Figures. Could you please provide the histograms of the residuals that you are suggesting here?

I sincerely hope that my comments will be useful.

Best,

Jonathan Lenoir

Suggested references

Angelo & Daehler (2013). Upward expansion of fire -adapted grasses along a warming tropical elevation gradient. *Ecography*, 36: 551-559

Bodin et al. (2013) Shifts of forest species along an elevational gradient in Southeast France: climate change or stand maturation? *Journal of Vegetation Science*, 24: 269-283

Brown et al. (2016) Ecological and methodological drivers of species' distribution and phenology responses to climate change. *Global Change Biology*, 22: 1548-1560

Brusca et al. (2013). Dramatic response to climate change in the Southwest: Robert Whittaker's Arizona Mountain plant transect revisited. *Ecology and Evolution*, 3: 3307-3319

Comte et al. (2020) BioShifts: a global geodatabase of climate-induced species redistribution over land and sea. figshare. Dataset. <https://doi.org/10.6084/m9.figshare.7413365.v1>

Cordova et al. (2016) Near-Surface Air Temperature Lapse Rate Over Complex Terrain in the Southern Ecuadorian Andes: Implications for Temperature Mapping. *Arctic, Antarctic and Alpine Research*, 48: 673-684

Dainese et al. (2017) Human disturbance and upward expansion of plants in a warming climate. *Nature Climate Change*, 7: 577-580

Frei et al. (2010). Plant species' range shifts in mountainous areas—all uphill from here? *Botanical Helvetica*, 120: 117-128

Geppert et al. (2020) Consistent population declines but idiosyncratic range shifts in Alpine orchids under global change. *Nature Communications*, 11: 5835

Guo et al. (2018) Land-use change interacts with climate to determine elevational species redistribution. *Nature Communications*, 9: 1315

Kirchner et al. (2013) Altitudinal temperature lapse rates in an Alpine valley: trends and the influence of season and weather patterns. *International Journal of Climatology*, 33: 539-555

Koide et al. (2017). An upward elevation shift of native and non-native vascular plants over 40 years on the island of Hawai'i. *Journal of Vegetation Science*, 28: 939-950

Körner et al. (2017) A global inventory of mountains for bio-geographical applications. *Alpine Botany*, 127: 1-15

Kuhn et al. (2016) Early signs of range disjunction of submountainous plant species: an unexplored consequence of future and contemporary climate changes. *Global Change Biology*, 22: 2094-2105

Mamantov et al. (2021) Climate-driven range shifts of montane species vary with elevation. *Global Ecology and Biogeography*, <https://doi.org/10.1111/geb.13246>

Ploquin et al. (2014). Bumblebee community homogenization after uphill shifts in montane areas of northern Spain. *Oecologia*, 173: 1649-1660

Rolland (2003) Spatial and Seasonal Variations of Air Temperature Lapse Rates in Alpine Regions. *Journal of Climate*, 16: 1032-1046

Rowe et al. (2015). Spatially heterogeneous impact of climate change on small mammals of montane California. *Proceedings of The Royal Society B Biological Sciences*, 282: 20141857.

Rumpf et al. (2019) Elevational rear edges shifted at least as much as leading edges over the last

century. *Global Ecology and Biogeography*, 28: 533-543

Scherrer & Körner (2010) Infra-red thermometry of alpine landscapes challenges climatic warming projections. *Global Change Biology*, 16: 2602-2613

Author Rebuttals to Initial Comments:

Responses to the reviewers' comments

Wei-Ping Chan, Jonathan Lenoir, Guan-Shuo Mai, Hung-Chi Kuo, I-Ching Chen, and Sheng-Feng Shen

Referee #1 (Remarks to the Author)

As the global climate warms, mean temperature isotherms will move uphill in mountain regions, and it is important to attempt to quantify this process – particularly the rate at which this is occurring. The rate of movement (in a particular mountain range) will depend on the amount of regional warming, coupled with the lapse rate (the rate of decrease of mean temperature with elevation). This paper attempts to calculate shifting elevation limits of isotherms for mountain regions worldwide, which is a worthwhile endeavour, but in my view is not 100% successful in doing this. The main assumption, critical to the whole paper, is that the mean lapse rate in a mountain range can simply be determined using mean temperature and atmospheric moisture content alone – i.e. by deriving the moist adiabatic lapse rate or MALR. The MALR is a theoretical construct which applies to the free atmosphere and tells us how fast the temperature should decrease with elevation within a cloud when latent heat of condensation is released. At higher temperatures more moisture can be released which results in a shallower lapse rate. Lapse rates of mean temperatures on the ground surface however (relevant for migration of species) are not the same as the theoretical free-air MALR for many reasons. The first is that the Earth's surface does not behave the same as the free atmosphere, the mountain slopes being subject to local heating (in the sun for example) and cooling (e.g. ponding of cold air near the ground at night). The second is that the MALR only applies when there is condensation or cloud formation (i.e. not all the time). In dry conditions or when clouds form well above the mountain summits, by day the DALR should apply (which is much steeper), and at night the lapse rate often reverses due to cold air drainage and other local scale processes. The end product is that the surface-based lapse rate of mean temperature is nowhere near the MALR in many cases, and varies substantially in mountains worldwide. Because the lapse rate used in this paper is unrepresentative, so will be the calculated climate velocity and therefore the comparison with species movements may also be flawed.

Authors' response 1 to Referee #1:

We agree with reviewer #1 that the moist adiabatic temperature lapse rate (MALRT) may not represent what the organisms are actually experiencing in terms of the true adiabatic lapse rate of temperature as expressed by the actual distribution of their habitats along elevation gradients. Here, reviewer #1 somehow suggests to use an adiabatic lapse rate of temperature that would incorporate microclimatic processes near the ground or as experienced by organisms in their habitats. Albeit we fully agree that such a thing would be the ideal information to use, we are far from being able to achieve such a level of precision requiring very high-resolution data at the

global extent. Yet, before we can reach this level of detail in the spatial distribution of microclimatic processes, we believe that it is worth using the MALRT rather than a simple and constant adiabatic lapse rate of temperature (cf. constant ALRT) throughout the planet, which is more or less what people are doing right now in ecology when trying to assess the link between the velocity of species range shifts and the velocity of isotherm shifts in mountains. Hence, we do argue that the MALRT represents an important step forward compared to using a constant ALRT (see our new comparisons with the constant ALRT as a kind of baseline or reference for what people are using so far). In this revised version of our work, we also now use another and totally independent method to calculate the local ALRT that is the slope between local elevation gradients and land surface temperature derived from satellite observations: the satellite-based adiabatic lapse rate of temperature (SALRT). We found that the SALRT is consistent with the MALRT (see L120-121 & Supplementary Results). In addition, the MALRT explains the variation in species range shifts better than the SALRT, so we consider the MALRT as a useful metric of lapse rate for understanding species range shifts in mountain ecosystems (see L182-183 & Supplementary Results).

As well as the theoretical problems, the way the MALR is calculated in this study is also arguable. The dataset CRU TS 3.24 is not well explained. I think it comes from the reference below (I searched online) because it is not referenced in the paper (a major omission). It is critical to cite the original reference and acknowledge the datasets fully. Harris, I., Jones, P.D., Osborn, T.J. and Lister, D.H. (2014), Updated high-resolution grids of monthly climatic observations – the CRU TS3.10 Dataset. *Int. J. Climatol.*, 34: 623–642. doi: 10.1002/joc.3711 https://crudata.uea.ac.uk/cru/data/hrg/cru_ts_3.24.01/Release_Notes_CRU_TS3.24.01.txt. It is stated that mean temperature and vapor pressure from 2011-2015 (a short five year period) are used to calculate lapse rates, but surely as temperatures change over time, so could the lapse rate. The 5 year moisture (and temperature?) values are combined with a 35 year “rate of warming” to calculate climate velocity. Would not 35 year mean lapse rates be more accurate? 5 years is not really long enough to eradicate short term variability. It is also the case that since the MALR depends on temperature, it will increase (become steeper) with elevation (i.e. the actual MALR profile at anyone time is not even linear) but in this paper only one lapse rate appears to be calculated for each pixel based on 5 year means (at least I think so, but it is not clearly explained).

Authors' response 2 to Referee #1:

In this revised version of our work, we have completely updated the dataset to CRU TS4.05, as suggested by reviewer #1, and added the correct citation data. In addition, our MALRT calculations now use the 2011-2020 data for comparison with the satellite data (cf. the SALRT).

Then our temperature change data are used for the difference between 1971-1980 and 2011-2020. We now make the methodological description clearer (see L291-L298).

There is next to no information about the data (i.e. stations) that go into the CRU dataset – so there is no guarantee that interpolated temperature and moisture values are realistic, particularly in mountain regions which are the focus of this study – some discussion of this issue at the very least (and quantification of possible error or uncertainty in lapse rate estimation) would be expected in an article in this journal. Finally, this whole method assumes that the lapse rate does not change in a warmer climate, and that the climate velocity can be obtained for a given location if the lapse rate is known (i.e. all isotherms are moving uphill at the same rate). The situation whereby lapse rates could steepen (isotherms converging closer together on a slope) or weaken (isotherms spreading further apart) is not acknowledged. Yet we know that elevation dependent warming is common in the literature (reference 10). This is another way of saying lapse rates may change in a warmer world and the spacing between isotherms may change. The 0degC isotherm may shift by a different amount in comparison with the 10degC isotherm, and therefore climate velocity in a particular mountain range at a particular location cannot be summarised with one rate. It may depend on the critical temperature being discussed. I am wondering whether the derived MALR could at least be compared with the observed lapse rate in the vicinity of the pixel in the CRU dataset (for the same 5 year period) – this would tell us whether using the theoretical MALR is anywhere near the actual mean lapse rate recorded on the mountain surface – at least in the historical data.

Authors' response 3 to Referee #1:

We do agree with reviewer #1 that the moist adiabatic lapse of temperature (MALRT) may change over time, but we decided to first focus on mapping the spatial distribution of the mean MALRT during a 10-yr period before trying to assess its change over time which would be another story for another project. Yet, we now discuss the implications of the MALRT potentially changing over time (see L192-195). As for the comparison of the MALRT with observed lapse rate in the vicinity of the pixel in the CRU dataset, we decided to use high-resolution satellite data collected near the mountain surface to determine the actual surface air temperature lapse rate (SALRT) from satellite observations during the same 10-yr period of 2011-2020. We compared our findings based on the SALRT with the MALRT. Our results indicated that SALRT exhibited more variability compared to MALRT, but MALRT was the most significant factor in explaining species range shifts. These findings suggest that the MALRT has a stronger explanatory power to capture the observed variation in species range shifts than the SALRT derived from satellite data. Overall, our research demonstrates that the MALRT calculation, based on the laws of thermodynamics, provides a better proxy than the SARLT calculation for explaining past changes in the range shifts of mountain organisms.

Therefore, we propose that the MALRT and corresponding vertical climate velocity calculation are the most effective indicators of vertical climate velocity globally for studying the effects of climate change on mountain organisms (Fig. 2 and Extended Data Fig. 7; Supplementary Results).

So overall, I find that there are quite a lot of conceptual issues raised by this paper, and I do not think the authors have considered these limitations or addressed them. I am not an ecologist and therefore I confine myself to the climate aspects and the derivation of climate velocity estimates, but these are central to the paper's findings since it is against this that the species shifts are compared. I cannot therefore recommend publication in its current form, especially in a high-profile publication like Nature.

Authors' response 4 to Referee #1:

We thank reviewer #1 for their helpful comments and suggestions. We have reanalyzed and improved the paper according to the reviewer's suggestions and we do understand the reviewer's concerns from the climate science point of view, as it seems the reviewer is not an ecologist but a climatologist. However, reviewer #1 needs to be aware that ecologists so far are using a much cruder estimate of the adiabatic lapse rate of temperature (they actually use a constant in most cases) than what we are proposing here. Hence, our estimate of the MALRT, although still not perfect, is a huge step forward for ecologists interested in the understanding of the drivers behind species range shifts in mountain systems. Of course, the dream data for ecologists would be to have microclimatic maps to compute a more relevant adiabatic lapse rate of temperature as experienced by organism in their habitats but we are still far away from having such an information, especially so because microclimate as perceived by living organisms in their habitats is almost species-specific and that all weather stations worldwide follow a standard that do not allow to capture such microclimatic processes near the ground.

My more specific comments below refer to line numbers

Lines 61-63. This surely incorrect and you divide the surface temperature change by the lapse rate (not what you state) – see line 90 where you state the opposite.

Chan et al.: Yes, indeed. That was a typo. Thank you for pointing it out, the correct information was indeed written on the original line 90. We have now revised our statement and corrected the typo mistake (see L285-302).

Line 74: the MALR is always lower (not often) – and it is not -6.5degC/km . This figure is a global mean lapse rate with no physical basis.

Chan et al.: True and to accommodate this comment from reviewer #1, we have completely rewritten the paragraph (L69-77).

Line 82: This statement is vague.... Does this mean 2.5degC/km in late summer but 7.5degC/km in spring – or both values in both seasons etc...

Chan et al.: This whole section of the main text has been completely rewritten (see L83-L95) to account for the reviewer's comment and provide a clearer and more concise explanation.

Line 97: What is a "mountain grid"? – just one grid point.... You normally calculate a lapse rate between locations (so you would need more than one grid point). This needs clarification.

Chan et al.: Sorry if this was unclear or confusing. We have reformulated this part of the text to accommodate to this comment from reviewer #1. To clarify, we have incorporated two different approaches: (1) one based on elevational transects to derive the SALRT, as recommended by reviewer #1, and (2) the other, namely the MALRT, based on the laws of thermodynamics. Additionally, we have included a substantial amount of detailed description (see L227-302) to provide clearer and more comprehensive information.

Line 102: The rate of surface temperature change and the mean temperatures/vapour pressures are from different periods... this does not seem appropriate

Chan et al.: Thank you for your comment and sorry for the confusion. We have now adjusted our sampling periods according to your earlier suggestions to be more consistent. Both the SARLT and MARLT calculations adopt averages based on data from the same 10-yr average period 2011-2020. Then, for calculating climate velocity, we divided the grid layer of the rate of mean temperature change, in °C/yr, between the period 2011-2020 and the period 1971-1980 by the grid layer of the SARLT or MARLT, in °C/km, thus generating a spatial grid of the vertical velocity of isotherm shifts in km/year.

Line 116: I would refer to lapse rates as shallow or steep (not high and low which is ambiguous).

Chan et al.: As per the reviewer's suggestion, we have made efforts to minimize the use of "high" or "low" terminologies to avoid potential confusion. However, we have chosen to maintain the use of "high" or "low" rather than switching to terms like "shallow" or "steep" When referring to lapse rates. This decision is based on the fact that "high" or "low" are commonly used for lapse rates in ecological studies and when discussing the magnitude or velocity of range shifts, providing a more familiar language for researchers in the field. This said, we are ready to switch to the “shallow” vs. “steep” terminoly when referring to lapse rates if the editors deem it necessary.

Line 144: Yes, but your method assumes that the lapse rate does not vary and so specifically outlaws elevation-dependent warming (see my earlier comment)

Chan et al.: As we wrote in our response to the reviewer's general comments, we acknowledge the validity of considering variation over time as well as variation in microclimatic processes near the ground, which would be ideal for a comprehensive analysis of the potential changes in adiabatic lapse rates as perceived by living organisms. However, achieving such a high level of precision at a global extent requires spatial information at very high resolution which still remains a challenge. In addition to that, we also agree that assessing temporal changes in the MARLT is needed but it first requires that we are able to compute a static, but spatially-explicit, version of the MARLT. While we aspire to attain this level of detail in the spatial distribution of microclimatic processes near the ground and to capture temporal changes in the MALRT, we

believe it is valuable to first compute and use a temporally static, but spatially-explicit, version of the MALRT in the meantime instead of relying on a simple and constant, in space and time, adiabatic lapse rate of temperature (referred to as the constant ALRT) across the entire planet, which is what most ecologists are doing so far. This distinction is particularly important in ecological studies that aim to assess the relationship between species range shift velocities and isotherm shift velocities in mountainous regions, where the use of constant ALRT is more prevalent.

Line 148: Since temperature decreases with elevation, your MALR will increase with elevation, so climate velocity will decrease with elevation (assuming the same amount of warming). Surely this is in part intrinsically built into your assumptions, so it is not surprising you get this result. What is more surprising is the accelerated climate velocity with elevation in the tropics.

Chan et al.: Indeed, our results are calculated in accordance with thermodynamic principles, so the decrease in climate velocity with elevation is expected and thus not surprising to scholars studying atmospheric science. However, it is this negative relationship that has been overlooked by ecologists in the past and that we hope ecologists will understand and use for future studies on biodiversity redistribution as climate is changing. In addition, our results in the tropics suggest that not only temperature but also water vapor and other factors may affect the vertical velocity of isotherm shifts. Therefore, we believe that our analysis is still valuable, chiefly so for ecologists but also for other research fields, including for climatologists, interested in using spatially explicit maps of the MALRT and the vertical velocity of isotherm shifts along mountain slopes.

Line 158 ff: It is unclear why you combine NW/SE and SW/NE aspects for example. Some more explanation is needed. Opposing aspects will be the most different in terms of climate regimes and microclimates so why combine them into one category? This will have the effect of damping the differences between categories because you are missing main windward/lee contrasts which occur within the categories not between them.

Chan et al.: Thank you for your comment. We have excluded the mentioned analyses and results from the current version of the paper and therefore this comment does not apply anymore.

Line 164 ff: The main result of the topographic analysis (Figure 4) is that water vapor appears to be higher on E-Equator mountains. Many of the other differences are insignificant. This result I think is because upslope winds come from the SW (in the northern hemisphere) and NW (in the southern hemisphere) and these directions align with the prevailing moisture tracks in mid-latitudes, so the orographic enhancement effect is maximised where the ranges directly face the moisture source.

Chan et al.: Thank you for your comment. We have excluded the mentioned analyses and results from the current version of the paper.

Line 207: Many ecological studies rely on lapse rates from sounding balloons? Do they? Which? References need to be given here. An observed free air lapse rate is probably better than the theoretical MALR, although both are problematic.

Chan et al.: Thank you for pointing this out. We have removed the specific statement from our manuscript. Instead, we have included a new paragraph (see L74-81) to further elaborate on our point and provide a more comprehensive explanation.

Line 221. Yes, this is a catch all sentence which sums up why the MALR is of limited use, and why we need more monitoring on the ground (i.e. the last sentence at line 227-229 which I agree with).

Chan et al.: Thank you for your comment. We retained a similar statement in the revised version of the main text (see L213-216).

Line 234-236 – Both temperature and vapor pressure were “averaged across coarse spatial extent”? What does this mean? This is vague but also concerning.

Chan et al.: Thank you for pointing this out. We have included a more detailed description (see L285-302) to provide a clearer view of our methodology.

Line 238: What are T and T² at a grid point – surely a grid point is just a point and only has one value. This relates to my confusion about how a temperature lapse rate can be derived from a grid point. This whole explanation is unclear.

Chan et al.: Here, T denotes air temperature (K), and T² is the square of T. The lapse rate of an air parcel can be derived according to the laws of thermodynamics. In our calculations, we treated each grid cell as an air parcel, allowing us to derive the lapse rate. We have now rewritten the text to clarify this point and we hope it is now clearer.

Line 254: This implies averaging from monthly to annual values – which again hides seasonal variability. The assumptions behind this are never discussed.

Chan et al.: Thank you for pointing that out. We have added two paragraphs (L83-95 and L192-195) to explain the variation in spatial and temporal scales. We understand that using annual averages for this analysis is a simplification, but we believe that this simplification is appropriate for global scale analyses, as analyzing on a monthly basis would make the results overly complicated. Moreover, our use of annual averages is consistent with the typical time scales used in the biological range shift database, allowing for a more targeted analysis.

Line 270: Diagrams or maps would make the explanation of elongation and orientation clearer, and how they relate to aspect. You never define how aspect is calculated. It sounds as though it is applied to a subset of elongated (Sierra-like) mountain ranges.

Chan et al.: Thank you for the suggestion. We have excluded the mentioned analyses and results from the current version of the paper. This comment does not apply anymore.

Line 303: At this point you say a grid is 1 x 1 degree resolution... which implies it is multiple cells (the actual data is 0.5 degree resolution – line 234?). So for the first time it appears a grid is more than one cell or pixel (but this is unclear).

Chan et al.: The analyses of the biological data have been revised to reflect the updated version of the biological dataset we used and which relies on the BioShifts database. For specific details, please refer to L350-370 in the main text of the manuscript, where you will find a comprehensive description of the updates.

Line 312: So the studies may be from different time periods, but the lapse rates and climate velocity are always from 2011-2015. This seems like a limitation.

Chan et al.: We agree that this is a limitation. However, our analysis shows that even if we don't calculate the climate velocity for the same time period as each empirical biological study, we can still explain to certain extent the biological shift rate, and our explanatory power is much improved over previous calculations of climate velocity using a single temperature lapse rate (see Figure 4, Supplementary Results, Supplementary Figures 3-7). Therefore, we believe that our research offers significant progress in understanding the impact of mountainous climate change on the redistribution of living organisms. We are confident that future research can utilize our method to carry out further calculations to improve the spatiotemporal resolution of adiabatic lapse rates as perceived by living organisms.

Line 352 ff: This last section is incomprehensible to me, but I admit I am not an ecologist – I did try to read it several times, but I think English hampers the explanation.

Chan et al.: We have completely rewritten the methodology section to also clarify this specific section of the analyses. Please refer to L350-398 in the manuscript for a comprehensive and enhanced description of our methodology.

General point about figures: it is confusing that there are three levels of figures: main figures, extended data figures, and supplementary figures. This may be the style of the journal but it is so

confusing for the reader. I am not convinced that all 17 figures are needed.

Extended data Figure 2: Why do some climate velocities have error bars but not others?

Chan et al.: To address the issue of extremely small error bars in some records, we have made revisions to Fig. 3d. Specifically, we have adjusted the size of the data points to be smaller and made the error bars more visually prominent.

Extended data Figure 3: Not needed

Chan et al.: Thank you, and we removed it as suggested.

Extended data Figure 6: I get the feeling that this could be important as some sort of validation attempt, but given the current labelling I find it difficult to interpret. One interpretation appears to be that using MALR is not that different from using a standard rate of 5.5degC/km (yellow symbols) apart from some outliers which does not make sense to me.

Chan et al.: Thank you for your comment. This figure is removed from the current version. Instead, the comparison between MALRT and SALRT has been addressed in multiple parts throughout the manuscript, including in the most relevant sections of the main text (see L117-126, L128-140, L722-748) as well as through the relevant figures (see Figures 1-3).

Referee #2 (Remarks to the Author)

The manuscript by Chan et al. is, in some ways, a re-analysis and update of the meta-analysis published by Chen et al. (Science 2011), focusing explicitly on elevational shifts in organisms in response to temperature, rather than both elevational and latitudinal shifts. The current manuscript starts with a quandary: Chen et al. 2011 reported surprisingly little concordance between predicted elevational shifts in regions (based off of warming temperatures) and observed elevational shifts. Chan et al., here, go on to try to fix this issue by re-calculating expectations (based off the moist adiabatic lapse rate) of shifts, and then testing these new expectations against the previous meta-analytic data of empirical range shifts. The vast majority

of this present manuscript is not about ecology, per se, but about variation in moist adiabatic lapse rates (and, by proxy, elevational climate velocities) around the world and in differing conditions. Nevertheless, the manuscript concludes that Chen et al. (2011) were overly-pessimistic about elevational tracking, and that, with updated expectations, more species appear to be tracking than previously thought. The most interesting finding is that elevational tracking appears to occur more in areas with lower thermal velocities, implying that high climatic velocities inhibits the ability of species to track climate.

Authors' response 1 to Referee #2:

Indeed, reviewer #2 is right, albeit the first part of our study strongly focuses on the physical science basis of isotherm shifts using a calculation based on the moist adiabatic lapse rate, one of the key findings and main messages of our study involves important ecological implications with elevational tracking of species range shifts being more likely in areas with lower thermal velocities, implying that high climatic velocities inhibit the ability of species to track climate, as stated by reviewer #2. This is an important conclusion and outcome for our understanding of biodiversity redistribution as climate warms globally.

Greater tracking at lower climate velocity is very intriguing, for the implication stated above, but unfortunately cause is very poorly connected to effect in this case. In particular, low climate velocity is coincident with a wide variety of other things, each of which could also impact the degree of (real or perceived) climatic tracking, therefore leaving the interpretation ambiguous. For example, as the manuscript describes, higher velocity occurs in more arid regions, where precipitation is likely to play a relatively greater role as a mechanism. This study only looks at average thermal niche tracking, and ignores all other aspects of the climate (e.g., seasonality, extremes, and non-thermal variables), many of which now have well demonstrated mechanistic links impacting range dynamics over time. One variable of importance for species is precipitation, which has been implicated as being critical for structuring range shifts for many species, from trees to mammals. Using an example dataset from this very meta-analysis, Tingley et al (2012) demonstrated that elevational range shifts were heterogeneous, and that the direction of range shifts (which could be calculated as the degree of climatic tracking) was equally influenced by precipitation as temperature. Yet in the present manuscript and meta-analysis, only thermal tracking is evaluated. This is a problematic limitation because, as stated above, the manuscript finds that tracking fails in areas of high climatic velocity, which are also described as being arid areas. Well, arid areas are exactly the areas where we would most expect species to track precipitation – perhaps even more so than temperature. So is the failure of species to track temperature in areas of high thermal velocity due to the fact that the thermal velocity is high, or is it because species aren't tracking temperature at all in those areas? The inability of this study

to differentiate between these two well-established ecological hypotheses is a major limitation, and ultimately means that I don't feel the present manuscript adds much new to the already expansive literature on elevational range shifts and their trends.

Authors' response 2 to Referee #2:

We would like to thank reviewer #2 for their time to assess our work and to provide meaningful feedback on the relevance of precipitation changes when analyzing the observed variation in species elevational range shifts. We know that Tingley et al. (2012) analyzed the relationship between changes in bird ranges and changes in both temperature and precipitation at different elevations in the California region over the past 90 years. Uniquely, California is one of the few places in the world where animal and plant distributions as well as temperature and precipitation data are recorded simultaneously at different elevations and where changes in precipitation patterns clearly drive the direction and magnitude of species range shifts (cf. Crimmins et al. 2011). We fully agree with that, and we are aware that temperature alone cannot explain everything. Our study, however, was designed to specifically improve current estimates of the actual rate of temperature change in mountain systems by incorporating, in a spatially explicit manner, an important dimension that was missing so far in all velocity metrics of climate change, namely the adiabatic lapse rate (cf. the vertical dimension of isotherm shifts). Without this key information, former estimates of the relationship between the rate of temperature change and range shifts of organisms in mountainous areas could not be accurate. Of course, we could do even better by also incorporating an adiabatic lapse rate for precipitation patterns along elevational gradients in our computation of the vertical velocity of climate change but that is a much more challenging endeavor than refining isotherm shifts and we first need to apply our methodological workflow on the temperature gradient before moving on to the precipitation gradient. Here, we really try to provide a better estimate of the actual velocity of temperature change in mountain systems worldwide to test whether this added information of the vertical dimension in isotherm shifts does improve the relationship with empirical species range shifts. Yet, we do not expect to explain all the observed variation just by making that correction. This kind of holy grail would indeed require to use a multidimensional assessment of climate change by also incorporating the velocity of isohyet (similar to isotherms but for precipitation) shifts. But this clearly goes beyond the scope of our study as good data on precipitation patterns are needed along the elevational gradient. This said, we agree that we need to better discuss, which

we try to do in this revised version, other important drivers of change, such as isohyet shifts, that can capture part of the unexplained variation that is left once we have accounted for the adiabatic lapse rate of temperature in mountains worldwide.

Line Edits:

53. Is it unclear? I don't know any study that suggests conclusively that species are keeping pace – whereas many studies suggest that species are lagging.

Chan et al.: Thank you for the comment. As suggested, the statement has been rewritten as “Whether species are closely tracking the rate of climate warming is chiefly assessed by comparing the velocity of species range shift with the velocity of climate change, i.e., the rate at which isotherms move through the geographic space” (L46-48)

60. “No opportunity” seems to be an incorrect interpretation of the data and their findings.

Chan et al.: Thank you for this comment. As suggested, the statement has been rewritten as “As our range shift analysis shows, species are unlikely to track isotherms quickly enough to match the high velocities at which isotherms are moving along some elevation gradients.” (L200-202)

67. Many studies correct for variation in actual lapse rates; for example, Elsen et al. 2017. Saying that actual lapse rate variation hasn't been considered in ecological research is an overstatement.

Chan et al.: Thank you for this comment and for suggesting the relevant work from Elsen et al. 2017. We added a paragraph to better acknowledge those earlier studies correcting for variation in actual lapse rates (L83-95).

142. I found the statement about the importance of climate adaptation strategies to be a throw-away line that lacked the specificity necessary to be convincing. Climate adaptation is a huge and growing area, and some strategies are more effective than others. Just saying developing strategies is “important” is overly vague.

Chan et al.: We agree with reviewer #2 that our former statement was overly vague. We have expanded the discussion regarding this aspect by incorporating additional sentences (see L200-212) to provide a more comprehensive and detailed explanation.

145. This study here suggests that it has been hard to study warming in mountainous regions due to poor climatic sampling of mountains regions. This is true, but I don't understand how this study escapes from that same limitation. The estimates of both lapse rates and climatic velocities assumes the validity and accuracy of the changing temperature over time in mountain regions. It seems like the current trends in climatic velocities could easily just be artifacts of poor modeling of mountain temperatures in the CRU data. How do we know that the failure to track high climatic velocity results isn't an artefact of poor CRU data in those particular regions?

Chan et al.: Indeed, we are not improving the spatial resolution of climatic velocities in mountain regions compared with former studies but we do add an important component which is the vertical dimension of isotherm shifts that is, hitherto, missing from all studies using climate velocity maps. In our opinion, this is a step forward that justifies our work and its novelty. We acknowledge that this study may not fully overcome the challenge of limited climatic sampling in mountain regions because we are still relying on the same climatic data as former studies mapping climate velocities worldwide but we are here adding the vertical dimension of mountains into the picture, which is a meaningful contribution that remains to be done. This said, the inclusion of the newly added approach of SALRT, based on satellite measurements and following suggestions and recommendation from reviewer #1 and reviewer #4, may provide improved sampling in these areas. Additionally, we emphasize the novelty of calculating both SALRT and MALRT, as well as considering the influence of water vapor. These aspects contribute to the unique and innovative aspects of our research.

154. The relationship described in this section (on slope and aspect) is interesting and the results are largely expected, but I also imagine that these results are entirely scale-dependent (i.e., dependent on the spatial scale of the unit of analysis). The manuscript only analyzes these trends at a single scale, which may not be a scale that is representative of biodiversity mechanisms.

Chan et al.: Thank you for your interest and the comment, but we have made the decision to exclude the mentioned analyses and results from the current version based on the scope and focus of the study.

Referee #3 (Remarks to the Author)

A. Summary of the key results

The authors show larger variations in climate change velocity across the world's mountains than previously acknowledged, which they attribute to over-simplifications in the way previous models were built. They also cross-analyse the literature on species' ability to track climate change along mountain slopes, both in continents and on islands, concluding that many species not moving fast enough – which implies a major and underestimated threat to mountain diversity worldwide.

Authors' response 1 to Referee #3:

Yes, but see our new analyses and updated results since this review was first performed.

B. Originality and significance: if not novel, please include reference

The models are new as well as the region-specific estimates for climate change velocity. These estimates are likely to play an important role in climate change research, in particular for mountainous regions since they have been challenging to model properly due to a variety of reasons (natural and artefactual). Although the authors promote the main significance of the work for biodiversity science, their models and data are likely to have important impacts on other areas of knowledge and practice, such as sustainable agriculture, climate change mitigation to human settlements and livelihoods, the spread of diseases, water security, etc. To me the most interesting result was the one presented in Fig. 2D, which I think will have a particular impact in the way that those regions may be targeted for further fine-scale climate change research and monitoring.

Authors' response 2 to Referee #3:

We wish to thank reviewer #3 for their positive comments on our work and its broad significance and relevance beyond biodiversity science. We totally agree with reviewer #3 that our findings have important impacts on other areas of knowledge and practice than just ecology, such as sustainable agriculture, climate change mitigation to human settlements and livelihoods, the spread of diseases, water security, etc. In this new and updated version of our work, we preserved similar maps which can now be found in Fig. 3c and Fig. 4a.

C. Data & methodology: validity of approach, quality of data, quality of presentation

The text is easy to follow and compelling, explaining complex terms and formulae in accessible ways even for a non-climatologist. The figures are quite good, although some labels and information could be made clearer to read (eg the use of a black background, and colours against a grey map, are somewhat challenging although I am not sure how much better they could be).

I am not able to assess the validity of the approach used to derive the climate models.

The choice of biological datasets is well explained, but I was somewhat disappointed that not more studies were available for the comparative (meta) analysis. As a result, the results were heavily biased towards certain parts of the world (e.g., 17 out of 47 studies were from the UK, and very few from eg Latin America or Africa). I would have imagined (but do not know by fact) more suitable datasets to be available, considering vast international collaborations such as GLORIA (Global Observation Research Initiative in Alpine Environments), ITEX (International Tundra Experiment), and large data providers such as sPlot.

Authors' response 3 to Referee #3:

Thank you for these helpful suggestions. To include more biological records, we have updated our biological dataset by using the BioShifts dataset (see Lenoir *et al.*, 2020 & Comte *et al.* 2020 for a free access to the dataset on figshare).

The definition of 'mountains' (l. 87) does not differentiate flat areas above 1000m (eg, high plateaus) from those that are indeed areas of varying relief, which are the ones most relevant for biodiversity shifts along slopes. See e.g. the proposal in <https://link.springer.com/article/10.1007/s00035-016-0182-6> . It may be adequate to evaluate whether the findings of this paper are robust to different mountain classifications.

Chan *et al.*: Our previous version of the description may be a bit misleading. Our mountain regions are now based on the GMBA mountain inventory V1.2

(<https://www.earthenv.org/mountains>), which was an updated version of the article mentioned by reviewer #3 (Körner et al. 2017; GMBA Inventory v1.0).

Reference:

Snethlage, M.A., Geschke, J., Spehn, E.M., Ranipeta, A., Yoccoz, N.G., Körner, Ch., Jetz, W., Fischer, M. & Urbach, D. A hierarchical inventory of the world's mountains for global comparative mountain science. *Nature Scientific Data*. <https://doi.org/10.1038/s41597-022-01256-y> (2022).

l. 108: it would be good to clearly define 'mountain areas' – are individual cells, or mountain ranges / individual mountains what the authors refer to? This has biological implications: if a whole range is under high climate velocity, species may struggle to keep track of their climatic niche, but if there are individual cells of low velocity within a mountain range that on average has high velocity, this could mean that those particular cells are especially important for providing rescue to species distributed across the range (a point the authors may want to make).

Chan et al.: This is a really good point. In our approach, mountain areas were adapted from expert's definitions, which were originally shapefiles or spatial polygons of mountain ranges or individual mountains (see also our response above). These vector maps or spatial polygons were later overlapped on to a raster map at 0.5-degree resolution for data extraction, so that spatial polygons were rasterized at a 0.5 degree resolution. Regarding the interest of within-mountain refuge (i.e., the areas with localized isotherms shifting downward and "against the flow" within a mountain range where the general trend is isotherms shifting upward), our results can cover part of it, but perhaps not thoroughly enough to make a strong statement as this would require a much more refine spatial resolution to incorporate topoclimate and ultimately microclimatic processes which are still lacking from our global maps. For example, we can observe those 'within-mountain refuge' cells in the Andes and Himalaya mountain ranges (Fig. 1b & d) but we would

need topoclimatic data at a much more refined spatial resolution to actually map the spatial heterogeneity in topoclimate that is available locally, within that cell. Plotting all those mountain ranges individually was inefficient. So instead, we provided the raw data of the maps we generated for people who are interested in further exploration of the dataset. Readily applicable scripts are also provided for reapplying our method on mountain ranges having topoclimatic data at finer spatial resolutions.

l. 109: the use of ‘biodiversity hotspots’ should be defined, as it is used in different ways in the literature. Do the authors refer to Myers’ original definition and subsequent updates?

Chan et al.: We have removed this part of the analysis in order to make the paper more focused.

l. 199-200: it would be good to expand here, in quantitative and/or qualitative terms, on the result that “mountain species are generally lagging behind the climate velocities” – the reader is now referred to two figures but this is such an important result that a brief summary seems well placed here.

Chan et al.: We would like to thank reviewer #3 for suggesting to expand a bit on this very important point on time lag dynamics in biotic responses to climate change. We have rewritten our paper and hope this part is now clearer and reads better, as follows: “Comparing the vertical velocity of isotherm shifts based on MALRT with the observed rate of species range shifts, the probability that a given taxonomic unit tracks the vertical velocity of isotherm movement decreases sharply with increasing velocities of isotherm shifts (Fig. 4f-g). Thus, we found that species appear to track climate change only at lower velocities along the elevational gradient, irrespective of the taxonomic group (Fig. 4g, Extended Data Figs. 3d-e and Extended Data Fig. 4). These results reveal the potentially catastrophic impacts of rapid climate change on mountain biodiversity.” (see L186-192)

l. 227: many papers end with similar statements of what should be ‘priority areas for conservation’, but in truth we know this has little value unless a much more solid recommendation can be provided. As mentioned above, perhaps the single low-velocity cells in the 24 mountain regions with overall high climate velocity might be a more tangible focus for conservationists, provided that they still contain natural habitats and biodiversity enough to allow such rescue. High-resolution maps and data from this study should be provided in formats amenable for use by those truly taking these findings into account in spatial conservation prioritisation.

Chan et al.: We have rewritten our discussion to address this valuable suggestion and we wish to thank reviewer # 3 for highlighting this important point. We also provide high resolution maps and access to the raw data behind those maps for readers to use and explore.

D. Appropriate use of statistics and treatment of uncertainties

I do not feel qualified to properly assess the choice of statistical methods, but my impression is that the authors have considered and incorporated several sources of uncertainties in their analyses and when reporting specific results. There should be further assessment of uncertainties related to the biodiversity datasets, or at least a discussion on the potential impact of biases (see also below).

Authors' response 4 to Referee #3:

We thank reviewer #3 for their honest comments and for their precious time to review our work. We have improved the discussion section in this revised version of our work to better highlight the potential pitfalls as well as the fact that temperature alone cannot explain all the observed variation in species range shifts (see L208-210). Yet, we do believe that our study makes a valid point on the relevance of refined velocity metrics for mountain systems globally.

E. Conclusions: robustness, validity, reliability

The authors mention in some places the lack of data availability for weather stations, but there is little discussion on the potential impact of the relatively scarce biodiversity data on the precision and accuracy of the analyses, and the conclusions derived from them. This is by no means the authors' fault – as this is a well-known and general problem – but it would be good to discuss further the potential effect of data biases (eg taxonomic and spatial) in biodiversity data. For instance, are the results for Italy and France in extended figure 6 true outliers, or related to denser data for those countries? It may also be helpful to plot the locations from Sup Table 2 onto a global map.

Authors' response 5 to Referee #3:

We wish to thank reviewer #3 for pointing this out. In this revised version of our work, we have used the BioShifts dataset (Lenoir *et al.*, 2020). This dataset has significantly increased the sample size of the biological component we were using in the former version of our work. This

said, geographical and taxonomic biases still exist in the BioShifts dataset (see Lenoir et al. 2020 for a discussion on taxonomic, geographical and methodological biases related to species range shifts data). We have added to the Discussion the uncertainty caused by the scarcity of biological data in some geographical regions or for some taxonomic groups and the fact that the available data are mainly concentrated in North America and Europe, as suggested, which is another imprint of Europe's history and colonialism.

The biodiversity conclusions in the abstract (l. 38-42) are very general and unquantified – ‘more cases’... ‘generally lagging’... I think they should be more precise and concrete. The last lines in the abstract (42-45) make some strong claims in conclusion, but I could not find where in the data or analyses those claims were directly derived.

Chan et al.: We have completely modified the abstract according to the new findings. We hope that this new version is more accurate and less general.

F. Suggested improvements: experiments, data for possible revision

The biological implications are rather thin – which seems a bit odd, since the main interest and implications of these findings will most probably be on the effects of climate change precisely on biodiversity. However, I think several interesting aspects could be further explored with the data at hand. For instance, what are the taxonomic (eg orders, families) and functional (eg trees/herbs) groups / mostly affected (i.e., not tracking climate change)? Are those groups/species phylogenetically constrained or randomly spread across the tree of life? Are there any particular biomes or biogeographical regions that are mostly affected? And so on – so that the reader really can *learn* more about potentially losers (and perhaps winners) of climate change in mountains. Fig. 4e covers only very large groups, and the figure is a bit too crowded to really extract useful information. Right now, there are not as many general learnings derived from the study as there could certainly be.

Authors' response 6 to Referee #3:

We are now using the BioShifts dataset to provide a more complete analysis of the relationship between the velocity of isotherm shifts in mountains globally, after incorporating the adiabatic lapse rate in a spatially explicit manner, and the velocity of species range shifts. We also analyze additional factors, including climatic, topographic, geographic, and environmental factors, to explain species range shifts (see L684-691). However, the discussion of taxonomic variation may

be beyond the scope of the current paper to discuss in detail and already and partially covered in Lenoir et al. (2020), so we have not included it in our revised manuscript.

G. References: appropriate credit to previous work?

There is a vast field of research in this and related fields and the study does a good job in referencing what I perceive as some of the most important studies. I understand there are word limitations, but some potentially additional research of relevance would be:

- Work by Kenneth Feeley, Miami (there one publication cited)
- Work by Christian Körner, Basel
- <https://science.sciencemag.org/content/334/6056/660>

Authors' response 7 to Referee #3:

We are thankful to reviewer #3 for suggesting references to former work from Kenneth Feeley and Christian Körner. We did our best to incorporate this work in the revised version of our work.

H. Clarity and context: lucidity of abstract/summary, appropriateness of abstract, introduction and conclusions

Although the text is well written and flows well, there are a few minor language improvements/typos that could be improved (e.g. line 55 'if they were TO track climatic changes...'; l. 222 delete 'in'; l. 327 'taken cared here'). In particular the Methods session could profit from careful external editing for style.

Similarly, there are some instances of unnecessary repetition (e.g. line 67 repeating line 62).

Please ensure that all tools etc are properly described and referenced upon first use (eg pySpark, l. 255, R in l. 350), as well as abbreviations (eg CRU).

Signed by
Alexandre Antonelli, 17 Feb 2021

Authors' response 8 to Referee #3:

We have now significantly improved the quality of the main text which we hope is now flowing better than during the previous round of review. We wish to thank again reviewer #3 for his useful, insightful and most important, constructive, comments and suggestions. We hope that we have addressed most, if not all, of the reviewer's comments in this revision.

Referee #4 (Remarks to the Author)

Mountain ranges provide important refugia for species in times of rapid global change. In order to find a favourable climate, on mountains, organisms just have to travel few kilometres while along latitudinal gradients a comparable change in climate would demand travelling several thousands of kilometres. This makes mountain ranges particularly relevant for the conservation of biodiversity over the next decades of climate change. But how fast is the climate changing with elevation in global mountain areas and are mountain species able to track it? These are the important questions the authors deal with in their paper "Climate velocities and lagged species elevational sifts in mountain ranges". Chan et al., first, quantify the MALR globally based on thermal dynamic theory and data on temperature and water vapor. Second, the authors quantify, again globally, the velocity of climatic change (using data on ground temperature over the last decades and estimated for the MALR) and identify regions showing high speed of change. Third, they use data on temporal changes in biodiversity (several data sets sampled around the world from the literature) to test if species can track their climatic niches and if this differs between mountains showing high or low speed of climate velocity. They find strong global variation in the MALR and in the velocity of climate change on mountains which can better explain a lag of climate tracking by species than previous data sets.

I think that the topic of the paper is of interest to readers of several disciplines as it addresses the consequences of climate change and produces a new global data set on the rate of climate change in mountain areas. To my best knowledge, this is something really missing in the literature.

However, as I point out below I am currently (based on the data presented in the paper) not convinced about the analysis conducted by the authors. My main point is that the authors, as far as I can see, do not present any validation of their model predictions for the MALR and for the predictions on climatic changes along elevation gradients over time. I think this is important as the variation in MALR and the variation in climate velocity is presented as a key finding and touches all subsequent analyses.

Authors' response 1 to Referee #4:

We thank reviewer #4 for their positive comments on the relevance of our work and for highlighting its novelty. We agree with reviewer #4 that a validation step of the adiabatic lapse rate we computed from thermodynamic theory, namely the MALRT, was clearly missing and we did our best when revising this ms, to address this important point. To address this comment, we have used satellite data of surface temperature to validate the predictions generated, from first

principles, by the more theory-driven MALRT. For more details and information on this key point, see also our *Authors' responses 1 & 3 to reviewer #1*.

Please find below some detailed critic:

- The authors use a very “rough” definition for mountains. They simply define mountains as all areas >1000 m asl. Classical definitions of mountains additionally or exclusively consider the steepness of the area. This has considerable consequences: For example, in Chan et al. the steep Andean eastern slopes below 1000 m are not considered as mountains while large parts of southern Africa are considered as mountains even though the area is quite flat (it is simply just above 1000 m). I do not know if this has any influence on the results. Might be that this is just a minor critical point but I am quite sure that people working mountain ecology, geography or climatology will see this critical. I would here suggest to recalculate the key results additionally for other mountain definitions and put this into the supplements, e.g. for the standard definitions by Körner et al. 2016, Kapos et al. 2000

Authors' response 2 to Referee #4:

We agree with reviewer #4 that it was a limitation in the former version of our work. We now provide a much better definition of mountain globally by relying on existing data, using the most updated layers delineating mountain ranges globally (Snethlage et al. 2022). Please see also our *Authors' response 3 to reviewer #3*.

Reference:

Snethlage, M.A., Geschke, J., Spehn, E.M., Ranipeta, A., Yoccoz, N.G., Körner, Ch., Jetz, W., Fischer, M. & Urbach, D. A hierarchical inventory of the world's mountains for global comparative mountain science. *Nature Scientific Data*. <https://doi.org/10.1038/s41597-022-01256-y> (2022).

- The MALR and climate velocity data is calculated from a model based on terrain surface temperature and vapor pressure of the CRU TS 3.24 data set. However, the authors do not present any validation of their predictions using true climate station data from mountains. What is the error in the prediction? Even though the authors write that climate station data is rare for mountain areas there should be some data sets (e.g. Appelhans et al. 2016 report a lapse rate of 5.5°C per km elevation for the southern Kilimanjaro and not a rate of about 3.5 as suggested by the model). I think that a check of the accuracy of the predictions is extremely important as the variation in MALR and consequent changes in climate velocity is presented as one of the major findings of the paper. All the results are depending on the accuracy of this prediction. It is therefore at this point hard to judge the value of paper if this information is missing.

Authors' response 3 to Referee #4:

We fully agree with reviewer #4 and we wish to thank reviewer #4 for his/her helpful comments and suggestions on the matter. First, we now provide in the introduction section of the main text a more detailed overview of the existing knowledge on the adiabatic lapse rate as reported in several regions worldwide, including the Himalayas (see L83-88 in the introduction section, L123-L126 and Fig. 1b & d). Additionally, we have updated the dataset to CRU TS 4.05 which provides more accurate estimates of the MALRT. We also used a totally new empirical data source from satellite observations to independently validate our MALRT calculations. Please see also our *Authors' responses 1 & 3 to reviewer #1* above.

- The CRU data sets offer a resolution of 0.5° so ca. 50 x 50 km. If I think of isolated mountains this may cover both the rainy and the dry slopes of mountains. So, if moisture plays a role, the model will predict for both sides of the mountain the same MALR but I guess that the species data sets used for testing range shifts in the paper were often sampled on a specific side of the mountain which could lead to potential errors in the predicted change. Could that be a problem?

Authors' response 4 to Referee #4:

We thank reviewer #4 for pointing that out. Indeed, spatial resolution matters for the climatic layers we used to derive MARLT and velocity maps at sufficiently high spatial resolution. Yet, this argument also holds for the biological data as reported in the BioShifts database which we now use to relate the velocity of isotherm shifts after incorporating MALRT to the velocity of species range shifts. In fact, species range shift data are also very crude and even more crude in terms of the spatial resolution than the climatic data we used to derive the MALRT and the velocity maps. Most studies report range shifts within a given region that can be quite large and several hundreds of kilometers in spatial resolution. Hence, the most refined spatial resolution might not be the one from the MARLT layer as suggested here by the reviewer. To mitigate this bias that is in fact likely due to the rather coarse spatial resolution in the BioShifts dataset, we included a bootstrap procedure in our analysis (see Extended Data Fig. 3a as well as L371-385).

We also found that when the direction of range shifts is not distinguished, sampling the leading or trailing edge can affect the predictability of environmental variables to some extent (see Extended Data Figure 7c). However, after separating the analysis depending on whether organisms are moving up or down, sampling the leading or trailing edge does not affect the results of climate influence on the velocity of species range shifts (see Extended Data Fig. 7c).

- CRU TS 3.24 was used for deriving temperature and vapor pressure data: I checked the CRU download area and saw that version 3.24 was withdrawn due to errors. Did you really use this version and not the corrected version 3.24.01? I see that there are always some delays from analysis to the final paper but I still wondered why not the more recent versions were here used (CRU TS 4.x). Additionally, I guess that there is more than one climate model than CRU TS which could be used here. It should be justified why the authors used this climate data set and not others. Generally, as the CRU TS data set is the base of all analyses I would introduce it with 2-3 sentences in the main text.

Authors' response 5 to Referee #4:

We thank reviewer #4 for this suggestion, and we have now updated the dataset to CRU TS 4.05, as suggested by the reviewer. So, we are now using the last and most updated version of the CRU dataset.

- 16 of the 41 data sets (39%) which were used for testing the relationship between elevational shifting rates of species and elevational climate velocity are from the UK, which are not identified as a mountain region in the paper (as far as I can see there is not a single pixel of mountain area indicated here...using the definition of areas > 1000 m). This should be aligned (either use another definitions of mountains, or exclude it from the analysis, or justify the use of the data)

Authors' response 5 to Referee #4:

In this revised version of our work, we now use the recently published BioShifts dataset (Lenoir *et al.*, 2020), which is a more complete database of species range shifts. We also updated our description of the mountain definition, which is based on these "expertly identified" mountains (GMBA mountain inventory V1.2) (see *Authors' response 3 to Referee #3* above).

- It does not become clear how much better the new predictions of climate velocity are in comparison to the older estimates for predicting the observed shifting rate. The authors write that

a general lag of upslope migrations observed in the past literature can be explained by wrong estimations of climate change on mountains but they now also found a quite general lag of upslope migrations.... I would here recommend to add some analysis how much better the new climate velocity data fits to the range shifts compared to the old data. Additionally, I find the observation of a general lag in species range shifts on mountains (which is not so much observed along latitudinal gradients) still really interesting – particularly given that the dispersal distances for tracking climate are some orders of magnitude smaller than along latitudinal gradients. I saw that in an earlier Science paper by some members of the author team this finding is discussed but I would advice to discuss this also briefly in this paper.

Authors' response 6 to Referee #4:

In this revised version, we have assessed and compared the relationship between empirical estimates of the velocity of species range shifts along elevational gradients and a series of different metrics incorporating the vertical velocity of isotherm shifts along elevational gradients by relying either on spatially explicit adiabatic lapse rates such as the MALRT and SALRT, or a constant value of ALRT (see Fig. 2 and Extended Data Fig. 4). By doing so, we found that the vertical velocity of isotherm shifts based on the MALRT better explains the velocity of species range shifts along elevational gradients.

Minor comments:

234-237: It did not fully become clear to me if the authors just used the TS3.24 data set or if they did some additional analyses using one weather station records here. If the former is the case, I would simply add: “In the TS3.24 data set, both mean annual temperature.....” to make this clear.

Chan et al.: We have modified the text as suggested in L263-L265.

It is really difficult to understand how the authors calculated the probability of species tracking the climate velocity. In particular from the main text but also from the methods: Here it is stated that “First of all, we used the bootstrap technique to subsample the dataset to control the inconsistencies induced by having different sample sizes across studies. For each taxon in each region, we set the sample size to n and drew n records (n in Fig. 4a).” But what is a record? I recognized that two different kinds of data sources were used here but even for detailed species

data it is unclear to me what a record is: a species? Or the single observation of a species on a study site/elevation? I think that this could be described more precisely so that it is easier for the readers to follow what the authors tested here.

Chan et al.: We have completely rewritten that part of the manuscript which should now be better explained. We hope that we have clarified that point with the new and revised version of the main text (see L372-398).

I would also advice to briefly describe the results based on the older approach of directly predicting mean change in species elevations to the mean change in the shift of temperature along the elevation gradient as this approach is more direct and easier to understand.

Chan et al.: Our Fig. 4f used the older approach mentioned by the reviewer and we briefly described this result in L186-188 as suggested.

To sum up, I think that the paper has potential but it remains unclear how much the climate predictions (MALR and climate velocity) reflect reality.

Chan et al.: For the analyses on how different velocity metrics for isotherm shifts relate to species range shift, we had to completely rewrite the method section because we now use the BioShifts dataset (Lenoir *et al.*, 2020). Hopefully, the method is better explained and clearer now.

Referee #5 (Remarks to the Author)

General comments

A. Summary of the key results

Using a formula to compute moist adiabatic lapse rate (MALR), instead of using a constant adiabatic lapse rate (ALR) of ca. 5.5°C to 6°C per km of elevation, the authors aim at: (1) refining the velocity at which isotherms are shifting vertically along the elevational gradient in mountain regions worldwide and (2) assess whether this new estimate of climate velocity better

explains the velocity at which species are shifting upslope as climate warms. Deriving MARL, at 0.5° resolution (ca. 55 km by 55 km at the equator) across all mountain regions on Earth, the authors found that MARL ranges between 3 to 9°C/km (vertically). Then, the authors computed, for each spatial grid cell of 3025 km², the temporal change in temperature conditions between 1971-1975 and 2011-2015 and divided this value by the amount of years over this time period to generate a temporal gradient in °C/yr. Finally, the authors divided the “local” MARL value (°C/km) of the focal grid cell by the “local” temporal gradient (°C/yr) of the same grid cell, which gives the vertical velocity of isotherms along elevational gradients in km/yr or in m/yr (when multiplying by 1000). Using MARL, instead of a constant ALR value, the authors found that the vertical velocity of isotherms along elevational gradients ranges between -16.67 m/yr to 16.8 m/yr, averaging 5.42 m/yr, which is slightly more than the average vertical velocity found with a constant ALR value (ca. 4.56 to 4.98 m/yr) (Extended Data Fig. 1). When relating the velocity of isotherm shifts with biodiversity data (velocity of range shifts), the authors found that the probability for species to lag behind climate warming is higher when the vertical velocity of isotherms is higher.

Authors' response 1 to Referee #5:

Indeed, the summary from reviewer #5 fits our main message and key findings which are still valid in this revised version of our work.

B. Originality and significance

This is a very interesting study and the novelty of the authors' study clearly lies on the use of the MARL to compute the vertical velocity of isotherm shifts under climate change. Indeed, by doing so, the authors genuinely account for the spatial variation in water vapour pressure on the value of the adiabatic lapse rate (ALR). Yet, I do have four major and important concerns (listed below throughout the different sections of my general comments) that I think the authors should consider carefully to improve the quality of their work.

First of all, considering novelty statements, the authors are wrong when claiming that the use of “local” or spatially variable and spatially-explicit ALR has not been explicitly considered in ecological research (see the authors' statement in lines 65-67). In fact, a very recent study (Lenoir et al. 2020: reference #9 in the authors' reference list) did also derive local and spatially-explicit ALRs before relating the local vertical velocity of isotherm shifts along a given elevational gradient to the observed velocity of species range shifts along the focal elevational gradient. However, Lenoir et al. (2020) did not account for the effect of varying water vapour

pressure on local ALRs, which is what I would consider novel in the authors' study. Hence, I suggest the authors to better emphasize what is the main novelty in their study (cf. accounting for variation in water vapour pressure) and to make a direct comparison with the approach used in Lenoir et al. (2020) to compute local ALRs. By doing so, it will be clearer for the reader what is the main novelty in this study. Besides this issue of better justifying novelty in light of the most recent scientific literature (cf. reference #9), I would also recommend the authors to carefully check the most recent meta-analyses on elevational range shifts: Guo et al. 2018; Rumpf et al. 2019; and Mamantov et al. 2021.

Authors' response 2 to Referee #5:

We have revised our statement to better acknowledge previous work (see L43-63).

C. Data & methodology: validity of approach, quality of data, quality of presentation

About the validity of the overall approach, it seems that the authors used temperature data at a very coarse spatial resolution (CRU data at 0.5° which is about 55 km by 55 km at the equator) to compute the vertical velocity of isotherm shifts along elevational gradients. This is my second major concern given how fast temperature are changing across 1 km distances in mountain regions (cf. topoclimatic variation) (Scherrer & Körner 2010). Because of that, the authors may underestimate the availability of local escapes for species redistribution in mountain regions (Scherrer & Körner 2010) and thus overestimate the velocity at which isotherms are actually shifting upslope as climate warms. Given the authors' main focus on the most recent period of climate warming (1971-2015) (see extended Data Figure 8), I would recommend the authors to use finer global climatic grids such as TerraClimate (<http://www.climatologylab.org/terraclimate.html>), WorldClim (<https://www.worldclim.org/>) or CHELSA (<https://chelsa-climate.org/>) to compute local ALRs using the same approach as in Lenoir et al. (2020) (see the subsection entitled "Climate velocity" in the Methods section) for comparison purposes with the "local" MALR the authors derived.

Authors' response 3 to Referee #5:

We chose CRU as our main dataset because: "for the climatological means, CRU slightly outperforms CHELSA, WorldClim shows a slightly worse performance compared to the former two" (Karger *et al.*, 2017). To compensate for the coarser spatial resolution of the MALRT we computed here, we use the higher resolution MODIS land surface temperature data (MOD11C3) in this version to compute a satellite-based adiabatic lapse rate of temperature (SALRT) for comparative purposes with the MALRT we computed first.

My third major concern is about data quality and the representativeness of the biodiversity data that the authors used for assessing the magnitude of observed species range shifts (cf. Supplementary Table 2). Indeed, looking more closely at the content of Supplementary Table 2, I am afraid the authors are missing quite a lot of important data on species elevational range shifts, including data from within their literature search period (up to 2017) (e.g. Angela & Daehler 2013; Bodin et al. 2013; Brusca et al. 2013; Dainese et al. 2017; Frei et al. 2010; Koide et al. 2017; Kuhn et al. 2016; Ploquin et al. 2014; Rowe et al. 2015) as well as some of the most recent studies (e.g. Geppert et al. 2020). Most of these data are now freely available throughout the BioShifts database (Comte et al. 2020). Hence, I suggest the authors to at least download the BioShifts database to get a more comprehensive set of raw data on species range shifts along the elevational gradient. It is rather important that the authors make sure to use the most updated dataset on species elevational range shifts. As it is now, the picture depicted by the authors in their main findings might be biased and far from representative of the current knowledge on species elevational range shifts (see also the most recent meta-analyses on species elevational range shifts: Guo et al. 2018; Rumpf et al. 2019; Mamantov et al. 2021).

Authors' response 4 to Referee #5:

We are now using the BioShifts dataset (Lenoir *et al.*, 2020) in this version, as suggested by reviewer #5 and under the guidance of reviewer #5.

D. Appropriate use of statistics and treatment of uncertainties

Finally, my fourth and last major concern relates to the statistical analyses the authors used to link the magnitude of observed shifting rate for plants and animals distributed along elevational gradients against the velocity at which isotherms are shifting vertically as climate changes (cf. Fig. 4). Indeed, except for balancing sample size among studies, the authors did not really correct for methodological differences among the studies from which they extracted data on species range shifts. Yet, it has been clearly demonstrated in the scientific literature that methodological biases can account for a very substantial variation in the data (Brown et al. 2016 and reference #9). Such methodological differences among studies must be accounted for either by running mixed-effect models on the raw range shift data or by running a dedicated meta-analysis to compute the pooled effect size. None of these methods or approaches have been used by the authors here, which is rather problematic. For instance, using the “metagen()” function from the “meta” package in R and specifying the method argument to the Sidik-Jonkman method, the contribution of a given study to pooled effect size is weighted by sample size and the degree of variation in the study’s data, such that a given study with many species range shift values and little variation in range shifts across species has a stronger influence on the value of the statistic than a study with few species and a high level of range shift variation. Alternatively, the authors could use mixed-effect models and add study ID as a random intercept (at the very least) in their models when relating the vertical velocity of isotherm shifts to observed shifting rate across several taxonomic groups. Also, the authors should consider to add taxonomic information as a random term in their models (see reference #9 for a similar approach) to account for potential

phylogenetic signal in the residuals of their models. Hence, there is no need to subsample the data to balance sample size, as the authors did when computing the probability of species tracking climate velocity, one just needs to use an appropriate statistical tool: either a true meta-analysis approach (e.g. Mamantov et al. 2021) or a mixed-effect modelling approach on the raw data (e.g. reference #9).

Authors' response 5 to Referee #5:

We included the methodological and taxonomic differences as variables in predicting species range shifts in our random forest analysis (see Supplementary Results and Extended Figure 7). Sampling methods and taxonomic differences do not have significant effects. MALRT still has the most important effect on the velocity of species range shifts.

E. Conclusions: robustness, validity, reliability

Given the four major concerns I have listed above and highlighted in parts B, C and D, I am afraid that the robustness, validity and reliability of the authors' main findings (and thus conclusions) are questionable in their current state, thus requiring more work to achieve a greater level of robustness, validity and reliability (see suggested improvements below and in my specific comments).

Authors' response 6 to Referee #5:

Thank you for your constructive comments and suggestions. In this new version we have properly addressed all four main concerns raised here by reviewer #5. Please see our more detailed responses in our *Authors' responses 1-5 to this reviewer*.

F. Suggested improvements: experiments, data for possible revision

As already mentioned in each of my four major concerns, suggested improvements include: (1) to better emphasize the novelty of the authors' work that lies on the use of an ALR metric that account for spatial variation in water vapour pressure whereas former work deriving local ALR did not account for the effect of water vapour pressure; (2) to derive local ALRs, based on spatially fine resolution climatic grids from WorldClim or CHELSA, following the method already used in reference #9 and for comparison purposes with the approach proposed by the authors (MARL); (3) to improve the representativeness of the data on species elevational range shifts by downloading the BioShifts database and querying raw data on species elevational range shifts; and (4) use more appropriate statistical tools, such as mixed-effect models or a true meta-

analytical framework, to better account for methodological differences among studies.

Authors' response 7 to Referee #5:

We thank reviewer #5 for highlighting these four major concerns that we each addressed in our revised version. We now believe that all of the reviewer's concerns have been properly addressed and incorporated into this version. Please see our more detailed responses in our *Authors' responses 1-5* to this reviewer.

G. References: appropriate credit to previous work?

Overall, the authors are providing appropriate credit to previous work, but see my suggestions in part A as well as the list of references I am providing at the end of my review and after my specific comments to the authors.

Authors' response 8 to Referee #5:

We now refer to the appropriate scientific literature following the reviewer's suggestions.

H. Clarity and context

Overall, the manuscript is well written but sometimes the methods are not clearly described and quite difficult to follow. Some information are not clearly provided in the main text or the Methods section but are hidden in the captions of the figure, such as the way the authors computed the temporal gradient in temperature changes. I have provided several suggestions in my specific comments to the authors to improve clarity in the main text and the Methods section.

Authors' response 9 to Referee #5:

We have improved the clarity of our paper by completely rewriting it.

Specific comments

Chan et al.: Given the numerous and constructive suggestions provided by reviewer #5 (Jonathan Lenoir), we have decided to invite reviewer #5 to join our research team in order to fully address his questions and suggestions. We have been closely collaborating to revise the entire analysis and article during the past six months, and due to the extent of the modifications, with reviewer

#5's agreement, we only responded to his major comments listed above. We hope that the editorial team at nature will agree with that decision and otherwise, if the editorial team deems it necessary, we are ready to provide a point-by-point response also for reviewer #5's comments

Reviewer Reports on the First Revision:

Referees' comments:

Referee #1 (Remarks to the Author):

I can see that the authors have made substantial revisions in response to my previous comments, including new analyses. The use of a surface-based lapse rate as well as the theoretical moist adiabatic lapse rate, as well as standardising the periods used for much of the data, and use of a better temperature dataset, are all welcome additions. As a result, I think the paper is now much more rigorous. There are still a few issues though remaining which I think need to be discussed. First, the terminology is still somewhat confusing (and sometimes wrong) and the interpretation could be much clearer. I know that the authors are not originally speaking to the climatological/meteorological community, but at present you can tell that they are not climatologists and since this is an inter-disciplinary journal it is important that all communities can read and understand the methods and terms used.

When talking about lapse rates (particularly lines 65-95), terms are still sometimes used incorrectly. The word adiabatic refers to the lack of exchange of energy between a rising parcel and its surroundings. This therefore leads to the theoretical specific dry adiabatic lapse rate (DALR) or a saturated/moist one (MALR) when condensation occurs. It is a specific case. The word cannot be used to describe any climatological lapse rate found on a mountain slope so the abbreviation ALRT should really just be LRT. The ALRT is not defined as the observed temperature difference along a slope (line 68) since the actual lapse rate observed is usually NOT adiabatic. The mountain slope exchanges energy with its surroundings and the adiabatic assumption is violated (it is purely a theoretical assumption/model).

The DALRT (9.8°C/km) is not the mean observed rate in the troposphere (as claimed at line 71-72), and only occurs when there is no condensation (or energy exchange). The mean observed rate is usually around 5.5 or 6.5°C/km depending on who you talk to.

The other issue is that lapse rates on the ground (slope based) are usually referred to in the context of gradients in air temperature (measured at 2 m above ground level) – which is again different from the free air (or vertical) lapse rates measured above a single point (measured by a weather balloon). This difference needs to be made clear. Here you are estimating a slope based lapse rate because you are wanting to estimate temperature changes along a mountain slope. The DALRT and MALRT are theoretical free air values in the absence/presence of moisture (so can be estimated for a point as done in this paper) but they are not really mountain slope rates.

The SALRT (should be SLRT because there is no such thing as a surface adiabatic lapse rate) as derived here is an attempt to measure the slope based lapse rate based on observed surface temperatures (at ground level not at 2 m) and so this will be different from the required 2 m air temperature lapse rate which the authors really want. Because the MODIS data measures the actual surface temperature (at ground level) it is influenced by microscale surface properties (such as albedo, emissivity, rock type, vegetation amount etc) and so there is a LOT of noise in the calculated lapse rate. It also is biased to cloud free conditions (which enhances this spatial variation) because when there is cloud there is no data. This needs to be explained/acknowledged and is probably why SALRT does not show such good relationships with the ecological data. Much of this noise would need to be edited out in order to make it representative of more than a very specific microclimate. Despite this, I do think it useful to use this lapse rate as well, because there is an ecological argument often made that organisms experience the microclimate near the ground, and so this is what should be more relevant to them. The fact that it appears not to be (!) is an interesting result.

At present none of this is explained and discussing this in more detail would also help explain some of the findings.

I also think that the use of the terms high and low for lapse rate is just too ambiguous (even if

ecologists use them as claimed). The climatological community defines a lapse rate as negative (since temperature decreases with height) – so the DALR would be $-9.8^{\circ}\text{C}/\text{km}$... This is a low absolute value (but a high number: is $-9.8^{\circ}\text{C}/\text{km}$ higher than $-6.5^{\circ}\text{C}/\text{km}$?) so confusing. To make it even more confusing the meteorological community sometimes defines the lapse rate with a positive number (e.g. $9.8^{\circ}\text{C}/\text{km}$). Thus without clarifying what high and low mean you are causing lots of potential confusion to different communities who will interpret this differently.

I strongly suggest you cannot use high and low in this context... you must use something like steep or rapid (meaning a fast change) or shallow/weak (meaning a less rapid change) and explain your terminology in the paper. Since in this paper the change is always one way (it always gets colder as you go up?) then steep and shallow I think make sense to most people.

More specific points

Line 67: adiabatic used incorrectly

Line 70: incorrect: temperature does not influence the dry adiabatic lapse rate

Line 71: incorrect: the DALRT is not the mean observed rate

Line 72: all not most?

Line 77: a constant rate cannot be "about" 5.5°C (it is that or not!)

Line 119/120: surely these must be the other way round... the SLRT shows more variation than MALRT and must go from -5.14 to $8.45^{\circ}\text{C}/\text{km}$?

Line 123: The negative rate (SLRT?) in northern mountains suggests the presence of persistent temperature inversions (with cold air stuck at the ground at lower elevations) – is this correct?

Line 138: I am a bit confused about how 32% of the area can have exceeded the 80th percentile? You would immediately think 20% because that is by definition. Is this because it can be exceeded with either of the methods? The fact that it is not 40% means there is some overlap between the methods? In any case I am not sure you should use the SLRT isotherm shifts to identify the most endangered locations since some of the SLRT isotherm shifts are unreliable because the lapse rate is near zero (see my comments later about figure 2).

Line 166: Isotherm velocities are higher at lower elevations (and so weaker in high mountains). This is despite EDW and the enhanced warming in high mountains? So does the change in lapse rate overcompensate for any enhanced warming? i.e. it would be quite nice to be able to separate the effects of warming rate (EDW) from changing lapse rates, and for this to be explained... because it seems like a contradiction?

Line 172: Why is the latitudinal effect reversed? Explain.. (is this expected or not)? I am left thinking whether this is due to positive and negative latitudes in the two hemispheres and the scales being inverted?

Line 194: The "impacts resulting from MALRT" – what does this mean?

Line 256: So the SALRT for a pixel is some sort of average of multiple transects? How many transects in each cell? Is there a relationship between variability in the SALRT and the number of transects? i.e. is it less extreme (more reliable?) when there are more transects contributing to it (this is what I would expect)?

Line 300: I thought $6.5^{\circ}\text{C}/\text{km}$ was the most commonly assumed lapse rate?

Line 312: Why 20% or 10% - do not follow – are there two classifications?

Figure 2: It is hard to see the detail here but it generally seems that isotherm shifts are larger in higher latitudes, because of both the enhanced rate of warming (arctic amplification) and the weaker lapse rate. There is a discontinuity however as the lapse rate gets shallower – if we have a very shallow (but +ve lapse rate) we end up with the isotherms moving uphill extremely rapidly. As soon as the lapse rate becomes negative the isotherms move downhill extremely rapidly. This cannot be correct in reality (for a very small change in lapse rate -say from $+0.1^{\circ}\text{C}/\text{km}$ to $-0.1^{\circ}\text{C}/\text{km}$) the situation completely reverses. Thus in the SLRT map you have areas of very rapid change in isotherms in both directions right next to each other?. This is an artefact of the methodology and needs mentioning (i.e. the method is not reliable once the mean lapse rate gets near zero). I think for that reason I would block out any rapid shifts (in both directions) which are caused in this situation. Are

any of these rapid shifts used to identify the critical regions in figure 3?. If so I am a bit concerned. For this reason, I do think the MALRT map is better which does not have the zero problem (as confirmed in lines 119/120)?.

Figure 4a: Should say Mexico

Figure 4f: It is quite clear that there is very little correlation between climate velocity and the rate of species movement uphill overall. I am wondering the overall r^2 for this graph (very low?) and whether there is any significant correlation at all. It is also interesting to see the breakdown for the different subsets. It seems maybe insects are the exception?

Referee #3 (Remarks to the Author):

I read with interest the fully re-worked revision of this study. I commend the authors for taking all the substantial input from myself and the other 4 reviewers into account. It is clear they have put great efforts into tackling the technical issues raised, and I cannot see much else they could have done given the limitations for space and in order to keep the study cohesive and focused.

While I was a bit disappointed initially with the relatively narrow inclusion of species datasets in the previous version, the incorporation of the BioShift database into the analysis has increased the taxonomic, spatial and environmental coverage.

I think this is an excellent contribution to one of the most important topics of our times (climate change) and to one of the most important aspects of climate change – its impact on biodiversity.

Below I list some general and specific aspects I think the authors could consider to further improve the manuscript.

1. Presentation of key results. Papers in Nature are usually very good at immediately attracting the reader's attention, and to appeal to a very broad range of readers. I think there are some simple elements that could improve simply through some re-writing and further thought:

a. Title: I did not find it very accessible and the inclusion of a question seems unusual for this journal. Perhaps a broader title might work better, although this could probably be discussed with the editor if the paper is eventually accepted.

b. Abstract: there is a contrasting mixture of overly technical as well as non-specific elements. For instance, "We found 17 mountain regions exhibiting high velocities of isotherm shifts that exceeded the 80th percentile by either MALRT or SALRT estimations" may be quite inaccessible to this journal's broad readership. And "upslope migrations of montane species have generally been lagging behind climate velocities" doesn't really say much – it may be better to focus on a few robust, quantitative results, as this and the coming sentences are quite general and do not come across as particularly compelling.

2. I did not feel fully content with the answers provided for my previous comment under "F. Suggested improvements: experiments, data for possible revision" where I suggested some further biological interpretation of the results. The authors now use the Lenoir et al 2020 dataset, which is an elegant compilation of species datasets across various taxonomic groups, geographies and habitats. They write that "the discussion of taxonomic variation may be beyond the scope of the current paper to discuss in detail and already and partially covered in Lenoir et al. (2020)".

I fully agree that a detailed discussion is beyond this study, but given that this is the first integration of the new isotherm estimates and Lenoir's dataset, I cannot see why the authors are unable to share a few key messages, and place them into our current knowledge of the topic – that would certainly be very appreciated by the biological community. After all, this paper is very focused on the biological impacts of climate change. Short considerations could be mentioned, such as whether plants or animals (or which major taxonomic and functional groups) are mostly likely to lag behind; an acknowledgement that distance alone – rather than ecological processes such as competition and other biotic interactions – may restrict migration; some mention that temperature extremes, rather than averages, may be more important to determine range shifts as constrained by physiological responses of the species affected; etc.

3. Data availability and access. The authors already make all data available (“we provided the raw data of the maps we generated for people who are interested in further exploration of the dataset”), but given that their dataset is arguably one of the most important contributions of their study, I would urge them to make an additional effort, if possible, to present the data in an even more accessible way. This is particularly important as the figures are at too large spatial scale for users to explore finer-scale questions, including those I listed in my previous review (such as the potential mountain refuges).

I might be asking for more than the authors would be able to provide – in which case please ignore this suggestion – but if at all feasible I think they could consider creating a simple, interactive website or Shiny App to allow for an easy zooming and exploration of these cell-specific velocity values for particular regions.

Alexandre Antonelli
July 3, 2023

Referee #4 (Remarks to the Author):

Thanks for the new version of the manuscript, which strongly improved in some parts (additional data set on ALRT, much broader biodiversity data). I am still convinced that this manuscript is timely and innovativ and will be of interest to a broad readership, but my major critical point was not really well solved: it is still unclear if we can trust the estimates of the adiabatic lapse rate (MALRT) – and this is of central importance to the paper. Its not enough to use satellite-estimates of temperature (SALRT); these are also estimates based on light emitted from the surface and no true temperature data. We have no indication if the lapse rates calculated by the authors and its global variation are realistic and this evidence has to be provided. Temperature data measured with weather stations or temperature loggers in the field should be available from many elevation gradients and I strongly recommend confronting the estimates of MALRT/SALRT with this data. This would also be a way of responding to the main critic of reviewer#1.

The statistical analyses appear to be sound and measures of error (typically standard deviation) are provided.

Other major points:

1) The authors did a good job in broadening the concepts to derive temperature lapse rates. However, as already criticized in my first review, the study does not validate the estimates of lapse rates with field data on temperature. Even though weather stations may be rare in some parts of the world, there are plenty of studies which measured temperature along elevation gradients. I am not a

climatologist, but I personally wonder about the quality of satellite-based estimates of temperature as long as they are not validated by true field data. Again, I strongly recommend to validate estimates of SALRT and MALRT with true data of adiabatic lapse rates from temperature loggers or weather stations from the field. This data is clearly available from many parts of the world (e.g. Appelhans et al. 2015, Int J Clim; Rapp & Silman 2012, Clim Res).

2) In the main text, some sentences on the data sources which were used to calculate SALRT and MALRT are missing. The same is true for the BIoShifts data set. These are not explained in detail or sometimes not mentioned in the main text. It also needs sentences in the main text why decisions for the use of certain data sets were done and why other data sets were not considered (e.g. CRU versus CHELSA). The quality of this data is of key importance for this study and readers should understand on what the whole analyses is based.

3) The authors sometimes rejected ideas of the reviewers, which would be somehow rather straight to incorporate – at least in a rough way. For example, I found reviewer#2s' critic on expanding the climate model by adding precipitation data really interesting and the authors could have done at least a rough model, testing if precipitation and its changes could better explain the high error/lack in species' tracking of temperature. But surprisingly, here the authors just argue that its not in the focus of the paper. I think this is not a satisfying reply.

4) I also want to repeat the following critic: It does not become clear how much better the new predictions of climate velocity are in comparison to the older estimates for predicting the observed shifting rate. Additionally, I find the observation of a general lag in species range shifts on mountains (which is not so much observed along latitudinal gradients) still really interesting – particularly given that the dispersal distances for tracking climate are some orders of magnitude smaller than along latitudinal gradients. I would advice to discuss this. What could be the reasons as distances are usually very short and it would be easy for an animals (or plant) to track the temperature in mountain areas?

5) I would recommend to base the MALRT on more than just the CRU data set. The authors reject the idea of using data with higher spatial resolution (e.g. CHELSA) by stating that the CRU data is better performing than other models but the sentences which was here cited (Karger et al. 2017) only concerned a small part (only for a dataset from China, I think it was even concerning precipitation data and not temperature data) of all the validation and comparisons which have been done in the Karger 2017 paper.

Minor points:

Lines 123-126: Here it is important not to compare the estimates only to two single studies but to all the data sets which are available. In case the estimates do not fit well to the field data, the authors should – if possible – try to analyze if the errors have a spatial pattern or can be explained by the environment.

Lines133-134: when seeing the estimates of adiabatic lapse rates and climate velocity from SALRT (is this realistic...maybe the mean but most likely not the extremes), I think it is even more important to validate the estimates to true temperature data from the field.

L182-185: This comes now a bit suddenly. This should be coming later and some data should be presented how much better the predictions of MALRT are.

L182...: Also the data set where all this biotic data comes from should be introduced with some key data.

L194: ..."resulting from the impact OF GLOBAL WARMING on MALRT are expected to remain urgent and serious."??

L210-212: I think that it is also of key importance to track climatic changes with weather stations in parallel with assessments of species range shifts – in the same area! Combining global data sets with a very low resolution with biotic data which is typically taken on study sites of small spatial extent is not unproblematic. There are a number of studies who present here already very nice data on temporal changes in temperature in parallel with changes in species ranges (Maihoff et al. 2023,

Diversity and Distributions; Kerner et al. 2023, Ecology).

L286-302: I am not a climatologist but I did not fully understand how the two gridded maps (MALRT/SALRT and the global temperature maps from CRU) were fused. I guess the CRU model incorporates already elevation information...). How well the CRU model perform in mountains? I would advise to give some indications. I also wondered – like other reviewers . why only the CRU data set was used as every climate model is a model with some errors. The authors argue in the response letter that CRU performs better than other climate models citing Karger et al. 2017. But the cited sentence from Karger et al. 2017 was not pointing to a general better performance of CRU over CHELSA and worldclim. The sentence just referred to a single comparison with data from China (in the paper multiple valdiations with different data sets were done and this was just one of them).

Fig 2g: typo: should be "satellite-based"

Fig 2: c,f,I are missing y axis labels and values (number of pixels?); additionally I would rescale the x axis ticks and values in 2i to the original scale. This is better interpretable.

Fig 4f,g: What does $p = 0.05$ on top of the panels mean? Should be explained. Additionally, I think that the colouring of the symbols is a bit complicated.

Referee #6 (Remarks to the Author):

The study explores the elevational velocity of climate change in mountainous regions of the world. The authors account for local differences in adiabatic lapse rates using two separate approaches—one based on satellite derived temperature and one based on the laws of thermodynamics involving the physical relationship between water vapor pressure and temperature. The revised manuscript does an excellent job of address most of the reviewers' concerns. Specifically, the increased resolution of the analyses, the updated climatic dataset, and the addition of the extensive species range-shift data greatly have greatly improved the study. I was asked to specifically focus on the concerns of reviewers 2 and 5. The concerns of reviewer 5 are well addressed. However, the concerns of reviewer 2, as well as at least one other important comment, have not been adequately addressed. I also have some minor comments and questions about the new methods and text.

1. Reviewer 2 noted that the lack of tracking of elevational shifts in temperature – particularly in dry regions—could be due to the fact that many species are responding to shifts in precipitation. While I agree that including precipitation gradients or other climatic factors in the analyses is too much to require, particularly because data to do so are limited, the revised manuscript does not adequately acknowledge the specific concern raised by the reviewer. There is mention of other climatic factors, but not specifically of the potential explanation of what might be happening in the drier regions.

2. One other point that was not adequately addressed was the need for empirical validation. The response to reviewers mentions this but I did not see anything in the manuscript. There must be some local datasets (like the two mentioned for the Himalayas and the Alps) that could be used to at least provide some assessment of the accuracy of the two approaches.

3. Line 228. This paragraph needs to be lightly edited for as there are typos and grammatical errors.

4. Line 372. This sentence is very difficult to understand.

5. Line 386. Were the 1000 iterations based on different sets of species making up the n species in the sample?

6. Line 394. I don't recall seeing the results of the sensitivity analysis. And did all grid cells (plots) have data for 100 species or were there holes in the data when you increased n?

Author Rebuttals to First Revision:

Referees' comments:

Referee #1 (Remarks to the Author):

I can see that the authors have made substantial revisions in response to my previous comments, including new analyses. The use of a surface-based lapse rate as well as the theoretical moist adiabatic lapse rate, as well as standardising the periods used for much of the data, and use of a better temperature dataset, are all welcome additions. As a result, I think the paper is now much more rigorous. There are still a few issues though remaining which I think need to be discussed.

Authors' response 1 to Referee #1:

We deeply appreciate the reviewer's overall positive feedback on our revised manuscript during the former round of revisions and we are glad to read that our work is now much more rigorous thanks to the helpful and insightful suggestions from Referee #1. We also wish to thank again Referee #1 for taking the time to assess again our work and for providing further helpful suggestions to further improve the quality of our manuscript.

First, the terminology is still somewhat confusing (and sometimes wrong) and the interpretation could be much clearer. I know that the authors are not originally speaking to the climatological/meteorological community, but at present you can tell that they are not climatologists and since this is an inter-disciplinary journal it is important that all communities can read and understand the methods and terms used. When talking about lapse rates (particularly lines 65-95), terms are still sometimes used incorrectly. The word adiabatic refers to the lack of exchange of energy between a rising parcel and its surroundings. This therefore leads to the theoretical specific dry adiabatic lapse rate (DALR) or a saturated/moist one (MALR) when condensation occurs. It is a specific case. The word cannot be used to describe any climatological lapse rate found on a mountain slope so the abbreviation ALRT should really just be LRT. The ALRT is not defined as the observed temperature difference along a slope (line 68) since the actual lapse rate observed is usually NOT

adiabatic. The mountain slope exchanges energy with its surroundings and the adiabatic assumption is violated (it is purely a theoretical assumption/model). The DALRT (9.8°C/km) is not the mean observed rate in the troposphere (as claimed at line 71-72), and only occurs when there is no condensation (or energy exchange). The mean observed rate is usually around 5.5 or 6.5°C/km depending on who you talk to. The other issue is that lapse rates on the ground (slope based) are usually referred to in the context of gradients in air temperature (measured at 2 m above ground level) – which is again different from the free air (or vertical) lapse rates measured above a single point (measured by a weather balloon). This difference needs to be made clear. Here you are estimating a slope based lapse rate because you are wanting to estimate temperature changes along a mountain slope. The DALRT and MALRT are theoretical free air values in the absence/presence of moisture (so can be estimated for a point as done in this paper) but they are not really mountain slope rates.

Authors' response 2 to Referee #1:

We deeply and sincerely appreciate the reviewer's positive attitude on our misuse of the terminology and for providing such nice clarifications that help us better communicate with interdisciplinary readers. According to the reviewer's suggestions here, we have revised, throughout the main text and figures, our acronyms and terminology so that ALRT is now replaced by LRT and SALRT by SLRT. Additionally, we have modified the description for DALRT as follows:

"According to the laws of thermodynamics⁶, the LRT is 9.8°C per km in the case of dry air^{1,6}. Nonetheless, given that the Earth's atmosphere is not entirely dry, the LRT experienced by terrestrial organisms in reality will be less steep than 9.8°C per km." (L72 - 74)

The SALRT (should be SLRT because there is no such thing as a surface adiabatic lapse rate) as derived here is an attempt to measure the slope based lapse rate based on observed surface temperatures (at ground level not at 2 m) and so this will be different from the required 2 m air temperature lapse rate which the authors really want. Because the MODIS data measures the actual surface temperature (at ground level) it is influenced by microscale surface properties (such as albedo, emissivity,

rock type, vegetation amount etc) and so there is a LOT of noise in the calculated lapse rate. It also is biased to cloud free conditions (which enhances this spatial variation) because when there is cloud there is no data. This needs to be explained/acknowledged and is probably why SALRT does not show such good relationships with the ecological data. Much of this noise would need to be edited out in order to make it representative of more than a very specific microclimate. Despite this, I do think it useful to use this lapse rate as well, because there is an ecological argument often made that organisms experience the microclimate near the ground, and so this is what should be more relevant to them. The fact that it appears not to be (!) is an interesting result.

Authors' response 3 to Referee #1:

We would like to thank Reviewer #1 for this very useful suggestion on the terminology of the SLRT, instead of SALRT, and most important for highlighting the potential reason why the SLRT does not provide a good fit with ecological data. We have now explicitly incorporated, as per the Reviewer's suggestion, the limitations of the SLRT into the main text of the manuscript as follows:

“This discrepancy between MALRT and SLRT is likely due to the fact that the satellite (MODIS) data measures the actual land surface temperature, which is influenced by microscale surface properties such as albedo, emissivity, rock type and vegetation cover. Hence, for the SLRT, the calculated lapse rate is characterized by significant noise. Moreover, the SLRT data is predominantly available in cloud-free conditions, which intensifies these spatial variations. As a consequence, satellite data presents multiple limitations, diminishing its capacity to explain species range shifts compared to insights obtained from theoretical calculations of the MALRT.” (L214 - 221)

At present none of this is explained and discussing this in more detail would also help explain some of the findings. I also think that the use of the terms high and low for lapse rate is just too ambiguous (even if ecologists use them as claimed). The climatological community defines a lapse rate as negative (since temperature decreases with height) – so the DALR would be $-9.8^{\circ}\text{C}/\text{km}$... This is a low absolute value (but a high number: is $-9.8^{\circ}\text{C}/\text{km}$ higher than $-6.5^{\circ}\text{C}/\text{km}$?) so confusing. To make it even more confusing the meteorological community sometimes defines the lapse rate with a positive number (e.g. $9.8^{\circ}\text{C}/\text{km}$). Thus without clarifying what high

and low mean you are causing lots of potential confusion to different communities who will interpret this differently.

Authors' response 4 to Referee #1:

We definitely agree with the Reviewer's suggestion here and we have consequently replaced all adjectives related to the lapse rate from 'low' and 'high' to 'shallow' and 'steep', respectively. This indeed clarifies the meaning in the manuscript.

I strongly suggest you cannot use high and low in this context... you must use something like steep or rapid (meaning a fast change) or shallow/weak (meaning a less rapid change) and explain your terminology in the paper. Since in this paper the change is always one way (it always gets colder as you go up?) then steep and shallow I think make sense to most people.

More specific points

Line 67: adiabatic used incorrectly

Line 70: incorrect: temperature does not influence the dry adiabatic lapse rate

Line 71: incorrect: the DALRT is not the mean observed rate

Authors' response 5 to Referee #1:

Modified as suggested. See *Authors' response 2 to Referee #1*.

Line 72: all not most?

Authors' response 6 to Referee #1:

The sentence has been removed as suggested by the comments above.

Line 77: a constant rate cannot be "about" 5.5C (it is that or not!)

Authors' response 7 to Referee #1:

We changed it to "a constant rate of 5.5°C per km" (L77)

Line 119/120: surely these must be the other way round... the SLRT shows more variation than MALRT and must go from -5.14 to 8.45°C/km?

Authors' response 8 to Referee #1:

Indeed, we have modified the sentence. (L121 -122)

Line 123: The negative rate (SLRT?) in northern mountains suggests the presence of persistent temperature inversions (with cold air stuck at the ground at lower elevations) – is this correct?

Authors' response 9 to Referee #1:

It is correct.

Line 138: I am a bit confused about how 32% of the area can have exceeded the 80th percentile? You would immediately think 20% because that is by definition. Is this because it can be exceeded with either of the methods? The fact that it is not 40% means there is some overlap between the methods? In any case I am not sure you should use the SLRT isotherm shifts to identify the most endangered locations since some of the SLRT isotherm shifts are unreliable because the lapse rate is near zero (see my comments later about figure 2).

Authors' response 10 to Referee #1:

Yes, that's overlap of high vertical velocities of isotherm shifts that exceeded the 80th percentile by either MALRT or SLRT estimations. In line with the reviewer's suggestion, we excluded the bottom N% of data closest to zero, irrespective of their direction. We tested multiple threshold values for N, including 0.5, 1, 2, and 5%. The results were largely consistent across these different chosen thresholds for outlier removal. For Fig. 3, we now present the case where N = 1 (representing the exclusion of 1% of data closest to zero). The results for other thresholds to remove extremes are shown in Supplementary Fig. 2. The results are similar.

Line 166: Isotherm velocities are higher at lower elevations (and so weaker in high mountains). This is despite EDW and the enhanced warming in high mountains? So does the change in lapse rate overcompensate for any enhanced warming? i.e. it would be quite nice to be able to separate the effects of warming rate (EDW) from changing lapse rates, and for this to be explained... because it seems like a contradiction?

Authors' response 11 to Referee #1:

We have incorporated your suggestions and added the analysis as follows:

“We further analyzed the effects of changes in surface temperature and MALRT on the rates of isotherm shift with elevation (Supplementary Fig. 1). We found no significant linear correlation between the rate of surface temperature change and elevation when the effect of latitude is statistically controlled. However, the MALRT becomes steeper with increasing elevation, leading to lower velocities of isotherm shifts at higher elevations compared to lower elevations (i.e., steeper MALRT corresponds to lower velocities of isotherm shifts).” (L189 - 195)

Line 172: Why is the latitudinal effect reversed? Explain.. (is this expected or not)? I am left thinking whether this is due to positive and negative latitudes in the two hemispheres and the scales being inverted?

Authors' response 12 to Referee #1:

This result is not due to the opposing signs of latitude in the Northern and Southern Hemispheres. We used absolute values of latitude in the analysis but added an interaction term with the factor variable “hemisphere” with two levels (Northern vs. Southern). We have added an explanatory section to the results as follows:

“The latitudinal effect we detected here is likely due to the reduction of land area towards higher latitudes, and before reaching Antarctica, in the Southern Hemisphere, where oceans predominate over land masses, leading to relatively higher water vapor pressure (Extended Data Fig. 2b) and consequently lower MALRT (Fig. 1d).” (L186 - 189)

Line 194: The “impacts resulting from MALRT” – what does this mean?

Authors' response 13 to Referee #1:

We have modified this sentence to:

“impacts resulting from shallow MALRT” (L230)

Line 256: So the SALRT for a pixel is some sort of average of multiple transects? How many transects in each cell? Is there a relationship between variability in the SALRT and the number of transects? i.e. is it less extreme (more reliable?) when there are more transects contributing to it (this is what I would expect)?

Authors' response 14 to Referee #1:

The median number of available transects for each grid cell is 8, with an interquartile range of 12 transects (see Extended Data Fig. 7). Nevertheless, we did not observe a significant correlation between the number of available transects and the variability in the SLRT (using the interquartile range for consistency). This lack of correlation might be due to the filtering procedure we adopted, which mandated a minimum R^2 value of 0.5 between temperature and elevation (as elaborated in the Methods section). However, it's important to highlight that the number of available transects does influence the average R^2 value between temperature and elevation across the transects ($R^2 = 0.16$, $p < 0.0001$). In particular, a grid with more transects tends to have a higher averaged R^2 (see the new Extended Data Figure 7 also appended below), indicating that the reliability of SLRT is partly determined by the volume of available transects within a grid. We have modified our text accordingly.

“Within the framework of our SLRT computations, the median transect count per grid cell stood at 8, showcasing an interquartile range of 12 (Extended Data Fig. 7a). We noticed that a higher transect availability within a grid cell was correlated with increased average R^2 values between temperature and elevation ($R^2 = 0.16$, $p < 0.001$; Extended Data Fig. 7c), underscoring the dependency of the reliability of the SLRT on the number of accessible transects.” (L296 - 301)

Extended Data Figure 7. Influence of number of available transects on SLRT result. (a) Distribution of number of available transects. (b) Correlation between number of available transects and the SLRT interquartile range. (c) Correlation between number of available transects and the averaged R^2 between elevation and temperature. Blue lines indicate simple regression between the two variables, with statistics labeled at the lower right of each panel. Orange lines represent LOESS (locally estimated scatterplot smoothing) lines. Significance levels are indicated: ***, $p < 0.001$.

Line 300: I thought $6.5^\circ\text{C}/\text{km}$ was the most commonly assumed lapse rate?

Authors' response 15 to Referee #1:

Not exactly. In the field of climate change biology, $5.5^\circ\text{C}/\text{km}$ is the most commonly used value.

Line 312: Why 20% or 10% - do not follow – are there two classifications?

Authors' response 16 to Referee #1:

We have revised our figures accordingly and we hope that now this is much clearer. (Fig. 3a)

Figure 2: It is hard to see the detail here but it generally seems that isotherm shifts are larger in higher latitudes, because of both the enhanced rate of warming (arctic amplification) and the weaker lapse rate. There is a discontinuity however as the lapse rate gets shallower – if we have a very shallow (but +ve lapse rate) we end up with the isotherms moving uphill extremely rapidly. As soon as the lapse rate becomes negative the isotherms move downhill extremely rapidly. This cannot be correct in reality (for a very small change in lapse rate -say from $+0.1^{\circ}\text{C}/\text{km}$ to $-0.1^{\circ}\text{C}/\text{km}$) the situation completely reverses. Thus in the SLRT map you have areas of very rapid change in isotherms in both directions right next to each other?. This is an artefact of the methodology and needs mentioning (i.e. the method is not reliable once the mean lapse rate gets near zero). I think for that reason I would block out any rapid shifts (in both directions) which are caused in this situation. Are any of these rapid shifts used to identify the critical regions in figure 3?. If so I am a bit concerned. For this reason, I do think the MALRT map is better which does not have the zero problem (as confirmed in lines 119/120)?.

Authors' response 17 to Referee #1:

We concur with the Reviewer's observations here that extremely low SLRT values, both positive and negative (i.e. shallow rates), imply pronounced elevational shifts in terms of the vertical projection of the velocities at which isotherms are moving upslope or downslope. Recognizing the potential artifact this represents when the mean lapse rate for SLRT approaches zero, we have adopted a more conservative data-filtering approach.

To address this, we've omitted data that are in close proximity to zero for the SLRT, irrespective of their positive or negative nature. Specifically, we experimented with excluding various percentages of data near zero, including 0.5%, 1%, 2%, and 5%. Our findings indicate a consistent trend across these exclusion thresholds. For the sake of clarity in Fig. 3, we illustrated the scenario where 1% of the data closest to zero was excluded. Additional outcomes, pertaining to other data exclusion thresholds, are presented in Supplementary Fig. 2.

We believe that this approach provides a more robust representation while acknowledging the limitations, as noted by the reviewer. We are grateful for this feedback, as it has significantly improved the rigor and clarity of our study.

Fig. 3. Identifying mountain regions threatened by high vertical velocities of isotherm shifts and underlying mechanisms. Consensus map of the vertical velocities of isotherm shifts as estimated from the satellite-based lapse rate (SLRT) or from the moist adiabatic lapse rate (MALRT) (see Fig. 2). (a-c) Mountain regions where velocities are greater than the 80% quantile (i.e. retaining 20%) in either

calculations of MALRT or SLRT are labelled as critically threatened (a-b) and displayed in red (c). (d) Orange bars represent mean annual temperature change between 1971-1980 and 2011-2020, while blue bars represent mean water vapor pressure during 2011-2020 for each of the 17 mountain regions affected by relatively fast vertical velocities of isotherm shifts (see Fig. 3). The error bars represent standard deviation. Supplementary Data 1 and the Data Availability section furnish a comprehensive breakdown for each region. Considering that near-zero SLRT values result in extremely high climate velocity, we remove 1% outliers that are close to zero when illustrating Fig. 3c. Data with alternative levels of outlier removal (0.5%, 2%, and 5%) are shown in Supplementary Fig. 2. Supplementary Data 3 provides the high-resolution map.

Supplementary Fig. 2. Identifying mountain regions threatened by high vertical velocities with different outlier removal levels. A consensus map of the vertical velocities of isotherm shifts as estimated from the satellite-based lapse rate (SLRT) or from the moist adiabatic lapse rate (MALRT) (see Fig. 2). (a-c) Mountain regions where velocities are greater than the 80% quantile (i.e. retaining 20%) in either calculations of MALRT or SLRT are labelled as critically threatened (a-b) and displayed in red (c-e). To address potential data artifacts, varying percentages of extremely low absolute SLRT values were excluded: (c) 0.5%, (d) 2%, and (e) 5%. For reference, the outcomes upon excluding 1% of these outlier SLRT values are detailed in Fig. 3.

Figure 4a: Should say Mexico

Authors' response 18 to Referee #1:

Modified as suggested. (Fig. 4a)

Figure 4f: It is quite clear that there is very little correlation between climate velocity and the rate of species movement uphill overall. I am wondering the overall r^2 for this graph (very low?) and whether there is any significant correlation at all. It is also interesting to see the breakdown for the different subsets. It seems maybe insects are the exception?

Authors' response 19 to Referee #1:

The correlation in Figure 4f is indeed low. This is why we employed the method illustrated in Figure 4g to calculate the probability of organisms tracking climatic velocities. Our results indicate that the climatic velocities calculated using MALRT is superior to that derived from satellite data and previous results using a fixed lapse rate (as presented by the AIC values).

“Indeed, the Akaike Information Criterion (AIC) values from our models are 35887, 37016, and 51398 for the MALRT, constant LRT and SLRT, respectively, ranking from best to worst in model fit.” (L212 - 214)

Referee #3 (Remarks to the Author):

I read with interest the fully re-worked revision of this study. I commend the authors for taking all the substantial input from myself and the other 4 reviewers into account. It is clear they have put great efforts into tackling the technical issues raised, and I cannot see much else they could have done given the limitations for space and in order to keep the study cohesive and focused.

Authors' response 1 to Referee #3:

We are very glad to read that very positive feedback from Reviewer #3 on the quality of our revisions during the former round of review. This kind of positive feedback means a lot to us as authors and we really appreciate it. We have done our best to account for the last remaining comments from the four remaining reviewers.

While I was a bit disappointed initially with the relatively narrow inclusion of species datasets in the previous version, the incorporation of the BioShift database into the analysis has increased the taxonomic, spatial and environmental coverage. I think this is an excellent contribution to one of the most important topics of our times (climate change) and to one of the most important aspects of climate change – its impact on biodiversity.

Authors' response 2 to Referee #3:

We would like to thank again Reviewer #3 for his very supportive and positive comments on our work and for taking the time to re-evaluate our work in the light of our revisions and for also providing additional and meaningful insights that definitely improved the quality and robustness of our work.

Below I list some general and specific aspects I think the authors could consider to further improve the manuscript.

1. Presentation of key results. Papers in Nature are usually very good at immediately attracting the reader's attention, and to appeal to a very broad range of readers. I

think there are some simple elements that could improve simply through some re-writing and further thought:

Authors' response 3 to Referee #3:

We agree and we did our best to improve the writing as per the reviewer's suggestions.

a. Title: I did not find it very accessible and the inclusion of a question seems unusual for this journal. Perhaps a broader title might work better, although this could probably be discussed with the editor if the paper is eventually accepted.

Authors' response 4 to Referee #3:

We have changed the title according to the reviewer's suggestion to avoid the question mark in the title and to make it more general. We hope that it now better reflects our work while providing a broad perspective. We are of course open to suggestions from the editorial board if the title needs to be further crafted. The title now reads:

"Climate Velocities and Species Tracking in Global Mountain regions"

b. Abstract: there is a contrasting mixture of overly technical as well as non-specific elements. For instance, "We found 17 mountain regions exhibiting high velocities of isotherm shifts that exceeded the 80th percentile by either MALRT or SALRT estimations" may be quite inaccessible to this journal's broad readership. And "upslope migrations of montane species have generally been lagging behind climate velocities" doesn't really say much – it may be better to focus on a few robust, quantitative results, as this and the coming sentences are quite general and do not come across as particularly compelling.

Authors' response 5 to Referee #3:

We thank Reviewer #3 for these thoughtful suggestions to make the abstract more accessible to a wide reader while being more accurate and quantitative in the result section of the abstract. We have revised our abstract accordingly, aiming to make it more accessible for general readers.

“We discovered 17 mountain regions exhibiting particularly high vertical velocities of isotherm shifts (> 11.67 m/yr for SLRT and > 8.25 m/yr for MALRT). High velocities are typically found in relatively dry parts of the world, but also occur in wet regions with shallow lapse rates, such as in Northern Sumatra, the Brazilian Highlands and Southern Africa. We further related our mountain-specific velocities of isotherm shift along elevation gradients to biotic velocities of species range shift and revealed cases of tight tracking in mountain regions undergoing lower exposures in terms of climate velocities. Nevertheless, many species are not shifting fast enough to track the climate velocities upslope. Such lagging dynamics suggest that species will continue to shift their range even if the climate were to stabilize immediately.”
(L30 - 39)

2. I did not feel fully content with the answers provided for my previous comment under “F. Suggested improvements: experiments, data for possible revision” where I suggested some further biological interpretation of the results. The authors now use the Lenoir et al 2020 dataset, which is an elegant compilation of species datasets across various taxonomic groups, geographies and habitats. They write that “the discussion of taxonomic variation may be beyond the scope of the current paper to discuss in detail and already and partially covered in Lenoir et al. (2020)”. I fully agree that a detailed discussion is beyond this study, but given that this is the first integration of the new isotherm estimates and Lenoir’s dataset, I cannot see why the authors are unable to share a few key messages, and place them into our current knowledge of the topic – that would certainly be very appreciated by the biological community. After all, this paper is very focused on the biological impacts of climate change. Short considerations could be mentioned, such as whether plants or animals (or which major taxonomic and functional groups) are mostly likely to lag behind; an acknowledgement that distance alone – rather than ecological processes such as competition and other biotic interactions – may restrict migration; some mention that temperature extremes, rather than averages, may be more important to determine range shifts as constrained by physiological responses of the species

affected; etc.

Authors' response 6 to Referee #3:

We agree with the Reviewer's suggestion here and we have added a few sentences related to this point in the discussion section, as suggested by the reviewer:

“Our results suggest that the vertical distance between isotherms in mountains is a critical factor constraining migration. Moreover, based on our findings, all taxonomic groups will be similarly affected in their abilities to track isotherms along mountain slopes. Considering that the distance of climate tracking is several orders of magnitude shorter in elevation compared to latitudinal gradients, the moving capability of organisms is less likely to be the key constraint in mountain systems. Mountainous regions, with their complex topography, occupy a relatively smaller proportion of land masses compared to other terrains in the lowlands²⁸. This, combined with biotic interactions like interspecific competition^{29,30}, might collectively limit the ability of mountain species to track isotherm shifts in the future.” (L238 - 246)

3. Data availability and access. The authors already make all data available (“we provided the raw data of the maps we generated for people who are interested in further exploration of the dataset”), but given that their dataset is arguably one of the most important contributions of their study, I would urge them to make an additional effort, if possible, to present the data in an even more accessible way. This is particularly important as the figures are at too large spatial scale for users to explore finer-scale questions, including those I listed in my previous review (such as the potential mountain refuges). I might be asking for more than the authors would be able to provide – in which case please ignore this suggestion – but if at all feasible I think they could consider creating a simple, interactive website or Shiny App to allow for an easy zooming and exploration of these cell-specific velocity values for particular regions.

Authors' response 7 to Referee #3:

This is a very valid and important point for making our data more attractive and thus potentially increase its usability by the scientific community at large. Hence, as suggested by the reviewer, we have created a Google Earth layer file (*.kmz; find the Supplementary File 3). This allows readers to clearly view and explore, in an

interactive manner, our results by zooming in on some of the mountain regions worldwide. Again, we wish to thank Alexandre Antonelli for his very constructive feedback on our work.

Alexandre Antonelli

July 3, 2023

Referee #4 (Remarks to the Author):

Thanks for the new version of the manuscript, which strongly improved in some parts (additional data set on ALRT, much broader biodiversity data). I am still convinced that this manuscript is timely and innovativ and will be of interest to a broad readership, but my major critical point was not really well solved: it is still unclear if we can trust the estimates of the adiabatic lapse rate (MALRT) – and this is of central importance to the paper. Its not enough to use satellite-estimates of temperature (SALRT); these are also estimates based on light emitted from the surface and no true temperature data. We have no indication if the lapse rates calculated by the authors and its global variation are realistic and this evidence has to be provided. Temperature data measured with weather stations or temperature loggers in the field should be available from many elevation gradients and I strongly recommend confronting the estimates of MALRT/SALRT with this data. This would also be a way of responding to the main critic of reviewer#1.

Authors' response 1 to Referee #4:

We thank Reviewer #4 for the suggestions to validate our computations of the MALRT and the SLRT. We now use ground-truth weather station data when these are available within our studied mountain regions to validate our estimates derived from either the MALRT or the SLRT. Because weather stations are scarce in mountainous regions, especially towards the highest elevations, this validation procedure cannot fully capture the entire elevation gradients we are considering in our computations at a global extent. However, it is true, as suggested by Reviewer #4 that it can provide some information on the quality and relevance of our computations. For that

reason, we decided to perform an additional analysis to address this key point (see the Methods section for the new paragraph on station-based LRT values). The results of this supplementary analysis indicate a good degree of consistency between our computations of the MALRT and SLRT with the ground-truth and empirical LRT values based on weather station data, and moreover, MALRT exhibits the highest relative importance in explaining the results from the weather stations. The detailed findings are as follows:

“For comparison purposes and external validation, we also extracted data from the Global Historical Climatology Network (GHCN²³) — focusing on empirical field data recorded by weather stations situated within mountain regions worldwide. We manage to obtain temperature lapse rates from 144 weather stations (i.e., station-based LRT; see Methods) across a total of 25 mountain sites from 2011 to 2019 (Extended Data Fig. 3a). This validation exercise shows spatial consistency between lapse rates calculated from empirical weather station data and both our computations of the MALRT and SLRT ($R^2 = 0.22 \pm 0.01$; mean \pm standard deviation). Furthermore, the MALRT has the highest explanatory power ($32.71 \pm 0.48\%$) in relation to station-based LRT compared to other variables, such as SLRT, latitude, and longitude (Extended Data Fig. 3b). However, due to the relative scarcity of weather station data and its primary concentration in North America and Europe, our subsequent analyses will focus solely on our computations of the MALRT and SLRT.” (L128 -139)

The statistical analyses appear to be sound and measures of error (typically standard deviation) are provided.

Authors’ response 2 to Referee #4:

We thank Reviewer 4 for assessing the soundness of our statistical analyses and for providing meaningful suggestions to improve our analyses.

Other major points:

1) The authors did a good job in broadening the concepts to derive temperature lapse rates. However, as already criticized in my first review, the study does not validate the estimates of lapse rates with field data on temperature. Even though

weather stations may be rare in some parts of the world, there are plenty of studies which measured temperature along elevation gradients. I am not a climatologist, but I personally wonder about the quality of satellite-based estimates of temperature as long as they are not validated by true field data. Again, I strongly recommend to validate estimates of SALRT and MALRT with true data of adiabatic lapse rates from temperature loggers or weather stations from the field. This data is clearly available from many parts of the world (e.g. Appelhans et al. 2015, *Int J Clim*; Rapp & Silman 2012, *Clim Res*).

Authors' response 3 to Referee #4:

See above our former response to this comment on validating our computations of the MALRT and the SLRT at the global extent (cf. *Authors' response 1 to Referee #4*) using field observations from local weather station data. We thank Reviewer #4 for the suggested references that we did consider here and for suggesting to perform an external validation to assess the accuracy of our products. We also wish to highlight here that Reviewer #1 is actually a climatologist who provided very relevant and important comments and advices to help us clarify this point on the quality and reliability of the MALRT and SLRT values. Reviewer #1 also mentioned that the SLRT values are highly dependent on the quality of the satellite images and more likely to be subjected to noise than the MALRT values, hence concluding on the MALRT being a more robust and reliable estimate for mapping the lapse of rate of temperature along mountain slopes worldwide (see Reviewer #1's comments and suggestions which we accepted and addresses in this new revised version).

2) In the main text, some sentences on the data sources which were used to calculate SALRT and MALRT are missing. The same is true for the Bloshifts data set. These are not explained in detail or sometimes not mentioned in the main text. It also needs sentences in the main text why decisions for the use of certain data sets were done and why other data sets were not considered (e.g. CRU versus CHELSA). The quality of this data is of key importance for this study and readers should understand on what the whole analyses is based.

Authors' response 4 to Referee #4:

We are sorry if some information were sometimes missing regarding the data sources. We have now addressed this point by clarifying the Methods section when necessary (see L275 - 302). We have also adjusted the main text accordingly to explain some of our choices regarding data sources or analytical decisions (see L137 - 139, L205 - 212). We have also followed the Reviewer's recommendation and calculated the MALRT using CHELSA as the input data instead of the CRU data. The results are highly consistent with those obtained from the CRU data (see Extended Data Fig. 8).

3) The authors sometimes rejected ideas of the reviewers, which would be somehow rather straight to incorporate – at least in a rough way. For example, I found reviewer#2s' critic on expanding the climate model by adding precipitation data really interesting and the authors could have done at least a rough model, testing if precipitation and its changes could better explain the high error/lack in species' tracking of temperature. But surprisingly, here the authors just argue that its not in the focus of the paper. I think this is not a satisfying reply.

Authors' response 5 to Referee #4:

In some of our analyses, we actually did include precipitation data, and specifically when analyzing species' tracking of temperature (see L850 - 854), as suggested by the reviewer in this comment. However, precipitation had a low explanatory power when related to the velocities of species range shifts in elevation, which is why it is not depicted in the main figures of the manuscript. Hence, as suggested by Reviewer #4 here, we did test if precipitation data could better explain the high error/lack in species' tracking of temperature, but it did not. For that reason, we did not deem it necessary to focus on this point in the main figures of the manuscript. Yet, if the editorial team deems it necessary and if space allows for it, we are very happy to further expand on this point and even add an additional figure showing the low explanatory power of precipitation when analyzing species' tracking of temperature. As for good historical data on precipitation changes along elevation over the past

decades and an equivalent of lapse rate but for precipitation patterns along elevation gradients instead of temperature, these kinds of data are simply unavailable for precipitation due to a lack of historical weather stations recording long-term precipitation time series in mountain regions, and especially so towards the highest elevational bands of a mountain. This said, we agree with Reviewer #4 that it is important to remind the reader about the potential influence of precipitation while also acknowledging the scarcity of precipitation data in mountainous regions worldwide. For that reason, we have also added a few sentences in the discussion section about precipitation:

“Furthermore, some studies have shown that changes in precipitation patterns can affect mountain species range shifts^{15,40}, but historical data on precipitation patterns along mountain slopes is extremely scarce compared with data on temperature lapse rates. For that reason, it remains challenging to assess the large-scale impacts of precipitation changes on mountainous organisms.” (L258 -261)

4) I also want to repeat the following critic: It does not become clear how much better the new predictions of climate velocity are in comparison to the older estimates for predicting the observed shifting rate. Additionally, I find the observation of a general lag in species range shifts on mountains (which is not so much observed along latitudinal gradients) still really interesting – particularly given that the dispersal distances for tracking climate are some orders of magnitude smaller than along latitudinal gradients. I would advice to discuss this. What could be the reasons as distances are usually very short and it would be easy for an animals (or plant) to track the temperature in mountain areas?

Authors’ response 6 to Referee #4:

We agree with Reviewer #4 here and we added a paragraph in the discussion section, as follows:

“Our results suggest that the vertical distance between isotherms in mountains is a critical factor constraining migration. Moreover, based on our findings, all taxonomic groups will be similarly affected in their abilities to track isotherms along mountain slopes. Considering that the distance of climate tracking is several orders of magnitude shorter in elevation compared to latitudinal gradients, the moving

capability of organisms is less likely to be the key constraint in mountain systems. Mountainous regions, with their complex topography, occupy a relatively smaller proportion of land masses compared to other terrains in the lowlands²⁸. This, combined with biotic interactions like interspecific competition^{29,30}, might collectively limit the ability of mountain species to track isotherm shifts in the future.” (L238 - 246)

5) I would recommend to base the MALRT on more than just the CRU data set. The authors reject the idea of using data with higher spatial resolution (e.g. CHELSA) by stating that the CRU data is better performing than other models but the sentences which was here cited (Karger et al. 2017) only concerned a small part (only for a dataset from China, I think it was even concerning precipitation data and not temperature data) of all the validation and comparisons which have been done in the Karger 2017 paper.

Authors' response 7 to Referee #4:

We agree with Reviewer #4 here and we have addressed this point by also testing our approach to compute the MALRT but using CHELSA data as input data instead of the CRU data. Please refer to *Authors' response 4 to Referee #4* for more details on how we addressed this point.

Minor points:

Lines 123-126: Here it is important not to compare the estimates only to two single studies but to all the data sets which are available. In case the estimates do not fit well to the field data, the authors should – if possible – try to analyze if the errors have a spatial pattern or can be explained by the environment.

Authors' response 8 to Referee #4:

We now provide additional analyses to validate our computations of the MALRT and SLRT using ground-truth observations from the weather stations that are available within our studied mountain regions. Please see *Authors' response 1 to Referee #4*.

Lines133-134: when seeing the estimates of adiabatic lapse rates and climate velocity from SALRT (is this realistic...maybe the mean but most likely not the extremes), I think it is even more important to validate the estimates to true temperature data from the field.

Authors' response97 to Referee #4:

Please see response *Authors' response 1 to Referee #4.*

L182-185: This comes now a bit suddenly. This should be coming later and some data should be presented how much better the predictions of MALRT are.

Authors' response 10 to Referee #4:

We have added the results of validating weather station data using MALRT and SLRT in the earlier section. We also add transition sentences as suggested, as follows:

“Next, we used our estimates of the vertical velocities of isotherm shifts in mountains and linked it to empirical data on the velocities of species range shifts along mountain slopes. We used a carefully curated dataset —BioShifts⁴— which provides the velocities of species range shifts (in m/yr along elevation gradients) per taxonomic unit after a standardization procedure of the raw range shift estimates as reported by authors in their original studies.” (L204 - 208)

We hope this revision makes the transition smoother for readers.

L182...: Also the data set where all this biotic data comes from should be introduced with some key data.

Authors' response 11 to Referee #4:

Modified as suggested. See response *Authors' response 10 to Referee #4.*

L194: ...”resulting from the impact OF GLOBAL WARMING on MALRT are expected to remain urgent and serious.”??

Authors’ response 12 to Referee #4:

We have modified the sentence as follows:

“it is important to note that the impacts resulting from shallow MALRT are expected to remain urgent and serious.” (L230 - 231)

L210-212: I think that it is also of key importance to track climatic changes with weather stations in parallel with assessments of species range shifts – in the same area! Combining global data sets with a very low resolution with biotic data which is typically taken on study sites of small spatial extent is not unproblematic. There are a number of studies who present here already very nice data on temporal changes in temperature in parallel with changes in species ranges (Maihoff et al. 2023, Diversity and Distributions; Kerner et al. 2023, Ecology).

Authors’ response 13 to Referee #4:

We thank Reviewer #4 for this suggestion. We concur that in locations where weather station data is accessible, directly integrating this data with biological information is a preferable approach. However, on a global scale, there are few places with weather stations situated along elevational gradients that also possess sufficiently long historical records. Hence, the need to utilize global datasets for estimation climate velocities. Nevertheless, we now provide a test of our estimates of the MALRT against ground-truth observations, when available, from local weather stations.

L286-302: I am not a climatologist but I did not fully understand how the two gridded maps (MALRT/SALRT and the global temperature maps from CRU) were fused. I guess the CRU model incorporates already elevation information...). How well the CRU model perform in mountains? I would advise to give some indications. I also wondered – like other reviewers . why only the CRU data set was used as every climate model is a model with some errors. The authors argue in the response latter that CRU performs better than other climate models citing Karger et al. 2017. But the cited sentence from Karger et al. 2017 was not pointing to a general better performance of CRU over CHELSA and worldclim. The sentence just referred to a single comparison with data from China (in the paper multiple valdiations with different data sets were done and this was just one of them).

Authors' response 14 to Referee #4:

Please see *Authors' response 4 to Referee #4*.

Fig 2g: typo: should be “satellite-based”

Authors' response 15 to Referee #4:

Modified as suggested.

Fig 2: c,f,l are missing y axis labels and values (number of pixels?); additionally I would rescale the x axis ticks and values in 2i to the original scale. This is better interpretable.

Authors' response 16 to Referee #4:

Modified as suggested.

Fig 4f,g: What does $p = 0.05$ on top of the panels mean? Should be explained. Additionally, I think that the colouring of the symbols is a bit complicated.

Authors' response 17 to Referee #4:

Added in the figure legend as suggested.

Referee #6 (Remarks to the Author):

The study explores the elevational velocity of climate change in mountainous regions of the world. The authors account for local differences in adiabatic lapse rates using two separate approaches—one based on satellite derived temperature and one based on the laws of thermodynamics involving the physical relationship between water vapor pressure and temperature. The revised manuscript does an excellent job of address most of the reviewers' concerns. Specifically, the increased resolution of the analyses, the updated climatic dataset, and the addition of the extensive species range-shift data greatly have greatly improved the study. I was asked to specifically focus on the concerns of reviewers 2 and 5. The concerns of reviewer 5 are well addressed. However, the concerns of reviewer 2, as well as at least one other important comment, have not been adequately addressed. I also have some minor comments and questions about the new methods and text.

1. Reviewer 2 noted that the lack of tracking of elevational shifts in temperature – particularly in dry regions—could be due to the fact that many species are responding to shifts in precipitation. While I agree that including precipitation gradients or other climatic factors in the analyses is too much to require, particularly because data to do so are limited, the revised manuscript does not adequately acknowledge the specific concern raised by the reviewer. There is mention of other climatic factors, but not specifically of the potential explanation of what might be happening in the drier regions.

Authors' response 1 to Referee #6:

We thank Reviewer #6 for providing insightful and helpful comments and suggestions on our work. We agree with Reviewer #6 here that we initially did a poor job at acknowledging the potential importance of precipitation in our findings. Hence, we have now added several sentences in the discussion section to address this point. It reads as follows:

“Furthermore, some studies have shown that changes in precipitation patterns can affect mountain species range shifts^{15,40}, but historical data on precipitation patterns along mountain slopes is extremely scarce compared with data on temperature lapse rates. For that reason, it remains challenging to assess the large-scale impacts of precipitation changes on mountainous organisms.” (L258 - 261)

2. One other point that was not adequately addressed was the need for empirical validation. The response to reviewers mentions this but I did not see anything in the manuscript. There must be some local datasets (like the two mentioned for the Himalayas and the Alps) that could be used to at least provide some assessment of the accuracy of the two approaches.

Authors’ response 2 to Referee #6:

Indeed, we agree and we have incorporated data from local weather stations to validate our results. Please refer to *Authors’ response 1 to Referee #4*.

3. Line 228. This paragraph needs to be lightly edited for as there are typos and grammatical errors.

Authors’ response 3 to Referee #6:

Thanks for pointing this out. We have carefully edited and revised the paragraph as follows:

“In assessing the SLRT, we focused on daily land surface temperature data from the MODIS Land Surface Temperature and Emissivity (MOD11C3) product⁴¹. This data, encompassing the period 2011-2020 and featuring a native spatial resolution of 1 km at the equator, was averaged from both daytime

and nighttime observations. Monthly mean values from this product were aggregated at an annual resolution to derive the mean annual temperature, which was subsequently averaged over the 2011-2020 decade. To harmonize the spatial resolution for subsequent computations with other gridded products relying on the CRU TS4.05 data, the MODIS data was aggregated, using the mean value, from its native spatial resolution to a 0.05° resolution (Extended Data Table 1), which is approximately 5 km at the equator, ensuring ample grid cells for subsequent analyses. Using a moving window centered on a grid cell of 0.5° resolution, which is about 50 km at the equator, elevational transects were derived to empirically compute the LRT from satellite observations. This involved pinpointing regional peaks and foothills within a 1.5° by 1.5° window centered on the 0.5° target grid cell, with elevation data sourced from a digital elevation model (DEM) that was aggregated to match the 0.05° resolution of the aggregated MODIS grid (Extended Data Fig. 1a). From these peaks and foothills, elevational transects connecting the nearest topographical features were established (Extended Data Fig. 1b-c). Linear regressions between mean annual temperature and elevation, both at the 0.05° resolution, were subsequently fitted for each transect intersecting the target 0.5° grid cell (Extended Data Fig. 1d-f). All pixel units intersected by a focal transect were considered, even if only marginally. Transects yielding significant lapse rates ($R^2 \geq 0.5$ and $p \leq 0.05$) were retained, with the slope coefficient (β) representing the SLRT value in °C/m — later converted to °C/km. If over ten transects intersected a target 0.5° grid cell, the median SLRT value was calculated to mitigate biases from transect count extremities. Within the framework of our SLRT computations, the median transect count per grid cell stood at 8, showcasing an interquartile range of 12 (Extended Data Fig. 7a). We noticed that a higher transect availability within a grid cell was correlated with increased average R^2 values between temperature and elevation ($R^2 = 0.16$, $p < 0.001$; Extended Data Fig. 7c), underscoring the dependency of the reliability of the SLRT on the number of accessible transects.” (L275 - 302)

4. Line 372. This sentence is very difficult to understand.

Authors' response 4 to Referee #6:

We have modified the sentence as follows:

“Then, we computed the likelihood of a specific species from a designated taxonomic group (plants, birds, mammals, gastropods, insects, amphibians or reptiles; details provided in the Supplementary Information) to track the vertical velocities of isotherm shifts within a particular mountainous area. To achieve this, we randomly resampled a fixed number of elevational range shift observations for each taxonomic group within each mountain region. This ensured relatively consistent and balanced sample sizes across all the examined mountain regions and taxonomic groups.” (L450 - 456)

5. Line 386. Were the 1000 iterations based on different sets of species making up the n species in the sample?

Authors' response 5 to Referee #6:

Correct. The 1,000 iterations is based on different sets of species making up the n species in the sample. Therefore, the species compositions are different among iterations, but note that if the total number of records is smaller than the sample n , all records were used.

6. Line 394. I don't recall seeing the results of the sensitivity analysis. And did all grid cells (plots) have data for 100 species or were there holes in the data when you increased n ?

Authors' response 6 to Referee #6:

The result is provided in Supplementary Data 2.

Reviewer Reports on the Second Revision:

Referees' comments:

Referee #1 (Remarks to the Author):

I can see that the authors have made substantial changes to improve the analysis in response to my previous review and I do think the paper is almost suitable for publication.

Rather than make a lot of detailed comments again, I think it worth clarifying a few arguments (the first of which other reviewers have also commented on).

The main concern is still the reliability of the lapse rate calculations. I think nearly all reviewers (and perhaps the authors?) agree that long term weather stations measuring air temperatures at 2 m on mountain slopes would be the best source to calculate slope specific lapse rates, but that they are not universally available. While it is true there are some mountain transects, it is also true that reliable long-term measurements across elevational gradients are few and far between. Extended Data Figure 3 does not convince me that MALRT is a great representation of lapse rates on the ground ($r=0.22$ is not a good predictive model!), even though it may be the best estimate available in this paper (better than constant rate or SLRT). I therefore think this point does need to come out in the discussion which should argue for better mountain meteorological networks along elevational gradients, preferably combined with ecological measurements, to really answer this question. The same goes for precipitation (observations of mountain precipitation are even worse than temperature – particularly where much of the precipitation falls in the form of snow). At the moment this is not really acknowledged.

Second in the interpretation/discussion I have a few questions/comments. At line 160 it says higher warming rates are concentrated in dry continental regions (this is probably due to the reduced thermal inertia as claimed). So this could increase the movement of isotherms uphill in these areas?.

Interestingly however these are the same regions which would have steeper lapse rates because of the dryness (the same factor) which therefore would decrease the movement of isotherms (for the same amount of warming)... so there are two compensatory effects going on? This leads me to question which effect is winning? This might be a simplistic way of thinking about it, but perhaps interesting to discuss none the less?

The other thing which is interesting is the decrease in isotherm shift with elevation. This implies that high mountains may not be as sensitive as we thought when it comes to species movements and they could decelerate as they move uphill and towards the highest elevations (which goes against a lot of the literature on enhanced mountain sensitivity?). This is of course because it gets colder and drier at the highest elevations, both of which steepen the lapse rate (and make isotherm spacing contract). It is important to note that both these conclusions are implicit in the methodology in which temperature and moisture content control lapse rate (which needs clarification when discussing).

Line 186 -this sentence structure is confusing.. The section (and before reaching Antarctica in the southern hemisphere) is essentially a sub-clause but this is not clear. Maybe put it in brackets, or remove it to make the rest of the sentence clearer. Change lower to shallower in line 189.

Referee #3 (Remarks to the Author):

Many thanks for carefully revising this manuscript in light of the extensive comments by the reviewers. I cannot really judge some of the more technical aspects of the climatic models and statistical analyses, but I find that the biological components have been properly addressed and provide very interesting and well-supported results. I am therefore happy to recommend acceptance.

Referee #4 (Remarks to the Author):

In the new version of the manuscript the authors make valuable improvements. I really appreciate that they tried to solve every single critic of the reviewers. All my critical comments were addressed. I especially thank the authors for providing now a comparison of field-measured data of LRT and predictions based on their models. Here, however, I have to say that the correlation is there, but its quite poor ($R^2 = 0.22$) and there seems even not to be a positive trend between field-measured LRT and MALRT for the lower 2/3 of MALRT axis. This again lets me doubt about the quality of the model predictions. It could be that the poor predictive power of the model is a major reason for the very low fit between climate data and upward movements of species. What stands on the pro side for the new approach is the better predictive power of MALRT then those of a fixed LRT for the shifts in species ranges. Nevertheless, from my point of view the authors should discuss this more critically (at the moment it reads as if MLRT predictions closely fit to the true field measured LRT values), i.e. that there is a very high level of noise in the correlation between MALRT and field-measured LRT; and that this could be a reason for the discrepancy between species range shifts and climatic shifts.

Minor issues:

Check the change of SALRT -> SLRT again. For example, Extended Data Fig. 3 still lists SALRT.

Lines 133-134: This validation exercise shows spatial consistency between lapse rates calculated from empirical weather station data and both our computations of the MALRT and SLRT ($R^2 = 0.22 \pm 0.01$; mean \pm standard deviation): Is the R^2 value the same for the comparison of both MALRT and SLRT? Unlikely?!

Extended Data Figure 3. (a) Scatterplot  Start with capital letter "S".

Referee #6 (Remarks to the Author):

The authors have done a nice job of addressing the outstanding points raised in the last round of reviews. I do not have any additional major comments. The following two passages did raise questions for me however. And I noted one specific typo in a figure listed below.

Line 39. Does this imply that the species are unable to move fast enough to track the rapidly shifting isotherms? As stated elsewhere in the manuscript, given that distances required to track shifting isotherms are relatively short in the mountains, species should be able to move fast enough. If other factors are causing the lag (e.g., competition with other species as suggested in the discussion), would slowing the change in temperature necessarily reduce the lag?

Line 243-246. Although it is clear to me how competition could limit species responses to shifting isotherms, I don't understand how the fact that there are fewer mountainous areas than flat areas on the planet limits mountainous species' ability to track elevational shifts in isotherms.

Extended Data Figure 2. Title for (b) reads "Water vapor"

Author Rebuttals to Second Revision:

Referees' comments:

Referee #1 (Remarks to the Author):

I can see that the authors have made substantial changes to improve the analysis in response to my previous review and I do think the paper is almost suitable for publication.

Rather than make a lot of detailed comments again, I think it worth clarifying a few arguments (the first of which other reviewers have also commented on).

The main concern is still the reliability of the lapse rate calculations. I think nearly all reviewers (and perhaps the authors?) agree that long term weather stations measuring air temperatures at 2 m on mountain slopes would be the best source to calculate slope specific lapse rates, but that they are not universally available. While it is true there are some mountain transects, it is also true that reliable long-term measurements across elevational gradients are few and far between. Extended Data Figure 3 does not convince me that MALRT is a great representation of lapse rates on the ground ($r=0.22$ is not a good predictive model!), even though it may be the best estimate available in this paper (better than constant rate or SLRT). **I therefore think this point does need to come out in the discussion which should argue for better mountain meteorological networks along elevational gradients**, preferably combined with ecological measurements, to really answer this question. The same goes for precipitation (observations of mountain precipitation are even worse than temperature – particularly where much of the precipitation falls in the form of snow). At the moment this is not really acknowledged.

Chan et al.: Many thanks for your positive comments on our revised work and for your time to provide additional suggestions for improvement. We really appreciate and value your feedback. We agree that an r^2 value of 0.22 is rather low when assessing the predictive accuracy of our mechanistic model for the MALRT against field-based measurements but this is also based on really external and independent data from a low sample size dataset ($n = 48$) which clearly lacks statistical power. Based on your advice to discuss the low predictive power of our model linking MALRT values against station-based LRT values

and to argue for better mountain meteorological networks along elevational gradients, we have revised our discussion section as follows (see lines 301-314):

“However, it is important to recognize that our thermodynamic model still suffers from a low predictive accuracy when compared with field-measurements of temperature lapse rates, which does not allow to accurately quantify local-scale lapse rates based solely on thermodynamic models. This highlights the need for refined mountain meteorological networks along elevational gradients to improve our holistic understanding of the processes underlying local temperature lapse rates along mountain slopes. Furthermore, some studies have shown that changes in precipitation patterns can affect mountain species range shifts^{15,40}, but historical data on precipitation patterns along mountain slopes is extremely scarce compared with data on temperature lapse rates. For that reason, establishing weather stations that also monitor precipitation patterns, along mountain slopes remains timely to assess the large-scale impacts of precipitation changes on mountainous organisms. We urge the establishment of monitoring networks for climate change and its impacts in mountain biodiversity hotspots, especially within mountains threatened by high velocities of isotherm shifts, which we identified in our study.”

Second in the interpretation/discussion I have a few questions/comments. At line 160 it says higher warming rates are concentrated in dry continental regions (this is probably due to the reduced thermal inertia as claimed). So this could increase the movement of isotherms uphill in these areas?. Interestingly however these are the same regions which would have steeper lapse rates because of the dryness (the same factor) which therefore would decrease the movement of isotherms (for the same amount of warming)... so there are two compensatory effects going on? This leads me to question which effect is winning? This might be a simplistic way of thinking about it, but perhaps interesting to discuss none the less?

Chan et al.: Indeed, in dry mountain regions, there is both an increase in temperature and a steeper lapse rate that are two compensatory effects on climate velocities. Hence, according to your suggestions, we have revised the discussion section and added this interpretation as follows (see lines 178-192):

“We further compared the impacts of high warming rates and steep temperature lapse rates, which act as compensatory effects on climate velocities, between arid and more humid regions. We found that in

arid mountain regions with low water vapour pressure, the temperature lapse rate accounts for 3.6% of the observed variation in climate velocity, while changes in surface temperature account for 96.4% of the observed variation, based on the random forest analysis. A detailed analysis using the Shapley value further revealed that steeper lapse rates have a smaller negative impact on climate velocities compared to higher warming rates which increase climate velocities (Extended data Fig. 4a). In humid regions, the temperature lapse rate accounts for 11.32% of the observed variation in climatic velocity, while changes in surface temperature explain 88.68% of the observed variation, based on the random forest analysis. Shapley value analysis showed that steeper lapse rates still have a smaller negative effect on climate velocities than higher warming rates (Extended data Fig. 4b). Importantly, the explanatory power of the lapse rate in wet mountains is relatively higher than in arid mountains. This difference is likely due to the lower magnitude of the surface temperature increase in wetter mountains (Extended data Fig. 4c and 4d)."

The other thing which is interesting is the decrease in isotherm shift with elevation. This implies that high mountains may not be as sensitive as we thought when it comes to species movements and they could decelerate as they move uphill and towards the highest elevations (which goes against a lot of the literature on enhanced mountain sensitivity?). This is of course because it gets colder and drier at the highest elevations, both of which steepen the lapse rate (and make isotherm spacing contract). It is important to note that both these conclusions are implicit in the methodology in which temperature and moisture content control lapse rate (which needs clarification when discussing).

Chan et al.: Thanks for pointing this out. We have revised our paper according to your suggestions as follows (see lines 258-272):

“Our results suggest that the vertical distance between isotherms in mountains is a critical factor driving species migration. Likewise, based on thermodynamic theory, colder and drier conditions at higher elevations make temperature lapse rates steeper, which in turn leads to a contraction of the vertical distance separating isotherms (i.e., isotherm spacing contracts when projected on the vertical axis) generating lower vertical velocities of isotherm shifts. This suggests that in many mountain regions, the vertical shift of isotherms decreases with increasing elevation. From the perspective of isotherms shifting upslopes due to warming, higher elevations will experience a slower rate of isotherm shift, meaning that organisms can reach habitats with suitable temperatures by moving shorter vertical distances. However,

a steeper temperature lapse rate also means that the environment changes more rapidly with elevation. Therefore, in the case of mountains with a broader base and narrower peaks²⁸, warming may result in a reduction of habitat area for organisms. Because the shape of a mountain affects the amount of habitat available to organisms²⁸, understanding the velocity of climate change, in addition to quantifying suitable habitat area under warming conditions, will be critical to understanding the effects of climate change on mountain biodiversity.”

Line 186 -this sentence structure is confusing.. The section (and before reaching Antarctica in the southern hemisphere) is essentially a sub-clause but this is not clear. Maybe put it in brackets, or remove it to make the rest of the sentence clearer. Change lower to shallower in line 189.

Chan et al.: Modified as suggested.

Referee #3 (Remarks to the Author)

Many thanks for carefully revising this manuscript in light of the extensive comments by the reviewers. I cannot really judge some of the more technical aspects of the climatic models and statistical analyses, but I find that the biological components have been properly addressed and provide very interesting and well-supported results. I am therefore happy to recommend acceptance.

Chan et al.: Thank you very much for your positive feedback on our revised work.

Referee #4 (Remarks to the Author):

In the new version of the manuscript the authors make valuable improvements. I really appreciate that they tried to solve every single critic of the reviewers. All my critical comments were addressed.

I especially thank the authors for providing now a comparison of field-measured data of LRT and predictions based on their models. Here, however, I have to say that the correlation is there, but its quite poor ($R^2 = 0.22$) and there seems even not to be a

positive trend between field-measured LRT and MALRT for the lower 2/3 of MALRT axis. This again lets me doubt about the quality of the model predictions. It could be that the poor predictive power of the model is a major reason for the very low fit between climate data and upward movements of species. What stands on the pro side for the new approach is the better predictive power of MALRT than those of a fixed LRT for the shifts in species ranges. Nevertheless, from my point of view the authors should discuss this more critically (at the moment it reads as if MALRT predictions closely fit to the true field measured LRT values), i.e. that there is a very high level of noise in the correlation between MALRT and field-measured LRT; and that this could be a reason for the discrepancy between species range shifts and climatic shifts.

Chan et al.: Thank you very much for your helpful and insightful comment which echoes the comment from reviewer #1 on acknowledging the low predictive accuracy of our thermodynamic model. We very much agree with that and we did our best to revise the discussion section so that it better acknowledges this point. We have revised our discussion to emphasize that our paper focuses on a qualitative comparison at the global extent and provides a better assessment for understanding the effects of climate change on species range shifts. However, our results cannot replace local-scale data on the lapse rate of temperature as obtained from a denser network of weather stations along mountain slopes. For a detailed response, please refer to our first response to reviewer #1. We hope that it addresses your last concern regarding our work.

Minor issues:

Check the change of SALRT -> SLRT again. For example, Extended Data Fig. 3 still lists SALRT.

Chan et al.: Ok, done. See the new version of Extended Data Fig. 3.

Lines 133-134: This validation exercise shows spatial consistency between lapse rates calculated from empirical weather station data and both our computations of the MALRT and SLRT ($R^2 = 0.22 \pm 0.01$; mean \pm standard deviation): Is the R^2 value the same for the comparison of both MALRT and SLRT? Unlikely?!

Chan et al.: Sorry for the confusion. Our initial analysis involved including MALRT, SLRT, longitude, and latitude altogether in a single Random Forest statistical model. Therefore, the R^2 value of 0.22 refers to the entire statistical model's R^2 . This is also why we originally stated "the MALRT has the highest explanatory power ($32.71 \pm 0.48\%$) in relation to station-based LRT compared to other variables". To avoid any potential confusion for the reader, we have now switched to running two separate simple linear regressions, one for MALRT and one for SLRT. By doing so, it clarifies the message, avoiding any potential confusion, and it allows us to compare the explanatory power of station-based LRT between MALRT and SLRT. Currently, the R^2 for MALRT is 0.11, while for SLRT it is 0.02. We have modified our text to make it clearer as follows (see lines 134-140):

"This validation exercise shows that there are very few mountain regions worldwide where the network of weather stations is dense enough along mountain slopes ($n > 2$) to compute the LRT. Nevertheless, we found a positive relationship between station-based LRT calculated from these very limited networks of weather station data and our computations of the MALRT (linear regression, $F_{1,46} = 5.54$, $p = 0.02$, $R^2 = 0.108$, $n = 48$, Extended Data Fig. 3a). In contrast, the relationship between SLRT and station-based LRT did not reach statistical significance (linear regression, $F_{1,46} = 0.774$, $p = 0.38$, $R^2 = 0.017$, $n = 48$, Extended Data Fig. 3b)."

Extended Data Figure 3. (a) Scatterplot  Start with capital letter "S".

Chan et al.: Modified as suggested.

Referee #6 (Remarks to the Author):

The authors have done a nice job of addressing the outstanding points raised in the last round of reviews. I do not have any additional major comments. The following two passages did raise questions for me however. And I noted one specific typo in a figure listed below.

Line 39. Does this imply that the species are unable to move fast enough to track the

rapidly shifting isotherms? As stated elsewhere in the manuscript, given that distances required to track shifting isotherms are relatively short in the mountains, species should be able to move fast enough. If other factors are causing the lag (e.g., competition with other species as suggested in the discussion), would slowing the change in temperature necessarily reduce the lag?

Chan et al.: We agree with your comment. We have removed this sense and replaced it with the following conclusion (see L39-42):

“Our research suggests that studying the impacts of climate change on isothermal shifts in mountain regions, and how it affects suitable habitat for mountain species, will provide critical information for global mountain biodiversity conservation strategies, especially so in the 17 mountain regions exhibiting high vertical velocities.”

Line 243-246. Although it is clear to me how competition could limit species responses to shifting isotherms, I don't understand how the fact that there are fewer mountainous areas than flat areas on the planet limits mountainous species' ability to track elevational shifts in isotherms.

Chan et al.: Sorry for the confusion, we have modified the text as follows:

“From the perspective of isotherms shifting upslopes due to warming, higher elevations will experience a slower rate of isotherm shift, meaning that organisms can reach habitats with suitable temperatures by moving shorter vertical distances. However, a steeper temperature lapse rate also means that the environment changes more rapidly with elevation. Therefore, in the case of mountains with a broader base and narrower peaks²⁸, warming may result in a reduction of habitat area for organisms. Because the shape of a mountain affects the amount of habitat available to organisms²⁸, understanding the velocity of climate change, in addition to quantifying suitable habitat area under warming conditions, will be critical to understanding the effects of climate change on mountain biodiversity.” (L264-272)

“As described above, the available habitat area for organisms in mountain regions is influenced by the shape of the mountain, and many mountains exhibit a reduction in area with increasing elevation. This,

combined with biotic interactions like interspecific competition^{29,30}, might collectively limit the ability of mountain species to track isotherm shifts in the future.” (L279-283)

Extended Data Figure 2. Title for (b) reads “Water vapor”

Chan et al.: Modified as suggested.

Reviewer Reports on the Third Revision:

Referees' comments:

Referee #1 (Remarks to the Author):

I can see that the authors have taken my previous comments and considered them carefully and I now think that the reflection and discussion is much improved.

I have one remaining comment: the analysis of the compensatory effects of lapse rate and absolute rate of temperature change on the isotherm shift (which I suggested) is interesting in that it suggests that nearly all of the variance in climate velocity (or is it difference between arid and humid areas?) is explained by the latter factor (i.e. faster rates of warming in dry continental locations). This in turn suggests that the lapse rate is rather inconsequential and I am left thinking why spend a whole paper on developing better lapse rate estimates then?. I know this is for a very specific humid vs arid comparison (and perhaps not the data in general). Is this a fair interpretation or have I somehow misinterpreted the discussion?. If I have misinterpreted it (very likely!), then other readers might... and it might be worth clarifying why the lapse rate is still important overall (which it clearly is because you get different maps for SLRT vs MALRT etc)?.

Otherwise I think this is much improved and I have no other comments.

Referee #4 (Remarks to the Author):

Thanks to the authors for another round of revising and improving the paper. My main concern was the poor performance of their climatic model (MALRT, SLRT) for predicting temperature data measured in the field (Lines132-143), i.e. the validation of the climatic model ($r^2 < 0.22$ depending on the complexity of the model). I think that the authors improved this section once more and I really appreciate the open way they discuss the poor predictive power of the models in the discussion. Concerning the model validations (with temperature data from the field) I still have two points/advice:

1) The authors downloaded data from 144 weather station, which seems to be all which is available in the cited database for mountain regions. This is really not much. However, it does not need a weather station to get estimates of mean annual temperature. In many field studies, temperature along elevation gradients is recorded with temperature or temperature/humidity data loggers and much of this data is published. Did you also check these resources? In the two study mountain areas (two elevation gradients) where I work mean annual temperature records of more than 70 study sites were already published, i.e. there is a dense network of temperature data loggers which could be used beneath those of weather stations. Maybe this improves the predictive power of your models.

2) You compared the r-square values of the SLRT and the better MALRT approach. It would make sense here also to show the r-square for an approach based on a constant lapse rate of 5.5°C/km (the classical approach).

Maybe these points could help to improve this critical aspect of the paper.

All other (minor) issues have been clarified by the author team.

Author Rebuttals to Third Revision:

Referees' comments 1:

Referee #1 (Remarks to the Author):

I can see that the authors have taken my previous comments and considered them carefully and I now think that the reflection and discussion is much improved.

I have one remaining comment: the analysis of the compensatory effects of lapse rate and absolute rate of temperature change on the isotherm shift (which I suggested) is interesting in that it suggests that nearly all of the variance in climate velocity (or is it difference between arid and humid areas?) is explained by the latter factor (i.e. faster rates of warming in dry continental locations). This in turn suggests that the lapse rate is rather inconsequential and I am left thinking why spend a whole paper on developing better lapse rate estimates then? I know this is for a very specific humid vs arid comparison (and perhaps not the data in general). Is this a fair interpretation or have I somehow misinterpreted the discussion? If I have misinterpreted it (very likely!), then other readers might... and it might be worth clarifying why the lapse rate is still important overall (which it clearly is because you get different maps for SLRT vs MALRT etc)? Otherwise I think this is much improved and I have no other comments.

Author response 1:

Chan et al.: We wish to thank again reviewer #1 for the very positive comments on our last revisions. About the last remaining comment from reviewer #1, it is important to keep in mind that our main goal is to estimate the correct lapse rate because this is the key to accurately estimate the velocity at which isotherms are moving upslope as climate warms. Obtaining the correct lapse rate is critical to our understanding of the impacts of climate change on mountain biota. As the reviewer rightly points out, in terms of mechanisms, changes in surface temperature over time, i.e., the warming rate, actually explain more variation in the vertical velocity of isotherm shifts than the lapse rate itself. We are grateful for the reviewer's earlier suggestions, which allowed us to analyze this mechanism more clearly. However, we believe that this does not diminish the importance of discussing the lapse rate. Indeed, the fact that the lapse rate as a relatively minor effect on the computed velocity compared with the absolute change in temperature conditions over time does not mean it is inconsequential. Besides, the explanatory power of the lapse rate in wet mountains (ca. 12%) is almost four times higher than in arid mountains (ca. 4%) which suggests strong spatial variation in the relative importance of the lapse rate. Hence, we need to use the correct lapse rate to capture the spatial variation in the vertical velocity of isotherm shifts in mountain areas that we can latter use as a yardstick to evaluate whether or not biodiversity is tracking the vertical velocity of isotherm shifts, which is the main motivation behind this paper. As suggested by reviewer #1, we added a sentence (see lines 187-190) to clarify why the LRT still matters even though its explanatory power is relatively low in comparison with the warming rate. More specifically, we wrote: "Although the explanatory power of the lapse rate is, in general, relatively much lower than the one of the warming rate, the striking differences we found, in terms of the relative importance, between arid and humid regions impacts the spatial variation we report in the vertical velocity of isotherm shifts."

Referees' comments 2:

Referee #4 (Remarks to the Author):

Thanks to the authors for another round of revising and improving the paper. My main concern was the poor performance of their climatic model (MALRT, SLRT) for predicting temperature data measured in the field (Lines 132-143), i.e. the validation of the climatic model ($r^2 < 0.22$ depending on the complexity of the model). I think that the authors improved this section once more and I really appreciate the open way they discuss the poor predictive power of the models in the discussion.

Concerning the model validations (with temperature data from the field) I still have two points/advice:

1) The authors downloaded data from 144 weather stations, which seems to be all which is available in the cited database for mountain regions. This is really not much. However, it does not need a weather station to get estimates of mean annual temperature. In many field studies, temperature along elevation gradients is recorded with temperature or temperature/humidity data loggers and much of this data is published. Did you also check these resources? In the two study mountain areas (two elevation gradients) where I work mean annual temperature records of more than 70 study sites were already published, i.e. there is a dense network of temperature data loggers which could be used beneath those of weather stations. Maybe this improves the predictive power of your models.

Author response 2:

Chan et al.: Yes, following earlier suggestions from Reviewer #4, we initially searched for elevational transects. Recognizing that humidity data is scarcer compared to temperature data, we included 'humidity' in our keyword search criteria. Specifically, we searched for 'elevational transect relative humidity' on Google Scholar on October 18, 2023, resulting in a total of 20,300 records. Although many studies mentioned the availability of raw data, we faced challenges with outdated links to access the data itself. Among the first 50 records, we identified six valid datasets. However, we noted a significant geographic bias in these available transects; they were predominantly collected in humid regions and focused on specific taxa such as mosses, lichens, ferns, bryophytes, and amphibians, hence strongly biased towards microclimatic conditions near the ground. This led us to pivot towards a more systematic data source: weather station-based data. Weather station data represents local weather conditions or synoptic conditions that are representative of the macroclimate, in contrast to miniature logger data, which represents microclimate conditions (usually near the ground, below the canopy or even in the topsoil layer) and is influenced by factors such as vegetation cover, canopy height, canopy structure and microtopography. For instance, the global buffering capacity of forest canopies can reach between 4 and 10°C between temperate and tropical forests (De Frenne et al. 2019). Considering this fundamental distinction between macroclimatic and microclimatic conditions, we decided to concentrate our analysis on data from weather stations because data from weather stations better match, in essence, with the MALRT and SLRT we computed here. Indeed, our estimates of the MALRT and LRT reflect macroclimatic temperature conditions or surface temperature, which is the temperature above the canopy layer, while temperature records from data loggers usually reflect microclimatic conditions close to the

ground and most often in the understory and even sometimes in the topsoil layer. Therefore, there is a striking difference for which we assume very different lapse rates if we were using microclimatic conditions near the ground or in the ground. This is a very interesting idea to use temperature records from the increasing availability of logger data but also a very different story as well for another study. Hence, we prefer not to mix microclimate data with macroclimate here. But we will very likely play with microclimatic data in the future to compute a lapse rate that will be relevant for what is happening near or inside the soil surface where most of the biodiversity lies (cf. ground-dwelling arthropods for instance).

Reference:

De Frenne, P., Zellweger, F., Rodríguez-Sánchez, F. *et al.* Global buffering of temperatures under forest canopies. *Nat Ecol Evol* **3**, 744–749 (2019). <https://doi.org/10.1038/s41559-019-0842-1>

Referees' comments 3:

2) You compared the r-square values of the SLRT and the better MALRT approach. It would make sense here also to show the r-square for an approach based on a constant lapse rate of $5.5^{\circ}\text{C}/\text{km}$ (the classical approach). Maybe these points could help to improve this critical aspect of the paper.

Author response 3:

Chan et al.: We were certainly intrigued by the comparison you proposed. However, upon comparing the station-based LRT with the constant LRT, we encountered a lack of variation on the x-axis, rendering linear regression analysis not very meaningful. To demonstrate this problem, the plot below is presented as an example using $5.5^{\circ}\text{C}/\text{km}$ for the constant LRT, where it becomes evident that the data points are aligned vertically.

Additional note from the authors (Chan et al.):

During the revision process, we noticed a labeling mistake of the variables in Extended Data Fig. 5a and 5b. This mistake has now been rectified, ensuring the variables are correctly ordered. It's important to note that the corresponding statement in lines 228-229 remains unchanged. It states: "...our analysis shows that the MALRT has much greater explanatory power for predicting the velocities of species range shifts than the SLRT (Supplementary Results and Extended Data Fig. 5)."

Previous incorrect Extended Data Fig. 5

The corrected Extended Data Fig. 5 with the raw data points as requested by the editor. The right panel in (a) and the middle panel in (b) has been corrected.